



**Spatially-coordinated airborne data and complementary products for aerosol, gas, cloud, and meteorological studies:**

**The NASA ACTIVATE dataset**

Armin Sorooshian[1,2,3], Mikhail D. Alexandrov[4,5], Adam D. Bell[6], Ryan Bennett[7], Grace Betito[2], Sharon P. Burton[8], Megan E. Buzanowicz[6,8], Brian Cairns[4], Eduard V. Chemyakin[6,8], Gao Chen[8], Yonghoon Choi[6,8], Brian L. Collister[8], Anthony L. Cook[8], Andrea F. Corral[1], Ewan C. Crosbie[6,8], Bastiaan van Diedenhoven[9], Joshua P. DiGangi[8], Glenn S. Diskin[8], Sanja Dmitrovic[3], Eva-Lou Edwards[1], Marta A. Fenn[6,8], Richard A. Ferrare[8], David van Gilst[7], Johnathan W. Hair[8], David B. Harper[8], Miguel Ricardo A. Hilario[2], Chris A. Hostetler[8], Nathan Jester[8], Michael Jones[6,8], Simon Kirschler[10,11], Mary M. Kleb[8], John M. Kusterer[8], Sean Leavor[6,8], Joseph W. Lee[8], Hongyu Liu[12], Kayla McCauley[2], Richard H. Moore[8], Joseph Nied[8], Anthony Notari[8], John B. Nowak[8], David Painemal[6,8], Kasey E. Phillips[8], Claire E. Robinson[6,8], Amy Jo Scarino[6,8], Joseph S. Schlosser[6,13], Shane T. Seaman[8], Chellappan Seethala[14], Taylor J. Shingler[8], Michael A. Shook[8], Kenneth A. Sinclair[4,5], William L. Smith Jr.[8], Douglas A. Spangenberg[6,8], Snorre A. Stamnes[8], Kenneth L. Thornhill[6,8], Christiane Voigt[10,11], Holger Vömel[15], Andrzej P. Wasilewski[4], Hailong Wang[16], Edward L. Winstead[6,8], Kira Zeider[1], Xubin Zeng[2], Bo Zhang[12], Luke D. Ziemba[8], Paquita Zuidema[14]

[1]Department of Chemical and Environmental Engineering, University of Arizona, Tucson, AZ, USA
[2]Department of Hydrology and Atmospheric Sciences, University of Arizona, Tucson, AZ, USA
[3]James C. Wyant College of Optical Sciences, University of Arizona, Tucson, AZ, USA
[4]NASA Goddard Institute for Space Studies, New York, NY, USA
[5]Department of Applied Physics and Applied Mathematics, Columbia University, New York, NY, USA
[6]Science Systems and Applications, Inc., Hampton, VA, USA
[7]Bay Area Environmental Research Institute, NASA Ames Research Center, Moffett Field, CA, USA
[8]NASA Langley Research Center, Hampton, VA, USA
[9]SRON Netherlands Institute for Space Research, Leiden, the Netherlands
[10]Institute of Atmospheric Physics, German Aerospace Center, Germany
[11]Institute of Atmospheric Physics, University Mainz, Germany
[12]National Institute of Aerospace, Hampton, VA, USA
[13]NASA Postdoctoral Program, NASA Langley Research Center, Hampton, VA, USA
[14]Rosenstiel School of Marine, Atmospheric, and Earth Science, University of Miami, Miami, FL, USA
[15]National Center for Atmospheric Research, Boulder, CO, USA
[16]Atmospheric Sciences and Global Change Division, Pacific Northwest National Laboratory, Richland, WA, USA

*Correspondence to*: Armin Sorooshian (armin@arizona.edu)



**Abstract.** The NASA Aerosol Cloud meTeorology Interactions oVer the western ATlantic Experiment (ACTIVATE) produced a unique dataset for research into aerosol-cloud-meteorology interactions with applications extending from process-based studies to multi-scale model intercomparison and improvement, and remote sensing algorithm assessments and advancements. ACTIVATE used two NASA Langley Research Center aircraft, a HU-25 Falcon and King Air, to conduct systematic and spatially coordinated flights over the northwest Atlantic Ocean amounting to 162 joint flights and 17 other single-aircraft flights between 2020 and 2022 across all seasons. Data cover 574 and 592 cumulative flights hours for the Falcon and King Air, respectively. The HU-25 Falcon flew conducted profiling at different level legs below, in, and just above boundary layer clouds (<3 km) and obtained in situ measurements of trace gases, aerosol particles, clouds, and atmospheric state parameters. In cloud-free conditions, the Falcon similarly conducted profiling at different level legs within and immediately above the boundary layer. The King Air (the high-flyer) flew at approximately ~9 km conducting remote sensing with a lidar and polarimeter while also launching dropsondes. Collectively, simultaneous data collected from both aircraft help characterize the same vertical column of the atmosphere. In addition to individual instrument files, data from the Falcon aircraft are combined into "merge files" on the publicly available data archive that are created at different time resolutions of interest (e.g., 1, 5, 10, 15, 30, 60 s, or matching an individual data product start and stop times). This paper describes the ACTIVATE flight strategy, instrument and complementary dataset products, data access and usage details, and data application notes.



## 1 Introduction

Aerosol-cloud interactions are responsible for the largest uncertainty in estimates of total anthropogenic radiative forcing (Bellouin et al., 2020). This uncertainty stems partly from the difficulty in experimentally characterizing such interactions in the atmosphere due to the need for airborne platforms. Also, it is challenging to isolate the relative influence of different factors that impact the life cycle and properties of clouds including meteorology and aerosol particles. Decades of airborne field studies focused on aerosol-cloud interactions have been limited in terms of data volume and number of variables measured,

diversity of aerosol and weather conditions, and vertical data coverage. These limitations motivated the conception of the NASA Aerosol Cloud meTeorology Interactions oVer the western ATlantic Experiment (ACTIVATE), which included systematic, extensive, and spatially-coordinated flights with two aircraft over the northwest Atlantic (Sorooshian et al., 2019). ACTIVATE is one of five Earth Venture Suborbital-3 (EVS-3) missions.

ACTIVATE flights were strategically executed in different seasons to increase the dynamic range of aerosol and

meteorological conditions that resulted in different cloud types spanning warm and mixed-phase clouds, and the continuum from stratiform to cumulus clouds. The northwest Atlantic differs from subtropical regions often chosen for aerosol-cloud interaction campaigns due to multiple cloud types within reach, rather than the stratocumulus clouds that are simpler to characterize owing to their high cloud fraction and well-defined vertical structure. With a disciplined strategy of conducting the same type of flight plan for over 90% of the flights (called "statistical surveys"), data were repeatedly collected at different

vertical levels in and above the marine boundary layer, including within and immediately below and above clouds. Another subset of flights called "process studies" comprised more customized flight patterns to capitalize on targets of opportunity for remote sensing algorithm assessments and detailed model intercomparison studies such as with wintertime cold air outbreaks and summertime developing cumulus clouds. This rich dataset is ideal for a number of research applications including studying processes, model evaluation and improvement, parameterization development, and remote sensing algorithm analysis and

advancement.

To aid the research community in the usage of the ACTIVATE data, the goal of this work is to provide a guide for users. The structure of this paper is as follows: (i) a description of the ACTIVATE campaign and flight strategy, which involved spatial coordination between a high-flying King Air and a low-flying HU-25 Falcon; (ii) summary of King Air instruments and associated datasets; (iii) summary of Falcon instruments and associated datasets; (iv) description of complementary data

products; (v) visualization of data products relevant to a representative case study flight; (vi) data/code availability and file format; and (vii) conclusions. To guide readers, Appendix A has a nomenclature table defining all acronyms and abbreviations used in this paper.

## 2 Field campaign description

### 2.1 Objectives, operations bases, and schedule



ACTIVATE generated a novel dataset that can be used to address three overarching objectives: (i) quantify relationships amongst aerosol particle number concentration ($N_a$), cloud condensation nuclei (CCN) concentration, and cloud drop number concentration ($N_d$), and reduce uncertainty in model parameterizations of aerosol activation and cloud formation; (ii) improve process-level understanding and model representation of factors that govern cloud micro/macro-physical properties and how
they couple with cloud effects on aerosol; and (iii) assess advanced remote sensing capabilities for retrieving aerosol and cloud properties related to aerosol-cloud interactions. To achieve these objectives, it was important to conduct a high number of flights across different seasons to collect sufficient statistics across a range of aerosol, cloud, and meteorological conditions for more robust calculations relevant to understanding the life cycle and properties of different types of boundary layer clouds (e.g., stratiform and cumulus; mixed-phase and warm clouds). To address the challenge of needing data for different vertical
levels relevant to the aerosol-cloud system and to achieve remote sensing objectives, two aircraft were employed that were kept highly coordinated in both space and time. These planes included the NASA Langley Research Center's HU-25 Falcon (low flyer, < 3 km) and King Air (high flyer, ~9 km). A critical element in the selection of the two aircraft was that both aircraft flew close to 120 m s$^{-1}$ at their respective sampling altitudes. The flights were limited by the endurance of the aircraft (< 4 hours) and so flights were designed to try to extend the spatial range as much as possible while also still being able to
characterize different vertical levels. This resulted in an approach of flying "statistical surveys" comprised of repeated "ensembles" that we describe below (sect. 2.2) and that have been discussed in detail elsewhere for ACTIVATE flights (Dadashazar et al., 2022b).

The northwest Atlantic study region is ideal for ACTIVATE objectives owing to the wide range of aerosol types and weather conditions (Corral et al., 2021; Painemal et al., 2021; Sorooshian et al., 2020) during the periods that flights would take place,
which ended up including November-June and August-September. Flights were mostly based out of NASA Langley Research Center (NASA LaRC) with only a few others based out of secondary bases, including Newport News-Williamsburg International Airport (Virginia), Quonset State Airport (Rhode Island), Rhode Island T.F. Green International Airport (Rhode Island), and L. F. Wade International Airport (Bermuda). The original goal for flights was to do 25 joint flights in each of 6 deployments between 2020 and 2022, including a Winter (February-March) and Summer (May-June) deployment each year.
As a result of operational delays, aircraft maintenance challenges, and COVID-19 emerging during the first deployment, deviations were necessary relative to the original plan. These deviations are evident in Table 1, which shows a summary of flight metrics for each of the six deployments. Table 2 further summarizes each individual flight, including details specific to each aircraft such as takeoff and landing time, and special features per flight. Figure 1 shows the flight tracks each year for the Falcon and King Air.

## 2.2 Flight strategy

The original goal of ACTIVATE was to allocate 90% of the flights to "statistical surveys" whereby the two aircraft would repeatedly conduct coordinated cloud and cloud-free ensembles (Fig. 2). Cloud ensembles performed by the lower-flying



Falcon included flying level legs (~3 min each unless otherwise dictated by flight conditions) in the following nominal order:
below cloud base (BCB), above cloud base (ACB), a second pair of BCB and ACB, minimum altitude (MinAlt), above cloud
top (ACT), below cloud top (BCT). MinAlt is defined as the lowest altitude the aircraft could fly at, which was ~150 m above
sea level when clear of cloud and in good visibility conditions. The slant ascent from MinAlt to ACT provided multiple in situ
vertical profiles across the range of relevant altitudes and included periods of cloudy and cloud-free sampling depending on
conditions. A caveat with the interpretation of these "vertical" profiles is that in environments with spatially varying conditions
(e.g., broken or episodic cloud), the slant ascent may not represent average conditions with any reliability. Clear ensembles in
cloud-free conditions included legs in the following nominal order: MinAlt, above boundary layer top (ABL), below boundary
layer top (BBL), Remote Sensing (RS) leg. The RS leg was implemented under conditions of high aircraft coincidence and
when no clouds affected the field of view. The RS leg provided a second low-altitude leg (~230 m) to help with lidar extinction
comparison in the challenging near-surface region. The altitude of the ABL leg was estimated by flight scientists based on
gradients in the available real-time data during ascents and descents. Occasionally deviations occurred to these leg orders for
both ensemble types based on atmospheric conditions and air traffic control challenges requiring changes in altitude. The time
span (distance) of each leg and cloud ensemble was ~3.3 min (~24 km) and ~35 min (~250 km), respectively, while clear
ensembles were typically ~15 min and ~100 km (Dadashazar et al., 2022b). Across 162 final joint flights, all but 12 were
classified as statistical surveys (93%), with classifications of each flight shown in Table 2. An archived forward camera video
from the HU-25 Falcon on a representative statistical survey is accessible at this link to show data users how the ensembles
appeared visually from the perspective of the aircraft: https://asdc.larc.nasa.gov/news/activate-data-webinar-materials. A
representative statistical survey flight is discussed in more detail in sect. 6.

The remaining 10% of flights were intended to be "process study" flights, with their number reduced to 12 out of 162 (7%) in
practice. The goal of these flights was to focus on a target of opportunity with more detailed characterization in one location
of a particularly interesting cloud scene. Four of the 12 process studies were conducted during wintertime cold air outbreak
events, with the remaining eight focused on summertime cumulus cloud fields. These flights typically entailed more detailed
vertical characterization in the same atmospheric column with the Falcon conducting stacked legs below, in, and above clouds
(often termed a "wall" pattern) with bounding vertical soundings at the beginning and end of the wall(s). During that time, the
upper-flying King Air would conduct a carefully designed module at high altitude to maximize coordination, but also to provide
detailed information about the scene that the clouds of interest were evolving in. For example, during some winter process
studies, the King Air conducted a large circle aloft with numerous launched dropsondes to derive relevant quantities such as
divergence profiles and surface fluxes to be used for model intercomparison studies (Chen et al., 2022a; Seethala et al., 2021;
Li et al., 2022). A visual representation of a generic process study flight is shown in Fig. 3. Note that the aircraft would still
conduct ensembles (Fig. 2) during process study flights during transits to and from the key area of focus where a "wall pattern"
would be conducted.

## 2.3 Recommended terminology



The following guidelines are encouraged when reporting information about specific flights based on information in Table 2. References should provide the RF number and date. In cases of two flights on a given day, one can additionally include "L1"
and/or "L2" to signify launch 1 and 2, respectively. Since each flight has a unique RF number, the launch number becomes more important if only flight dates are used without reference to the RF number. Therefore, examples include: "RF1 (14 February 2020)"; "RF6 (22 February 2020) or "22 February 2020, L2". Furthermore, it is encouraged to refer to the six deployments according to their season and year for simplicity (e.g., Winter 2020, Summer 2020, Winter 2021, Summer 2021, Winter 2022, Summer 2022) as shown in Table 1, with the caveat that Winter 2022 still includes November-December flights
occurring in 2021. This is encouraged for simplicity even though the months of flights do not perfectly align with typical seasonal definitions (e.g., DJF = winter, JJA = summer).

**2.4 Special flight details**

A few special features are worth expanding on that impacted flight execution:

(i)     Single aircraft flights (17 in total) were conducted when one of the aircraft remained grounded, usually for a maintenance issue. In rare cases such as RF177 (16 June 2022), both planes began a joint flight, but one plane (Falcon in this case) experienced a maintenance issue during flight and returned to base without any science data archived. This meant the flight qualified as a single aircraft flight as only the King Air obtained archivable data. For single Falcon flights, statistical surveys were usually conducted with one process study flight; RF163 on 2
June 2022 was a unique process study flight in that it was conducted with the Falcon alone and involved wall patterns. The King Air also conducted its usual flight strategy in single aircraft flights, flying aloft around ~9 km and sampling targets of opportunity that were deemed to be too important to miss, even in the absence of the Falcon, such as cold air outbreaks (e.g., RF42 on 29 January 2021).

       (ii)    Flights based out of either NASA Langley Research Center or Newport News-Williamsburg International Airport
170             almost always included transits to one of two waypoints (ZIBUT [36.938° N, 72.666° W] or OXANA [34.363° N, 73.759° W]) to adhere to strict air traffic control restrictions, beyond which farther offshore there was more flexibility for waypoint selection. Those two waypoints can be thought of as 'pivot-points' that are visually evident and labeled in Fig. 1. A few flights included transits from one of the two Virginia bases to the northeast to waypoint ZIZZI (38.941° N, 74.529° W; shown in Fig. 1) to strategically sample upwind conditions in cold
175             air outbreaks. Due to limitations associated with the COVID-19 pandemic in the first four deployments (2020-2021), secondary bases for the purpose of extending ACTIVATE's spatial range were only used in deployments 5-6 in 2022.

               Notable was a series of flights based in Bermuda in June 2022 to make up for not flying there earlier in the campaign. The rationale for data collection around Bermuda was multifold: (i) farther removed from continental
180             pollution sources and thus closer resembling a remote marine aerosol regime; (ii) conditions simplify parsing out causal drivers for aerosol-cloud interactions (e.g., less impacted by terrestrial boundary layer & Gulf Stream





effects). The coastal region by the mid-Atlantic states has a strong airmass disequilibrium (e.g., high air-sea contrasts), but farther downwind airmasses relax to a more (quasi-) steady state, which has more global relevance than coastal regions; (iii) connect aircraft measurements with long-term surface measurements conducted at

Bermuda (Sorooshian et al., 2020), including notable long-term aerosol and precipitation datasets collected through the Bermuda Institute of Ocean Sciences with demonstrated utility for ACTIVATE as shown in recent studies (Aldhaif et al., 2021; Dadashazar et al., 2021a); and (iv) bridge the gap for aerosol-cloud studies done in polluted conditions versus low-CCN conditions observed during missions like the North Atlantic Aerosols and Marine Ecosystems Study (NAAMES) (Behrenfeld et al., 2019) and the Aerosol and Cloud Experiments in the

Eastern North Atlantic (ACE-ENA) (Wang et al., 2022).

(iii)      Numerous flights were conducted directly underneath satellites to achieve remote sensing objectives. Six and eleven of these 'underflights' of satellites were conducted in coordination with the Advanced Spaceborne Thermal Emission and Reflection Radiometer (ASTER) and Cloud-Aerosol Lidar and Infrared Pathfinder Satellite Observations (CALIPSO), respectively. In a few instances, the two aircraft coordinated to observe

aerosol particles in clear sky conditions with the complete set of remote sensing polarimeter and lidar data with a matching full vertical profile of in-situ observations. This type of aircraft observation module, which previously did not exist in any known aircraft dataset, and that must include an ascent/descent or spiraling aircraft pattern by the in-situ aircraft, became known as "unicorn aerosol modules". This name stuck thanks to the artwork of a team member's elementary schooler. These modules included the Falcon conducting a vertical spiral sounding

with a slower climb rate (2-5 m s$^{-1}$) from its lowest possible altitude (usually ~120-150 m) to usually upwards of 5 km to reach the ceiling of high aerosol loadings, with the King Air flying aloft as it normally does. These modules targeted cloud-free scenes with relatively high aerosol concentrations to address aerosol optical and microphysical property remote sensing objectives, with a demonstration of results reported by Schlosser et al. (2022). Examples are associated with RF28 (26 August 2020), RF29 (28 August 2020), RF130 (2 March 2022),

RF131 (3 March 2022), RF144 (26 March 2022) and RF155 (17 May 2022). Although not labelled as unicorn modules in Table 2, several spiral profiles were conducted with the Falcon just offshore of the Tudor Hill Marine Atmospheric Observatory during the set of Bermuda flights in June 2022 with the King Air flying overhead; these profiles sometimes included cloud (e.g., RF169 on 8 June 2022, RF178 on 17 June 2022) and were farther removed from the polluted eastern coast of the U.S. However, African dust was present during some of these

cases and thus may interest some data users. Examples of Tudor Hill spirals with King Air overpasses are in RFs 166, 167, 169, 170, 172, 174, 175, 178 (dates shown in Table 2). The Tudor Hill site managed by the Bermuda Institute of Ocean Sciences was used during the June 2022 deployment for extensive surface and tower measurements relevant to atmospheric chemistry as part of the Bermuda boundary Layer Experiment on the Atmospheric Chemistry of Halogens (BLEACH).



(iv)    The HU-25 Falcon experienced a significant maintenance issue at the completion of RF47 (21 February 2021), resulting in a reduced instrument payload for the remainder of the Winter 2021 deployment (RF48-61, from 4 March to 2 April 2021). The following instruments (described in sect. 4) were not allowed to operate or collect data to minimize electrical power demand: trace gases (Picarro, 2B Tech.), AMS, PILS, CVI. The 11-day gap between RF47 and RF48 (4 March 2021) was due to the adaptation of the Falcon aircraft to the new payload

strategy. To make up for most of Winter 2021 flights not having full payload capability, the Winter 2022 deployment was essentially the equivalent of two deployments, with flights starting as early as 30 November 2021 and ending 29 March 2022 (55 total flights rather than the nominal 25). No research flights occurred from 10 December 2021 to 11 January 2022 to observe the winter holiday period.

   (v)    Effort was made to keep the two aircraft as spatially coordinated as possible throughout the 162 joint flights. This

at times was challenging due to pronounced differential wind speeds (and direction) between the boundary layer (Falcon) and at the ~8-10 km altitude (King Air), and due to unforeseen delays in takeoff for the second aircraft on a given day, typically due to the airfield operations. The goal was to try to keep the aircraft within approximately 5 minutes and 6 km of each other. This goal was attained for ~73% of the dataset (Schlosser et al., in review). If one aircraft was too far ahead, often it would conduct a "delay loop (i.e., racetrack)" whereby

it would fly in a reverse track until the other aircraft caught up after which it would turn around again and fly in joint fashion. An example is shown in Fig. 3a for RF13 (1 March 2020, L1). Sometimes the trailing aircraft would turn around sooner at the "turn point" of an out-and-back flight to help reduce the spacing.

## 3 King Air measurements

Two separate King Air aircraft were used during the campaign, with nearly identical flight performance characteristics. The science payload was moved from the King Air with tail number N528NA (UC-12) to a second King Air with tail number N529NA (B200) for RF94 through RF119 to accommodate science flights during a planned maintenance period on N528NA. All other King Air research flights were flown on NASA528. Table 3 summarizes the King Air payload along with measured variables from each instrument and associated uncertainties and resolutions. Figure 4 shows a visual summary of the interior

King Air layout. Table S1 (supplementary information) summarizes performance of each instrument on both aircraft for each flight to aid data users requiring at least some minimum combination of functional instruments for their applications. Each instrument package is described in detail below.

### 3.1 Applanix navigational data

For basic navigational and aircraft motion information, an Applanix 610 system acquired 1 second data for calendar day, time, latitude, longitude, GPS altitude, ground speed, vertical speed, true heading, track/drift/pitch/roll angle.



### 3.2 High spectral resolution lidar – generation 2 (HSRL-2)

The NASA Langley High Spectral Resolution Lidar (HSRL-2) is a multiwavelength airborne HSRL providing vertically
resolved extensive and intensive aerosol properties. Extensive properties are those that depend both on aerosol particle
properties and concentration whereas intensive properties depend only on the particle properties and are independent of
concentration. Archived HSRL-2 core data include high resolution profiles of particulate backscatter and depolarization at
three wavelengths (355, 532, 1064 nm) and simultaneous and independent measurements of particulate extinction at two
wavelengths (355, 532 nm) via the HSRL technique (Hair et al., 2008; Burton et al., 2018). These profiles are used to derive
horizontally and vertically resolved curtains of extinction and backscatter Ångström exponent, lidar ratio (i.e., extinction-to-
backscatter ratio), backscatter Ångström exponents for spherical and nonspherical particles (dust, crystalline sea salt)
(Sugimoto and Lee, 2006), and aerosol type (Burton et al., 2012). The HSRL-2 backscatter and depolarization products are
reported as 10 second averages while the extinction and lidar ratio products are averaged to 60 seconds. Higher resolution
products are available from the HSRL-2 team upon request.

The aerosol backscatter product is also used to derive an aerosol mixed-layer height (MLH) (Fast et al., 2012; Scarino et al.,
2014). Mixed layer heights are based on sharp gradients in aerosol backscatter profiles that are found using a modified Haar
wavelet approach (Scarino et al., 2014). The MLH remains challenging to accurately determine in complex atmospheric
conditions, such as shallow marine boundary layers (MBLs) and multiple aerosol layers as a function of altitude. There are
multiple ways MLH can be defined and retrieved, and thus users should use discretion in how they use MLH data for their
given applications. Aerosol typing (maritime, polluted maritime, pure dust, dusty mix, smoke, fresh smoke, urban, and ice) is
based on an algorithm using depolarization, depolarization wavelength dependence, aerosol backscatter wavelength
dependence, and the aerosol lidar ratio (Burton et al., 2012).

Under ACTIVATE, additional new HSRL-2 geophysical products have been developed (or under development), including an
aerosol hygroscopic growth parameter for well-mixed MBLs, 10 m surface wind speeds, multiple cloud products, and an in-
ocean backscatter product.  A new product that is under development is the aerosol hygroscopic growth parameter f(RH),
which is produced using the HSRL-2 aerosol backscatter product and state parameters retrieved from the AVAPS dropsonde
system (sect. 3.5) in well-mixed MBLs (Ferrare et al., forthcoming). 10 m neutral stability (U10) surface wind speeds are
estimated using HSRL-2 retrievals of sea surface backscatter, i.e., the reflectance of the transmitted laser pulses from the ocean
surface (Dmitrovic et al., forthcoming). The surface backscatter, retrieved with a 1.25 m vertical resolution that corrects for
ocean subsurface scattering, is highly correlated with sea surface wave-slope variance, which is then related to wind-speed
through various empirical relationships (Cox and Munk, 1954; Hu et al., 2008). New HSRL-2 cloud retrieval products include
cloud top height, cloud top extinction, and cloud top lidar ratio at horizontal resolutions of 75 m, 150 m, and 150 m, respectively
(Hair et al., forthcoming). Relevant to ocean-air interactions such as marine biogenic emissions (Corral et al., 2022a), ocean
subsurface particulate backscatter coefficients at 532 nm are estimated at a depth of 10 m (Schulien et al., 2017) and made
available for selected flights.



Figure 5 provides a visualization of many of the aforementioned HSRL-2 data products for a representative flight (RF157 on 18 May 2022). Figure 5a shows profiles of aerosol backscatter (532 nm) for the entire flight from Bermuda to NASA LaRC in southeastern Virginia. Note the horizontal and vertical variability of aerosol particles throughout the flight. The labeled boxes indicate regions where subsets of HSRL-2 data products are shown in the corresponding small boxes below Figure 5a;

these are shown for clouds (5b), boundary layer and lower troposphere aerosols (5c), and an elevated aerosol layer (5d). These small boxes provide brief visualizations of these various data products. Blue dots in Figure 5b show (left subplot) cloud top height and (right subplot) cloud top extinction, averaged over the first optical depth, for this region. Figure 5c shows HSRL-2 products including mixed layer height (blue dots), surface wind speed (black line), aerosol type, aerosol depolarization (UV (355 nm), VIS (532 nm), IR (1064 nm)), and backscatter Ångström exponents corresponding to spherical and nonspherical

particles (dust, crystalline sea salt) in the boundary layer and lower troposphere. Figure 5d shows HSRL-2 products in the aerosol layer between 4.5-6.5 km including aerosol backscatter (UV (355 nm), VIS (532 nm), IR (1064 nm)), backscatter Ångström exponents (VIS/UV and IR/VIS), lidar ratios (UV and VIS), aerosol extinction (UV, VIS), extinction Ångström exponent (UV/VIS), and total column AOT (UV, VIS) (indicated by the blue and green lines in bottom of right figure).

**3.3 Research scanning polarimeter (RSP)**

Retrievals of aerosol, cloud, and surface reflectance properties were provided by the Research Scanning Polarimeter (RSP), which is a passive, downward-looking polarimeter, with nine spectral bands (band centers: 410, 470, 550, 670, 865, 960, 1590, 1880, and 2260 nm) that scans its 14 mrad instantaneous field of view (~100 m) along the King Air ground track (Cairns et al., 2003). Each RSP scan views the earth over an angular range of ±55° from nadir (~ 140 views) every 0.8 seconds providing

radiance and linear polarization measurements in all nine spectral bands. Each scan includes stability, dark reference, and calibration checks. A few decisions in flight planning and execution aimed to enhance RSP data quality: (i) as much as possible to keep the aircraft stable (e.g., yaw and roll); (ii) unless there was a high priority reason to fly under cirrus clouds, plan the typically joint flights for days with minimal cirrus clouds forecast above the flight track, to allow for more accurate determination of the incoming solar radiation; and (iii) fly as close as possible to the solar principal plane (i.e., azimuthally

toward or away from the Sun) based on the scientific benefits of observing sunglint and maximizing the range of scattering angles observed including in the range from 135 to 165 degrees for the polarimetric cloud bow retrievals. The public data archive contains readme files provided by the RSP team for their Level 1C and Level 2 cloud and aerosol products, including important details about biases and uncertainties that data users should consult.

Because of the scanning nature whereby the RSP views areas behind and ahead of the plane, data are re-ordered in archived

Level 1C files such that rather than being time-ordered, the data are sorted so that all the viewing angles that see the same nadir scene are put together. In cloud and cloud-free scenes, this amounts to data being aggregated to the cloud top and surface, respectively. Data from the Level 1C files are then used to develop Level 2 data files housing the aerosol and cloud data variables shown in Table 3. The RSP is ideally suited for characterizing warm cloud properties owing to the high angular density of observations per scene, with the polarized observations of the cloud bow allowing the retrieval of information about



the droplet size distribution and also the detection and characterization of drizzle (Alexandrov et al., 2012b). Spectral bands in the regions where liquid and ice absorb (1.59 and 2.26 μm, respectively) also allow the RSP to obtain bi-spectral retrievals of droplet sizes, using the same technique as applied to satellite instruments such as the Moderate Resolution Imaging Spectroradiometer (MODIS) and the Visible Infrared Imaging Radiometer Suite (VIIRS). The primary cloud properties retrieved include cloud flag/test, cloud top altitude, cloud top phase index, cloud optical thickness, and cloud droplet size

distribution (i.e., effective radius and variance). The cloud flag/test indicates whether a cloud was detected underneath the aircraft. A multi-angle parallax approach is used to estimate cloud top heights (Sinclair et al., 2017). The cloud top phase index variable indicates whether there is liquid at cloud top (Van Diedenhoven et al., 2012). Multi-angle polarimetry is used to retrieve effective radius and variance of the drop size distribution at cloud top for both liquid and mixed-phase clouds (Alexandrov et al., 2012b; Alexandrov et al., 2012a) and, for observations close to the solar principal plane, the drop size

distribution itself (Alexandrov et al., 2012b; Alexandrov et al., 2012a). These multi-angle polarimetric retrievals have been validated against in situ observations (Adebiyi et al., 2020; Alexandrov et al., 2018) and found to be much more robust against artifacts than bi-spectral retrievals (Fu et al., 2022). Bi-spectral retrievals were also conducted for effective radius and cloud optical thickness (Nakajima and King, 1990). Column water vapor amount is provided above either the surface (cloud-free scenes) or cloud top (cloud scenes) (Sinclair et al., 2019).

Level 2 aerosol products (Stamnes et al., 2018; Schlosser et al., 2022) for both the fine and coarse mode include aerosol optical depth, aerosol size distribution parameters (effective radius/variance and number concentration), single scattering albedo (SSA), and complex refractive index, and also ocean properties (ocean diffuse attenuation coefficient, ocean hemispherical backscatter coefficient, chlorophyll-a concentration, surface wind speed) are reported in these files based on a model for open ocean waters (Chowdhary et al., 2006). An aerosol layer height is also retrieved from the RSP observations (e.g., Wu et al.,

2016), but we note that the HSRL-2 sensor provides far greater detail regarding the vertical distribution of aerosol particles.

### 3.4 Joint HSRL-2 and RSP retrieval products

Vertically-resolved $N_a$ is derived for the first time using the vertically-resolved extinction backscatter coefficient [1/m] measured by HSRL-2 at 532 nm, combined with the column-averaged aerosol extinction cross-section for the fine-mode

aerosol retrieved by RSP at 532 nm. The details of this combined lidar-polarimeter algorithm and comparisons against in-situ $N_a$ are provided in Schlosser et al. (2022). Forthcoming work will summarize additional joint retrieval products that will be archived for public use once they are developed, including retrievals of $N_d$.

### 3.5 Dropsondes

The National Center for Atmospheric Research (NCAR) Airborne Vertical Atmospheric Profiling System (AVAPS) was deployed on the King Air to release dropsondes to obtain vertical distributions of pressure, wind (u, v, w components), static air and dew point temperature, and relative humidity (RH). Note, the horizontal wind components are measured directly, while the vertical wind is estimated using the dropsonde fall velocity. Manual releases were done using a dropsonde launch tube



relying on NCAR NRD41 mini sondes, which have been summarized elsewhere and used in recent airborne campaigns such
as the Organization of Tropical East Pacific Convection (OTREC) (Vömel et al., 2021) and the in-progress Investigation of
Microphysics and Precipitation for Atlantic Coast-Threatening Snowstorms (IMPACTS). An extensive summary of the
AVAPS system performance and quality control procedures during ACTIVATE is provided by Vömel et al. (in review).

Table 1 summarizes the number of dropsondes released per deployment, with a total of 785 providing full profiles of all
variables with good parachute performance. Table 2 additionally shows the number of such full profiles per flight. The
dropsondes provided vertical profiles between approximately the surface and ~9 km, which was the typical flight level of the
King Air. However, releases were sometimes as low as ~5.2 km. Usually between 2-4 dropsondes were used per statistical
survey flight with spatial separation such that each one gave a representative view of the atmospheric column in different
portions of the flight. Process study flights involved more dropsondes (up to 23 in RF173 on 11 June 2022) to do more detailed
characterization warranted for model intercomparison studies such as for cold air outbreaks (Chen et al., 2022; Li et al., 2022;
Seethala et al., 2021) and summertime cumulus cloud systems (Li et al., in preparation).

### 3.6 Airborne camera images

Airborne camera images are useful for a variety of data analysis applications, and were collected by a nadir-facing camera
mounted beneath the airplane and forward-facing camera mounted in the aircraft cockpit. One important application is the
development of cloud masks to identify the presence of clouds above and below the aircraft, as detailed in sect. 5.4, which has
been demonstrated already for the nadir camera on the King Air (Nied et al., 2023). Table 4 summarizes the camera details on
the King Air, with different types of cameras used in nadir (Garmin VIRB Ultra 30 for RF1-RF61; AXIS F-1005-E for RF62
and onwards) and forward (GoPro for RF1-RF40; AXIS F-1005-E for RF41 and onwards) configuration throughout
ACTIVATE. Photos taken with these cameras were stitched with UTC time stamps and archived as mp4 videos. Playback can
be sped up on most MP4 viewers for faster viewing.

### 4 HU-25 Falcon measurements

Table 5 summarizes the instrument payload on the HU-25 Falcon with Table S1 summarizing instrument performance for each
flight. Figure 6 shows visually the exterior probes and the interior layout of the Falcon. As noted earlier, a subset of instruments
were not operated in the Winter 2021 deployment (RF48-61 from 4 March to 2 April 2021) to accommodate a power issue on
the Falcon. Those instruments were deemed to be the lowest priority in terms of satisfying the three baseline ACTIVATE
objectives summarized in sect. 2.1.

### 4.1 Applanix navigational data

Similar to the King Air, basic navigational and aircraft motion data (calendar day, time, latitude, longitude, GPS altitude,
ground speed, vertical speed, true heading, track/drift/pitch/roll angle) were obtained with an Applanix 610 system with the



exception that data were obtained natively at 20 Hz resolution and then averaged to 1 Hz resolution for archival. Data at 20 Hz resolution are available upon request. Similar to the King Air, Applanix data were recorded internally and on the real time data system and post-processed to obtain increased accuracy and precision via Applanix's proprietary software.


### 4.2 Diode Laser Hygrometer and trace gases

Three different instruments were used to measure trace gases including water vapor ($H_2O(v)$), $CO_2$, $CH_4$, $CO$, and $O_3$. The Diode Laser Hygrometer (DLH) is an open path, near infrared absorption spectrometer (Diskin et al., 2002) with its optical path entirely outside the Falcon cabin between a window in the cabin and a retroreflector affixed to the instrumentation pylon

on the starboard wing. The round-trip beam path was on the order of 8 m with a vertical extent of ~1.5 m and a longitudinal extent of ~2 m, which, coupled with the optical data acquisition rate, define the limit on the temporal/spatial resolution of the measurement. DLH reported water vapor through 1 Hz and 20 Hz data products, but data are available upon request as fast as 60 Hz depending on airspeed. DLH data are available in clouds, but there was occasional data loss in very dense clouds due to a backscatter artifact. There was also occasional data loss caused by ice formation on the retroreflector, which prevented

sufficient optical power from reaching the detector to make a measurement. These data were detected and removed, which reduces the water vapor data available within clouds and during/following icing. In addition to the primary DLH data product, water vapor mixing ratio, DLH water vapor data are converted to relative humidity with respect to both liquid water and ice using the on-board in situ measurements of ambient pressure and temperature described in sect. 4.3.

The other two instruments were located entirely within the cabin in a trace gas rack and were extractive, sampling from

fuselage-mounted inlets to measure concentrations internally. A PICARRO G2401-m measured $CO_2$, $CH_4$, and $CO$ at 0.4 Hz resolution (Digangi et al., 2021) using a modified Rosemount total air temperature probe gas inlet (Buck Research Instruments, LLC) mounted on the crown collocated with the aerosol inlets (Fig. 6a). These measurements were calibrated hourly during flight with a 1-minute single point calibration and weekly during deployments on the ground with a three-point calibration, with all standards traceable to WMO X2019 ($CO_2$), WMO X2004A ($CH_4$), and WMO X2014A ($CO$) scales. Some data from

the PICARRO were omitted due to inlet leaks predominantly at high altitude (i.e., RF1-9 on 14-27 February 2020). $O_3$ was measured at 0.5 Hz by a 2B Technologies Inc. $O_3$ monitor (Model 205) using a forward-facing J-probe inlet mounted on the Falcon nadir panel and relied on a custom sampling apparatus to enhance data quality at high altitude (Wei et al., 2021). $O_3$ data were zeroed for 1 minute with a KI filter hourly in-flight to account for baseline drifts to ensure high data quality, and the monitor was calibrated before and after each deployment with a NIST-traceable standard (Model 305, 2B Technologies Inc.).

The $O_3$ data are vulnerable to altitude/pressure dependence that is accounted for based on these routine calibrations, but it is cautioned that there could be residual effects about which interested data users can consult the instrument team.

The trace gas mixing ratios can be used in conjunction with back-trajectory analysis to link air masses to source regions and can also be used in studies of wet scavenging and aqueous production as both $CO$ and $CH_4$ can be considered conserved tracer species. For example, $CO$ and $CH_4$ are well correlated with a similar relative enhancement ratio for much of the ACTIVATE





dataset, consistent with the hypothesis that the observed air was influenced by urban emissions with relative pollutant levels
       dependent on the degree of dilution. However, there were occasionally periods where the enhancement factor differed, with
       CO enhancements much greater than $CH_4$ in relation to the typical enhancement ratios during the campaign. This is consistent
       with less efficient forms of combustion, such as biomass burning, with incidences of this observed briefly during several flights
       when near the coast and for longer segments offshore during two flights, RF28 (26 August 2020) and RF38 (23 September

2020). Enhancement ratios of $O_3$ and CO also can be used effectively to infer chemical information about the airmass. One
       example is early during the Winter 2022 deployment (Jan-Feb) when $O_3$ and CO were inversely correlated, consistent with
       $NO_x$ titration of $O_3$ in a VOC-limited chemistry regime. As the flights moved farther toward spring, this correlation became
       weaker (March), then reversed to become a roughly positive correlation between the species (May/June). This is consistent
       with the switch to a $NO_x$-limited regime of $O_3$ photochemistry as VOC emissions increase with the warmer temperatures and

the growth of MBL heights further diluting the anthropogenic $NO_x$ emissions; this highlights another unique advantage of the
       routine, long-duration measurements of the ACTIVATE dataset.

## 4.3 Fast-response three dimensional winds and state parameters

       High resolution in situ measurements of three dimensional winds (u, v, w components), temperature and pressure were obtained

using the Turbulent Air Motion Measurement System (TAMMS) (Thornhill et al., 2003). The system has been installed on the
       NASA P-3 for over 20 years. This is the first time it was integrated onto the NASA HU-25. The raw data were recorded
       between 100 and 200 Hz with a UEIPAC-300 real time controller (United Electronics Industries, Inc.) and then averaged down
       to 20 Hz for archiving and analysis work. Five flush-mounted ports (0.417 cm diameter) were positioned in a cruciform pattern
       on the nose of the HU-25 in order to not have any interference in the airflow around the aircraft. The angle of attack was

derived from the vertically positioned ports whereas the slideslip angle was obtained from the horizontally aligned ports. The
       center tap was a backup for the dynamic (impact) pressure measurement. High time resolution and high precision pressure
       transducers (Honeywell PPT-2 and Rosemount) were placed as close as possible to the pressure ports to minimize time delays.

       Whereas the five-port pressure system helps determine the speed of the air relative to the aircraft, the speed of the aircraft
       relative to the earth was obtained with inertial/GPS data measured via the Applanix 610. Aircraft velocity components are a

blended solution using the inertial and GPS data via a Kalman filtering technique (e.g., Brunke et al., 2022). The u and v
       components are zonal and meridional, respectively, while w is the vertical wind speed (positive is upwards). The three
       dimensional winds are computed using the full version of the well-established air motion equations (Lenschow, 1986).

       The total air temperature, from which the ambient air temperature and true airspeed were calculated, was measured by the non-
       deiced version of the Rosemount Model 102 total air temperature sensor with a fast-response sensing element (E102E4AL, >

5 Hz response). The pressures (total, static, and impact (dynamic)) were obtained with a Rosemount pressure transducer and a
       Rosemount Micro Air Data Transducer (model 2014MA1A) that was tied into the co-pilot's pressure port to minimize the
       pressure defect. An ancillary measurements of the infrared (IR) surface temperature was also included in the TAMMS



instrument suite of measurements. IR surface temperature was obtained from a downlooking Heitronics KT-15 Infrared Thermometer.

Multiple dedicated calibration flights during each deployment year were performed in order to establish the primary calibration coefficients necessary to ensure the highest data quality. Calibrations were done at different altitudes above the boundary layer in clean homogenous air masses to determine:

- Angle of attack slope and offset – via speed variations

- Sideslip slope – via crabbing the HU-25 with wings level

- Pressure defect – via along wind reverse headings

- Heading offset (sideslip offset) – via cross wind reverse headings

These calibration results were then applied to the final data along with any time lag adjustments (Brunke et al., 2022). The Applanix data were also post-processed to reduce the velocity and position errors. The error in positioning for the final data was reduced to less than 1 m. The calibration data were repeatable from year to year and allowed for a final and consistent set 460 of calibration coefficients to be utilized for all the variables except for the heading offset. That value changed between deployments due to the removal and re-installation of the Applanix on the HU-25.

There are several caveats that a potential user should be aware of prior to using these data. For the three-dimensional winds, users should nominally restrict use to times when the HU-25 is flying straight and level as significant changes in pitch, roll, and altitude can introduce artifacts and noise into the winds calculation. If non-straight/level times are needed for analysis, 465 users are advised to consult with the TAMMS instrument team and at the very least look at the data in great detail to look for correlations with pitch or roll that are adversely influencing the derived winds. In addition, care should be taken when averaging the horizontal winds as the averaging should be done to the u and v components and then the wind speed and direction should be recomputed post averaging. When looking at fine scale details such as turbulent fluxes via eddy correlation or the average updraft velocity under clouds, users are advised to consider using time windows that overlap by 50% in order to increase 470 statistics. The time window length should be sufficiently long to capture all the eddy sizes that contribute to the turbulent fluxes. Assuming the typical ACTIVATE leg length of 3 minutes and an average airspeed of 100 m s$^{-1}$, a segment of 512 samples can resolve eddy sizes of up to 1.28 km and if not overlapped then 7 full segments can be averaged together to compute the average turbulent fluxes. If the suggested overlap of 50% is used then 13 full segments can be averaged together to increase statistics significantly.


**4.4 Aerosol characterization**

In situ measurements of aerosol properties were conducted with the Langley Aerosol Research Group Experiment (LARGE) instrument package used in previous NASA campaigns such as Studies of Emissions and Atmospheric Composition, Clouds and Climate Coupling by Regional Surveys (SEAC$^4$RS) (Toon et al., 2016) and the Cloud, Aerosol and Monsoon Processes



Philippines Experiment (CAMP²Ex) (Reid et al., 2023). The majority of aerosol measurements were conducted with instruments integrated inside the fuselage and air provided by two manually-switched inlets mounted on the Falcon's exterior crown (top of Fig. 6a). An isokinetic Clarke-style shrouded solid double diffuser inlet (Brechtel Manufacturing Inc. [BMI]) was relied on during cloud-free scenes for aerosol characterization (Mcnaughton et al., 2007) whereas a counterflow virtual impactor (CVI; BMI) was used while in clouds (Shingler et al., 2012) for measurements of droplet residual particles (i.e.,

particles remaining after droplet evaporation). An inlet flag data product is archived indicating which inlet (i.e., the CVI or the isokinetic inlet) was used at a given time for the HR-ToF-AMS and LAS instruments (described below), whereas all other LARGE instruments summarized in this section only sampled downstream of the isokinetic inlet. Those instruments that are not switched to the CVI require in-cloud filtering to remove periods potentially biased by droplet shattering artifacts (discussed in sect. 4.4.5). The upper-size limit for all bulk observations (unless otherwise noted below) is governed by the isokinetic inlet

performance (Mcnaughton et al., 2007) with a nominal cutoff point at 5 μm diameter (Table 5); note though that this cutoff diameter is for ambient RH conditions while the final in situ aerosol measurements will be more representative of dried (and thus smaller particles) conditions owing to heating during inlet transmission. All LARGE measurements are archived at 1 Hz time resolution (unless otherwise noted) and at standard temperature and pressure (STP; 273.15 K and 1013.25 mb). The LARGE measurements can be categorized into optical, microphysical, and chemical, which are described in order next.


### 4.4.1 Optical

Dry scattering and absorption coefficients were measured at three wavelengths using a nephelometer (TSI Inc. Model 3563; 450, 550, 700 nm) (Ziemba et al., 2013) and a particle soot absorption photometer (PSAP; Radiance Research; 470, 532, 660 nm) (Mason et al., 2018), respectively. Scattering coefficient measurements have been corrected for angular truncation

(Anderson and Ogren, 1998) and absorption coefficients were corrected using (Virkkula, 2010) A measurement of aerosol hygroscopic growth factor, f(RH), was calculated in the form of the ratio of total light scattering at high and low RH. Scattering measurements were made by two independent nephelometers in parallel; one at low RH (i.e., generally less than 40%) and one at high RH (controlled targeting 85%) using a custom Nafion humidifier (Ziemba et al., 2013). These measurements allow calculation of the hygroscopicity gamma parameter, which is then used with the dry scattering coefficient to calculate scattering

at any RH up to saturation. The f(RH) data archived are calculated specifically between 20% and 80% RH. f(RH) is only reported for conditions when 550 nm scattering coefficients (at both high and low RH) exceeded 5.0 Mm⁻¹ and controlled RH was between 72% and 92%.

A 1 μm cyclone was utilized upstream of both nephelometers for 2021-2022 flights and thus the scattering coefficients and f(RH) represent submicrometer aerosol in contrast to PSAP data, which represent bulk aerosol; the nephelometer data in 2020

correspond to an upper cutoff point of 5 μm. The scattering and absorption coefficient data are used to compute secondary properties including scattering and absorption Angstrom exponents and single scattering albedo (SSA) discussed in sect. 4.4.4.

### 4.4.2 Microphysical



Total $N_a$ was measured with two independent condensation particle counters (CPCs). One CPC was sensitive to all particles with diameter greater than 3 nm (TSI Inc. Model 3776) and the other only to particles with diameter greater than 10 nm (TSI Inc. Model 3772). The difference in number concentration between the two CPCs is informative about ultrafine, and presumably newly formed, particles between 3 and 10 nm for data users interested in research into particle nucleation (Corral et al., 2022b). Non-volatile particle concentrations (for particles with diameter greater than 10 nm) were recorded by an additional Model 3772 CPC that was coupled to a 350° C thermodenuder. The CPC concentrations are useful for assessing the evolution of the full aerosol population, for understanding particles sources and formation processes, and to provide "closure" checks on the integrated size distribution data.

Dry aerosol size distributions are measured by different instruments for varying diameter windows. The ultrafine/Aitken-mode window between 3-100 nm diameter is measured with a scanning mobility particle sizer (SMPS; Model 3085 DMA, Model 3776 CPC, and Model 3088 Neutralizer; TSI Inc.), which classifies particles based on their electrical mobility diameters. The accumulation-mode diameter window extending from 100 to 5000 nm is captured based on optical diameters using a laser aerosol spectrometer (LAS, TSI Inc. Model 3340) (Froyd et al., 2019). The LAS was calibrated using mono-disperse ammonium sulfate particles (i.e., with a refractive index of 1.52) to optimize relevance to ambient aerosol particles (Shingler et al., 2016), and both sizing instruments were spot-checked frequently to ensure long-term stability using NIST-traceable polystyrene latex spheres at appropriate sizes. Independent empirical size-dependent corrections have been applied to both the SMPS and LAS datasets that allow "stitching" the distributions at 100 nm; excellent closure is demonstrated for most ambient conditions by adding integrated SMPS and LAS number concentrations compared to total CPC concentrations. A demonstration of this is provided in Fig. 7 for RF12 on 29 February 2020. While the LAS provides 1 Hz data, the SMPS data are at lower time resolution (~45 s) and require caution to interpret when concentrations are rapidly changing in-flight. Droplet residual LAS particle size distributions are archived (using the inlet flag) during CVI in-cloud sampling periods. Interpretation of these data has not been demonstrated previously but should provide supplementary information to compositional analysis towards improving our understanding of cloud processing. The LAS-CVI data require the use of the InletFlag (0 = isokinetic; 1 = CVI) for separation of the two categories of data.

Cloud condensation nuclei (CCN) concentrations and spectra for submicrometer particles were measured with a CCN spectrometer (Droplet Measurement Technologies [DMT] Inc.) using both constant and scanning flow techniques (Moore and Nenes, 2009).

### 4.4.3 Chemical

Non-refractory mass concentrations of sulfate, nitrate, ammonium, chloride, organics, and numerous mass spectral markers (mass-to-charge ratio [m/z] 42, 43, 44, 55, 57, 58, 60, 79, 91) were measured by a High Resolution Time of Flight Aerosol Mass Spectrometer (HR-ToF-AMS; Aerodyne) (Decarlo et al., 2008). The nominal vacuum aerodynamic diameter window of the AMS was 60 to 600 nm. As summarized for ACTIVATE already (Dadashazar et al., 2022a), the 1 Hz fast-MS mode AMS data were averaged to 30 s time resolution for the data archive. A brief overview of what types of species the aforementioned





m/z mass spectral markers represent is as follows: 42 (amines, $C_2H_4N^+$), 43 (mixed hydrocarbons, $C_3H_7^+$ or $C_2H_3O^+$), 44 (oxidized hydrocarbons, $CO_2^+$), 55 (aliphatic hydrocarbons, $C_4H_7^+$), 57 (aliphatic hydrocarbons, $C_4H_9^+$), 58 (sea salt/marine,

$NaCl^+$), 60 (biomass burning, $C_2H_4O_2^+$), 79 (methanesulfonate/marine, $CH_3SO_2^+$), 91 (aromatic hydrocarbons, $C_7H_7$). The AMS is operated using a custom pressure-controlled inlet (at 500 torr) and all mass concentrations are reported at STP. The overall AMS ionization efficiency was calibrated using mono-disperse 400 nm ammonium nitrate particles throughout the 3-year measurement period, and a collection efficiency value of unity was applied to all data based on comparison to simultaneously measured PILS-based sulfate mass concentrations. AMS-CVI data are reported in separate files as compared

to other AMS data from cloud-free air sampling. The AMS-CVI data include only relative mass fractions. The CVI was extensively characterized previously by Shingler et al. (2012), with a demonstration of the utility of AMS-CVI data during ACTIVATE provided by Dadashazar et al. (2022a).

Water-soluble ionic composition was measured by a particle-into-liquid sampler (PILS; BMI) coupled to offline ion

chromatography (Sorooshian et al., 2006; Crosbie et al., 2020). The time resolution varied between 5 and 7 minutes depending on the deployment. The PILS data represent bulk aerosol between approximately 50 and 5000 nm, including the following ions: anions = chloride, nitrite, bromide, nitrate, sulfate, oxalate; cations = sodium, ammonium, dimethylamine, potassium, magnesium, calcium. Details of the ion chromatography instrument and analysis methods for anion and cation speciation are provided in recent ACTIVATE studies (Corral et al., 2022a; Gonzalez et al., 2022). The PILS was operated without denuders

and thus users should account for this aspect of the data when interpreting concentrations for semi-volatile species such as ammonium for which there may be positive biases due to gas-phase contributions.

### 4.4.4 Secondary aerosol products

The archived "optical" and "microphysical" files are useful starting points for data users interested in summary statistics and

special calculated parameters. For example, the "optical" files include data for submicrometer dry scattering (450, 550, 700 nm) and calculated extinction (532 nm) coefficients, total aerosol absorption coefficient (470, 532, 660 nm), f(RH) and its associated gamma parameter at 550 nm, aerosol scattering (450/700 nm) and absorption (470/660 nm) Angstrom Exponents, and SSA (at 450, 550, 700 nm). Note that the submicrometer designation applies to 2021-2022 flights and that 2020 flights correspond to bulk aerosol (< 5 μm). The extinction parameter was calculated by summing submicrometer scattering and bulk

absorption, with scattering data at 550 nm adjusted to 532 nm using the measured Angstrom Exponent. Note that the gamma parameter allows one to estimate scattering at any RH (Ziemba et al., 2013); scattering coefficient, extinction coefficient, scattering Angstrom Exponent, and SSA are all provided in archived files at ambient RH.  The "microphysical" files provide the CPC concentrations along with sub- and supermicrometer number, surface area, and volume concentrations from the LAS with the assumption of spherical particles. During data processing, additional filters are applied to the 1 Hz data such as

thresholding and smoothing to obtain secondary products such as SSA, which can introduce gaps that do not exist in the raw



data. Caution should be taken when averaging ratio-based values such as SSA as this can introduce unrealistic values in the data.

### 4.4.5 Data usage notes

Additional notes on data usage are provided here with the reminder that data users should always also consult with International Consortium for Atmospheric Research on Transport and Transformation (ICARTT) data file headers (files described more in sect. 7) for guidance on data usage. Mass loadings and concentrations are all reported at standard temperature and pressure. Conversion factors at 1 Hz resolution are provided in the ICARTT data files for data users interested to convert the data back to ambient temperature and pressure conditions. The latter step is important for users aiming to compare in situ data to remote

sensing data because the remote sensors retrieve information at ambient conditions.

Aerosol measurements are vulnerable to contamination due to cloud droplet shatter on the sampling inlet when aircraft fly in clouds or precipitation below a cloud; this usually is manifested in unrealistically high particle number concentrations often with high-frequency variability as measured by either of the CPCs. It is recommended that data users use strict criteria to filter aerosol data for cloud-free conditions, with a recent ACTIVATE study encouraging criterion of cloud liquid water content

(LWC, recommended to be provided by the FCDP) being less than 0.001 g m$^{-3}$ (Schlosser et al., 2022). However, users concerned about more confidently separating cloud hydrometeors from coarse aerosol should consult with instrument teams operating the probes described in sect. 4.5 and/or develop the types of analyses (e.g., joint histograms) that compare different variables like LWC and $N_d$ to see more clearly where clusters emerge for coarse aerosol and how to better separate them from cloud droplets (see Fig. 2 of Schlosser et al., 2022).

Since it is a differencing technique, the AMS can produce negative mass concentrations in clean conditions which should be retained in statistical calculations whenever possible. Removal of such points during a level leg for instance can positively bias the resulting value for the leg averaged value.

Owing to the relatively long time resolution of the PILS (5-7 min) and the 'smearing' of data without step function responses in composition (Crosbie et al., 2020), data users should use caution with how the data are used for their applications. More

specifically, PILS data are unreliable for vertically-resolved depictions of ionic composition due to the short amount of time spent during most level legs during ACTIVATE (~3 min) and the fact that spiral and slant profiles were usually shorter than the time needed to collect a PILS sample. In contrast, the data are well-suited for statistical assessments of concentrations and chemical ratios relying on many flights of data as demonstrated by Hilario et al. (2021).

**4.5 Wing-mounted probes (aerosol and cloud droplet size distributions)**

Four optical probes were used to characterize aerosol and cloud droplet size distributions extending from 0.5 to 1465 μm. All such data are reported at ambient conditions (temperature, pressure, RH), which requires caution when trying to compare these aerosol data to dry aerosol measurements described in sect. 4.4. A DMT Cloud Droplet Probe (CDP; 2-50 μm) was mounted on the crown of the aircraft fuselage, while a Cloud and Aerosol Spectrometer probe (CAS; 0.5-50 μm) was mounted on the



starboard wing (Fig. 6a). Both instruments measure the scattered light pulses as coarse mode aerosol particles and cloud droplets pass through a laser beam, where the count rate and light intensity are related to the particle number and size, respectively. Particle concentration is computed by multiplying the measured count rate by a sample volume that is the product of the probe sample area and the aircraft true airspeed (TAS). The CDP sample area was experimentally measured by DMT to be 0.323 mm$^2$, while an assumed sample area for the CAS of 0.25 mm$^2$ was used. In addition, cloud liquid water content

(LWC), effective variance, and effective radius were calculated assuming spherical particles with unit density. The CAS is able to measure particles between 0.5-2 μm but its shrouded inlet may make the instrument susceptible to in-cloud droplet shatter, unlike the open path CDP. The CAS data are archived at 1 Hz, while the CDP data are archived at >= 1 Hz depending on the deployment. For 2020, it was observed that 1 Hz data made it hard to distinguish cloud centers and edges, so the data sampling rate was increased for subsequent years of flights.

On the port side wing (Fig. 6a) was a Fast Cloud Droplet Probe (FCDP; 3-50 μm) and a Two-Dimensional Stereo (2D-S; 29-1465 μm), both of which are manufactured by SPEC Inc. The FCDP is a forward scattering probe with a rapid sampling rate of 25 ns to enable single particle detection for all particles. Its fast electronics and other features like a small pinhole for coincidence reduction imply lower uncertainties in particle sizing and counting (Baumgardner et al., 2017; Kirschler et al., 2022; Kleine et al., 2018; Knop et al., 2021; Voigt et al., 2021). Archived FCDP data include aerosol and droplet number size

distributions, LWC, effective diameter, and median volume diameter. Extensive processing and corrections to the FCDP data are described in Kirschler et al. (2022). Meanwhile, the CDP and CAS data have not been similarly corrected to date, which may introduce biases particularly for high cloud droplet number environments exceeding 500 cm$^{-3}$ (Lance, 2012).

The Two-Dimensional Stereo (2D-S) optical array probe from SPEC Inc. relies on 128 photodiodes to produce shadow images of single particles (Lawson and Baker, 2006; Lawson et al., 2019). Archived 2D-S data include cloud number size distributions

for liquid/ice/total, liquid and ice water content, ice flag, effective diameter for liquid/ice/total, median volume diameter for liquid and total. 2D-S images are provided on request, which can be illustrative of hydrometeor shapes (liquid droplets versus ice) and coarse aerosol types such as bioaerosols. The probe has two identical arms that are perpendicular with 785 nm wavelength lasers associated with each to generate a diffraction pattern for traversing particles. The recorded ensemble of 'slices' obtained rapidly by triggered photodiodes help generated 2D images of particles (Knollenberg, 1970). The 2D-S used

on the Falcon is described in detail by Kirschler et al. (2022), who note that with the fast response time of 41 ns, the 2D-S has less uncertainty for characterizing spheroids,  and is in the middle of the range for ice particles, compared to other optical array probes (Baker and Lawson, 2006; Gurganus and Lawson, 2018; Lawson and Baker, 2006; Bansmer et al., 2018). For data users interested in stitching together 2D-S size distributions with the other probes like FCDP, the method discussed by Kirschler et al. (2022) is a suitable option to confront the overlap of the two probes between 16-51.3 μm. They did an overlap

calculation for the diameter space between the lower FCDP bin bound at 27 μm and the higher 2D-S bin bound at 39.9 μm. Linear interpolation can be applied using the next 2D-S bin and proportionality between the last FCDP bin and the new 2D-S bin. Examples of FCDP and 2D-S data products are shown in sect. 6 for a representative case flight.



In terms of data usage notes, a few factors should be considered by users:

- Consider that the scattered light spectrometers in use are designed for cloud measurements and uncertainties increase in the case of aerosol measurements. For instance, sizing for these probes is calibrated assuming water's refractive index and so if there is coarse mode dust, biological particles, and/or sea salt there will be sizing biases due to the varying refractive indices for these aerosol types relative to water.

- The use of the 2D-S horizontal arm is preferable, as the vertical arm did not operate properly in all flights and was disabled in those cases. The data locations are marked accordingly in the vertical arm.

- If the size distributions of the FCDP and 2D-S shall be combined, it is recommended not to make the transition above 30 μm, because the measurement area difference of the instruments increases quadratically with size and causes a non-negligible statistical difference, which can manifest itself in unfilled size bins.

- Precipitation particles occur in a considerably lower number than ordinary cloud droplets and accordingly the statistics in abundance are lower for the 2D-S in this case, which is reflected in an increased measurement uncertainty. This should be accounted for when comparing in-situ precipitation measurements with remote sensing platforms and models.

## 4.6 Cloud water composition

A special aspect of ACTIVATE was the focus on cloud water measurements due to the extensive amount of time the Falcon spent in clouds. Cloud water samples were also collected using the Axial Cyclone Cloud water Collector (AC3), which was characterized and described in detail by Crosbie et al. (2018). The AC3 was mounted on the Falcon's exterior crown close to the CVI (top of Fig. 6a). The AC3 extracted cloud water from the air stream when the aircraft was in cloud. A shutter was used at the inlet of the AC3 when the Falcon was out of cloud to reduce contamination. Cloud water was collected by vacuum through a Teflon sampling line inside the Falcon and deposited in 15 mL HDPE centrifuge tubes. Samples were stored in a refrigerator post-flight and then analyzed subsequently with ion chromatography (IC), a pH meter, and inductively coupled plasma mass spectrometry (ICP-MS). Owing to varying liquid volume in each sample vial, the top priority was IC analysis, followed by ICP-MS, and finally pH. The variable volume was due to different periods of time the aircraft was in cloud per vial, varying amounts of cloud LWC during sample collection, and other AC3 performance factors (Crosbie et al., 2018). For context, 70% (90%) of the 535 total vials were collected within 6 minutes (13 minutes).

The details of the three analytical methods used at the University of Arizona and quality control details such as collection of sample blanks are described elsewhere for interested readers (Corral et al., 2022a; Gonzalez et al., 2022; Stahl et al., 2021). The IC was able to speciate and quantify the following ions in order of elution: anions = glycolate, acetate, formate, methanesulfonate, pyruvate, glyoxylate, chloride, nitrite, bromide, nitrate, glutarate, adipate, succinate, maleate, sulfate, oxalate, phthalate; cations = sodium, ammonium, dimethylamine, potassium, magnesium, calcium. ICP-MS elements detected

include: Li, Be, B, Na, Mg, Al, S, Cl, K, Ca, Ti, V, Cr, Mn, Fe, Co, Ni, Cu, Zn, Ge, As, Br, Rb, Sr, Y, Zr, Nb, Mo, Ru, Rh, Pd, Ag, Cd, Sn, Sb, Te, I, Cs, Ba, Ce, Hf, Ta, W, Re, Os, Ir, Pt, Au, Hg, Tl, Pb, Th, U.

Cloud water species concentrations from the IC and ICP-MS are reported in aqueous units (mg L$^{-1}$), and for conversion to air equivalent units (μg m$^{-3}$) data users can apply their own specific criteria. For context, past ACTIVATE studies have conducted the conversion with knowledge of cloud LWC as derived from the FCDP by using the average LWC during periods of sample

collection when LWC exceeded a threshold of 0.02 g m$^{-3}$ (Corral et al., 2022a; Gonzalez et al., 2022). Aqueous concentrations can be multiplied by the aforementioned mean LWC value during sample collection divided by the density of water. In environments dominated by broken and more vertically developed cumulus clouds, cloud water in edges or tenuous clouds is ineffectively captured. To combat this, Crosbie et al. (2022) used a threshold of 0.1 g m$^{-3}$ and provide a sensitivity analysis for combining cloud water with microphysical data.


### 4.7 Forward camera imagery

Depending on the application of Falcon data, forward camera imagery can be critical to visually determine the conditions the aircraft was flying through at a given time. Camera details were already discussed in sect. 3.6 and summarized in Table 4. All videos start based on the takeoff times listed in Table 2 and continue until the landing time.  However, a significant number of

the files end before landing (sometimes up to 15 minutes) due to the fact that the last file did not close properly once the power was turned off. The files were recorded at 2 second resolution for 2020 and 1 second for 2021 and 2022.

### 4.8 Merge files

Specific to the Falcon aircraft are "merge files" on the publicly available data archive (sect. 7) that are created at different time

resolutions of interest (e.g., 1, 5, 10, 15, 30, 60 s, or matching an individual data product start and stop times). The aim of these files is to accommodate data analysis efforts by synthesizing different time resolutions among instruments in the aircraft payload as well as sampling location. An online merge tool puts different in situ datasets on a common time base using weighted time averages of each dataset. The final archived time base can either be a time series with constant interval between points or based on an individual dataset's time stamps. The merge tool accounts for data points that have missing or limit of

detection data codes by skipping over them to not bias the resultant values. The merge files have been converted into netCDF file format (.nc) at 1 s and 60 s time resolutions for 2020 (2021 and 2022 forthcoming) to be more conducive to modelling and analysis applications by providing more machine-actionable metadata as well as metadata provided by individual instrument teams.

## 5 Complementary data products

### 5.1 Flight reports





Each individual flight has an archived flight report drafted and reviewed by flight scientists and pilots, which can serve as a useful resource for data users aiming to learn more about special features in a particular flight. A caveat is that these reports incorporate notes from scientists and pilots during flight without any post-flight data analysis to provide extra evidence for
certain documented features such as sources of dust or biomass burning. It is recommended to consult these files and the "special notes" column of Table 2 to see if relevant details are provided fitting a particular interest for a data user such as instances of mixed-phase clouds, satellite underflights, or air mass types of interest like dust or biomass burning.

## 5.2 Falcon flight leg index files

The repeated nature of stairstepping legs flown by the Falcon motivated the need for a way to identify leg types as a function of time. This can aid in analysis of data across multiple flights focused on statistics as a function of leg type. To address this, an individual file was generated per flight day (i.e., a single file contains two flights for double flight days) that the Falcon flew identifying 14 different leg types with start and stop times per leg in flight. Ten digit indices are provided describing the deployment number, flight number, flight type (process study versus statistical survey), leg type, ensemble number, and
ensemble type (cloud-free or cloud). The 14 leg types identified include: takeoff and landing, transit leg (usually after takeoff and before landing), ACB, BCB, BCT, ACT, MinAlt, Ascent, Descent, Slant/Spiral (i.e., dedicated soundings covering a significant vertical distance beyond what ascents and descents cover during typical stairstepping), BBL, ABL, RS, Other (any other leg not defined otherwise). It is important to note that leg types are assigned based on the intention of the leg as determined by the flight scientist and not a description of the data that was collected during that period. For example, an ACB leg could
have been flown in a region of scattered cloud above the nominal bases yet resulted in no cloud penetrations. Also, process study flights with numerous legs at different levels in cloud may have the legs between ACB and BCT called "Other" (e.g., RF173 on 11 June 2022) and in some cases two legs very close to cloud top can be called BCT such as RF13 on 1 March 2020 (Fig. 3b). We caution that the usage of these leg files is ideal for analyses depending on large amounts of statistics but that for more detailed case studies and/or for higher confidence of legs in or out of cloud for a certain percentage of time of the leg it
is important to look at as much data as possible to best understand the environmental conditions during a typical leg. An example of why this is important is for leg types in the immediate vicinity of clouds owing to the sometimes low cloud fraction and the changing structure of clouds, including sometimes multiple layers of clouds. For applications requiring high confidence in where a plane was relative to clouds, forward camera videos (Table 4) are very helpful.

**5.3 Aircraft collocation product**

To address the challenge of geographical and temporal collocation for two separate measurement platforms, a data collocation product (i.e., collocation mask) is available. This product is broadly applicable for any research where data from a secondary platform are required to be within some required spatio-temporal difference with the primary platform. To accommodate different needs, data files are archived when considering either the King Air or the Falcon as the primary platform.



Within the contents of each file are the primary platform's 1 Hz time series and collocated secondary platform time segments along with the corresponding horizontal distance (in km) between each aircraft at each time segment. A collocated time segment is one where the secondary platform is nearest to the primary platform within 15 km and 30 minutes. If there are multiple separate time segments, that means there were points where the two platforms flew outside of 15 km and back within the 30-minute time segment. Each period was checked, and the nearest collocated time stamp is provided with the corresponding horizontal separation (in km) between the platforms. There are a maximum of 10 collocated segments allowed for each 1 second time step. This product will be described in greater detail in forthcoming work (Schlosser et al., in review).

**5.4 Cloud detection neural network algorithm**

Above-aircraft clouds impact the downwelling and upwelling radiation fields by the King Air aircraft, and thus impact the measurements of airborne passive sensors and their retrieval products, such as the retrieved aerosol and cloud optical and microphysical products. For ACTIVATE, the forward-facing camera on the King Air (sect. 3.6) was used to create a manual cloud mask product indicating whether or not a cloud is present above the aircraft. In order to automate this process, the cloud detection neural network (CDNN) algorithm was developed to detect above-aircraft clouds efficiently and automatically using the camera images. The CDNN uses convolutional neural networks to find clouds using forward-viewing camera images. A center-top crop of the forward-facing camera's field of view is used to identify clouds closer in proximity to the aircraft. However, this crop may not be fully optimized such that clouds that are too far away to impact passive sensors onboard the aircraft may still be flagged as contaminated by above-aircraft clouds. Also, clouds that are not directly visible in the forward-facing camera, such as above-aircraft clouds behind the aircraft that are nonetheless blocking the sun, are unable to be detected. The description of the CDNN, its performance, and the resulting archived ACTIVATE cloud mask product results are detailed in Nied et al. (2023).

**5.5 MERRA-2 data along flight tracks**

The Modern-Era Retrospective analysis for Research and Applications, version 2 (MERRA-2) (Gelaro et al., 2017) is NASA's latest reanalysis generated with the Goddard Earth Observing System, version 5 (GEOS-5) atmospheric data assimilation system (Rienecker, 2008). It has a horizontal resolution of $0.5° \times 0.625°$ with 72 vertical levels from the surface to 0.01 hPa. Its aerosol reanalysis (Buchard et al., 2017; Randles et al., 2017) uses the GEOS-5 Goddard Aerosol Assimilation System (Buchard et al., 2015), which utilizes the Goddard Chemistry, Aerosol, Radiation, and Transport model (GOCART) (Chin et al., 2002) to simulate 15 externally mixed aerosol tracers: hydrophobic and hydrophilic black carbon (BC) and organic carbon (OC), dust (five size bins), sea salt (five size bins), and sulfate. GOCART includes wind speed-dependent emissions for dust and sea salt, fossil fuel combustion, biomass burning and biofuel emissions for primary sulfate and carbonaceous aerosols, and additional biogenic sources for organic carbon. Secondary sulfate is formed by chemical oxidation of $SO_2$ and DMS. Volcanic $SO_2$ emissions are included. The major sinks for aerosol particles are gravitational settling, dry deposition, and wet removal due to stratiform and convective precipitation. MERRA-2 assimilates AOD from ground and satellite-based remote sensing





sensors, including the Advanced Very High Resolution Radiometer (AVHRR), the Aerosol Robotic Network (AERONET), the Multi-angle Imaging Spectroradiometer (MISR), and the Moderate Resolution Imaging Spectroradiometer (MODIS/Terra and MODIS/Aqua). MERRA-2 aerosol data have been evaluated by Randles et al. (2017) for AODs and by Buchard et al. (2017) for aerosol vertical distribution and absorption.

We have archived a data product that samples MERRA-2 for selected 3-D fields along the Falcon flight tracks during the ACTIVATE deployments (Table 6). We interpolate the original MERRA-2 three hour instantaneous 3-D fields to the latitude,
longitude, and pressure altitude of the aircraft every 60 seconds along the flight track. Data files for February-March and August-September 2020 are archived and the product files for subsequent years are being generated for archival at the same location (details of accessibility in sect. 7). These sampled MERRA-2 data facilitate the comparison between aircraft measurements and reanalysis and provide quantities that are not measured during ACTIVATE (such as $SO_2$ concentration; Corral et al., 2022b). They are also useful for doing statistical analysis of aircraft in-situ data in comparison with reanalysis as
well as model evaluation.

## 5.6 FLEXPART back trajectory products

The Lagrangian transport and dispersion model, FLEXPART (FLEXible PARTicle dispersion model, https://www.flexpart.eu/) (Pisso et al., 2019; Eckhardt, 2008), is used to simulate transport pathways of air masses associated
with ACTIVATE aircraft measurements. In its backward mode, FLEXPART calculates trajectories of a multitude of particles and simulates advection, convection, and turbulent dispersion of the particles during the transport period. Detailed descriptions about FLEXPART transport schemes and parameterizations can be found in the literature (Eckhardt, 2008; Zhang et al., 2014). All FLEXPART simulations were driven by the Global Forecast System Analysis (GFS-ANL 003, 1° × 1°, 26 levels, 3 hourly; https://www.ncei.noaa.gov/data/global-forecast-system/access/grid-003-1.0-degree/analysis). FLEXPART version 9.2 was
used for the ACTIVATE February-March and August-September 2020 deployments. For 2021 and 2022 campaigns, FLEXPART v10.4 (Pisso et al., 2019) was used to accommodate the recent upgrade in the GFS-ANL data as well as to gain better capacity in simulating turbulence in the boundary layer. The purpose of this simulation series is to depict general transport pathways from a large-scale perspective. Model configurations here (e.g., output frequency, boundary layer turbulence) are not prioritized for small-scale analysis. The FLEXPART trajectory products for both 2020 and 2021 campaigns
are now available to assist with the analyses of aerosol sources and aging history associated with aircraft measurements; 2022 files are forthcoming.

In FLEXPART backward mode, a plume of passive particles is released from aircraft location and advected and dispersed backwards in time. For each 60 second-merged aircraft measurement every 10 minutes, FLEXPART initiates 10,000 passive particles at the sampling location and calculates backwards for 10 days, resulting in a spatiotemporal distribution of air parcel
residence time (RT). The spatial distribution of RT can be readily used to determine transport patterns (e.g., height, timescale). The trajectory product is provided in a format of a spatial distribution of RT, which can be used to indicate air mass transport pathways and determine pollution sources.



For each of the six ACTIVATE deployment periods, two types of files can be found in the ACTIVATE data. One type includes trajectory plots associated with aircraft data of every 10 minutes. For each trajectory, a map plot and a vertical plot of RT distributions are included. Examples are shown in Fig. 8 for aircraft measurements at 19:22 UTC during the second flight of 1 March 2020 that is discussed in more detail in sect. 6. These plots are generated for quick-look purpose to visualize transport pathways, and the plot quality is thus constrained to limit total file size.

The other file type includes original FLEXPART output for 10-day backward trajectories released every 10 minutes along flight tracks. Each netCDF file contains gridded specific residence time (RT, "s m$^3$ kg$^{-1}$") of all released particles. RT is saved in such a unit instead for time ("s") so that it can be easily multiplied by any upwind source / emission ("kg m$^{-3}$ s$^{-1}$") to calculate source contributions affecting the receptor point. For example, FLEXPART RT can be used to calculate a time series of tracer concentrations at the receptor contributed by a certain emission source (e.g., anthropogenic or biomass burning) by multiplying the residence time in the lowest 300 m by the emission flux.

Uncertainties in transport pathways simulated by FLEXPART can be due to the parameterizations representing temporally and spatially unresolved transport processes (Stohl et al., 2010). In terms of vertical transport processes, boundary layer mixing and convective updrafts are both treated in FLEXPART using information from the driving meteorology. Time-varying planetary boundary layer (PBL) height determines the vertical mixing of air parcels. In FLEXPART, PBL height is calculated using the Richardson number concept based on the wind and temperature fields (Vogelezang and Holtslag, 1996). Another highly parameterized sub-grid process is cloud convection. FLEXPART redistributes air parcels vertically in convection-activated grids using the approach of Emanuel and Živković-Rothman (1999), which determines air parcel displacement in up- and down-drafts based on temperature and humidity fields. Model results with such schemes have been tested and validated using surface and in situ measurements (Brioude et al., 2013; Stohl et al., 1998).

**5.7 MODIS, GOES-16, MERRA-2**

To assist data analysis efforts for ACTIVATE that can benefit from contextual satellite and reanalysis data for overlapping and prior time periods, various satellite and reanalysis data products are archived with a common format and spatial resolution. This dataset is intended to facilitate understanding of the large-scale meteorological context of the ACTIVATE domain. Merged satellite-reanalysis daily files combine 3D meteorological fields from MERRA-2 (already described in sect. 5.5) with daytime aerosol and cloud properties derived from MODIS on Aqua (~ 1:30 pm overpass time) for the January 2009-July 2022 period and the domain defined by the 84.5˚W-30.5˚W, 10.5˚N-59.5˚N box. MODIS cloud retrievals are taken from the Cloud and the Earth's Radiant Energy System (CERES) Edition 4 (Minnis et al., 2021) level 3 Single Scanner Footprint (SSF1deg-Day), gridded at 1˚ × 1˚ resolution. CERES-MODIS cloud properties in the merged file are cloud amount, cloud effective pressure, cloud effective temperature, cloud effective height, cloud particle effective radius (ice and liquid) derived using the 3.7 μm channel, water path (ice and liquid), cloud optical depth, and liquid cloud droplet number concentration estimated following Painemal (2018). MODIS aerosol optical depths (Levy et al., 2013) at 1˚ × 1˚ resolution for 7 wavelengths (0.47 μm, 0.55 μm, 0.66 μm, 0.86 μm, 1.24 μm, 1.63 μm, and 2.13 μm) are obtained from the MODIS Level 3 Atmospheric Gridded



Product Collection 6 (MYD08_D3). Examples of ACTIVATE applications of this dataset include climatological characterization of the atmospheric circulation and cloud field (Painemal et al., 2021), assessment of the meteorological factors that modulate clouds and aerosol variability and their implications for aerosol-cloud interactions (Dadashazar et al., 2021b),

and description of the synoptic-scale processes that give rise to boundary layer cloud variability (Painemal et al., 2023).

MERRA-2 meteorological parameters at 0.625˚ × 0.5˚ resolution are spatially collocated with MODIS via nearest neighbor interpolation. We selected MERRA-2 products at 18:00 UTC as it is the closest match to the Aqua overpass time for the northwest Atlantic. In addition, 15 isobaric levels are stored, corresponding to (units of hPa): 1000, 975, 950, 925, 900, 875, 850, 825, 800, 775, 750, 725, 700, 650, 600. MERRA-2 3D fields (longitude × latitude × vertical level) include: air temperature,

RH, sea level pressure, edge heights, eastward wind, northward wind, vertical pressure velocity; and 2D fields (at a fixed vertical level) are: surface skin temperature, 2-m eastward wind, 2-m northward wind, and lifting condensation level.

Cloud retrievals from the Advanced Baseline Imager (ABI) on the 16[th] Geostationary Operational Environmental Satellite (GOES-16) are derived using the NASA Satellite ClOud and Radiation Property System (SatCORPS) algorithms (Minnis et al., 2008; Minnis et al., 2021). SatCORPS algorithms have been adapted from those for CERES-MODIS to take advantage of

radiometric channels similar to those of MODIS and other Earth-orbiting satellites (Minnis et al., 2021). Additional consistency between MODIS and GOES-16 is achieved by calibrating GOES-16 visible radiance against its Aqua-MODIS counterpart following Doelling et al. (2018). GOES-16 cloud retrievals are produced every 20 minutes during the ACTIVATE deployment. Files are archived for two regions covering the ACTIVATE flight tracks: a small domain (78˚W-60˚W, 29˚N-46˚N), and a large domain (93˚W-49˚W, 18˚N-55˚N). Cloud properties for the small domain are produced at the native resolution of the

infrared channels, that is, 2 km at nadir. For the large domain, 2-km cloud properties are subsampled every other pixel to achieve a spatial resolution of 4 km. Cloud products derived from GOES-16 include cloud mask and phase, temperature, height and pressure, particle effective radius (ice and liquid), water path (ice and liquid), and optical depth. The ability of GOES-16 products of resolving the diurnal cycle at a relatively high spatial resolution makes the retrievals particularly useful for describing the evolution of the cloud fields during the research flights (GOES-16 snapshots are included in the flight reports

described in Sect. 5.1). GOES-16 products have been used in the context of ACTIVATE for validating mesoscale simulations of clouds (Chen et al., 2022b), assessing the evolution of liquid water path in large eddy simulation (LES) experiments (Li et al., 2022), and for quantifying the cloud-top entrainment rate and its role in the CCN budget (Tornow et al., 2022). In addition, GOES-16 retrievals are well suited for matching with the aircraft tracks to complement in-situ observations, and for Lagrangian studies.


## 6 Case flight example

The afternoon joint flight on 1 March 2020 is highly representative of the majority of the ACTIVATE flight dataset in terms of how the aircraft flew and the science that was targeted. This section aims to share representative data collected to summarize how the aforementioned data products in Sections 3-5 can be visualized and used; this day of flights was also summarized





during an open data workshop that was recorded and archived at https://asdc.larc.nasa.gov/news/activate-data-webinar-materials. This particular day was highest ranked in the scientific goals of the weather forecasting meeting on the previous day and considered an excellent flight day. This is because of forecasted cold air outbreak (CAO) indicators of boundary layer instability (Papritz et al., 2015; Painemal et al., 2021; Fletcher et al., 2016) coinciding with strong, cold, northwesterly winds and "cloud streets" (Dadashazar et al., 2021b). The day was forecasted also to have high cloud fraction and without high level

cirrus and mid-tropospheric cloud layers that would negatively impact remote sensing objectives. Forecasting analysis conducted the previous day suggested there would be a broken to overcast low cloud deck (deepening to the east) with a western edge moving farther offshore throughout the day. GEOS forward processing data hinted at fairly low aerosol loading, with increasing sea salt concentrations offshore. Actual conditions were consistent with forecasted information.

The first joint flight of 1 March 2020 was a process study flight (Fig. 3a) since the aircraft transited to an area of high interest

and conducted maneuvers deviating from the ensemble approach shown in Fig. 2. More specifically, the Falcon conducted stacked level legs (a "wall") approximately perpendicular to the estimated boundary layer winds while the King Air flew a large circle encompassing the wall location followed by an overpass of the extended axis of the Falcon wall. This particular flight has also been simulated and discussed in recent studies (Chen et al., 2022a; Li et al., 2022; Tornow et al., 2022). Both aircraft returned to the base of operations (Newport News) to refuel and then returned to the same region as the morning, flying

a downwind survey that started at the wall center point and extended as far as fuel permitted (Fig. 9a). The downwind survey leg allowed for a semi-Lagrangian characterization of the air mass evolution and also resampled the air mass from the morning flight. Both flights captured elements of the cloud morphology common to CAOs, but the afternoon flight characterized the evolution from the upwind clear region to scattered cumulus transforming into a thicker and more extensive layer before finally transitioning into open-cellular stratocumulus organization. This can be seen from flight tracks overlaid on GOES-16 visible

imagery (Fig. 9a).

Shown already in Fig. 8 were FLEXPART simulation results pertaining to air mass trajectories arriving at the point of the Falcon during this flight at 19:22 UTC. Figure 10 shows the level of detail possible with dropsondes, with the markings of where the two were launched shown in Fig. 9a with nadir camera imagery from the King Air at those times in Fig. 9b. Representative data from the HSRL-2 in the form of vertical 'curtains' of aerosol backscatter as a function of flight time are

shown in Fig. 9c; these data show higher aerosol loading is located in the MBL closest to the ocean surface. This panel shows the altitude of the Falcon while flying below the King Air aircraft as well as the locations where the dropsondes were launched from the King Air.

Figure 11 summarizes selected variables measured by the Falcon in time series format. The dashed vertical black bars denote the beginning of either clear or cloud ensembles. The first ensemble begins right after the high altitude transit after takeoff and

was a clear ensemble with the following legs in order (MinAlt, ABL, BBL, RS, MinAlt). That ensemble was followed by three consecutive cloud ensembles with the first two containing the nominal order of legs described in sect. 2.2 while the third ensemble was truncated at MinAlt owing to the absence of clouds, which is clearly visible in Fig. 9a with clear conditions closer to the coast. The vertical gray shaded bars make use of leg index files (sect. 5.2) and distinguish the two level-leg types





in cloud including ACB (above cloud base) and BCT (below cloud top). Clearly those periods are marked by enhancements in $N_d$ and LWC as measured by the FCDP, but note that cloud penetrations also occur outside of designated cloud legs, such as during altitude transitions. Many of the other plotted variables associated with trace gases, aerosol particles, temperature, and wind data show interesting structure that at least partly have dependence on aircraft altitude, which can be teased out in these forms of multi-panel time series depictions as in Fig. 11 that can aid data users. Aerosol microphysical data have been screened to remove data collected in clouds and, in the case of the LAS (which was used to determine number concentration

above 100 nm), using the inlet flag variable to remove CVI data from this illustration. Note that AMS data are archived separately for isokinetic and CVI time periods, so this screening is not necessary for the AMS. An important note with the aerosol composition data is that the PILS data for $Na^+$, used here as a proxy for sea salt that the AMS cannot provide, have coarser time resolution than the AMS. Also, the PILS data include influence from cloud periods and thus data users need to use caution about such data in terms of what their applications are. If data users want aerosol data without any cloud

contamination, they should only use PILS data in cloud-free areas such as clear ensembles and transit periods. For interested readers, a figure analogous to Fig. 7 is shown in Fig. S1 for this case flight too to demonstrate again how to conduct closure types of analyses between different data parameters such as aerosol number concentration in this case.

   Lastly, Fig. 12 provides a summary of cloud probe products specifically from the FCDP and 2D-S combination probe from the Falcon's port side wing. Figure 12a shows a time series of cloud droplet size distributions from the FCDP combined with

2D-S. Sections with cloud penetrations are clearly visible with enhanced number concentrations above 10 µm. Also evident from the time series are periods with noticeable number concentrations below 10 µm during periods without clouds, which is indicative of coarse aerosol particles such as sea salt. Figure 12b shows various forms of size distributions that data users can produce from FCDP alone, in addition to the 2D-S/FCDP combination and 2D-S horizontal ice and liquid products. The stitched size distribution for 2D-S/FCDP was explained briefly in sect. 4.5 and described more extensively by Kirschler et al.

(2022). The 2D-S imagery in Fig. 12c covers a 20 second period that nicely represents a broad variety of large particle shapes, including liquid droplets and rimed ice particles.

## 7 Data/code availability and file format

   NASA's Atmospheric Science Data Center (ASDC) plays a key role in data curation and dissemination as they archive the

latest versions of publication quality data, including observational, derived, and value-added data products and is responsible for long-term preservation of ACTIVATE data. They also house contextual information to facilitate data use by research community at large, in addition to documentation for maintaining reprocessing capability and openness. Digital Object Identifiers (DOIs) are assigned at both the project-level and data product (collection) level for ACTIVATE. All data from the King Air and Falcon, including complementary data products from sect. 5, unless otherwise stated, are publicly archived on

ASDC's Distributed Active Archive Center (DAAC; ACTIVATE Science Team, 2020) and accessible through the ACTIVATE landing page: https://asdc.larc.nasa.gov/project/ACTIVATE, with each data file containing data from one flight





or one calendar day. Various tabs at that webpage include different data products (collections) with their unique DOI codes, which are summarized in Table 7 along with other resources described in this paper. The open data workshop content listed in Table 7 is especially important to guide new data users through each step of the process to access and visualize data beginning

with establishing a free account at earthdata.nasa.gov and then proceeding to download ACTIVATE data with the Sub-Orbital Order Tool (SOOT; https://asdc.larc.nasa.gov/soot/power-user). ACTIVATE data are also available to download via Earthdata Search: https://search.earthdata.nasa.gov/search?fpj=ACTIVATE.

Most files are in a special format called ICARTT files (Northup et al., 2017), which is traditionally used by NASA and other agencies for airborne data. Falcon in-situ observations are reported in ICARTT format, while remote sensing data uses a

combination of ICARTT format and HDF format. It is critical for any data user aiming to use airborne science data to review the ICARTT file headers that provide guidance for how to both use and interpret data from individual instruments.

File names constitute the following details in order: campaign, instrument, sampling method, start date, revision number, and the (optional) end date. Publication-quality data include a revision number in their file name (R0+) and are time synced to the platform time standard (DLH instrument time for Falcon and GPS time for King Air). The contents of each ICARTT file

include data notes in a README tab including principal investigator (PI) name, PI institution, campaign name, start date of data collection, the most recent data revision date, the number of variables, data flags, instrument details and description of the data, and revision log. The revision log states what revision the data is currently on and lists the previous revisions and their relative status. Each instrument will have its own unique column headers based on what was being measured.

While the instrument teams have time synchronized datasets with one another to account for different sampling techniques

(e.g., varying times for sample air to travel from an inlet to instruments), it is noted still though that it is possible that a variation of a few seconds can occur. No post-submission time alignment is done by the data management team, merge process, or ASDC DAAC and thus data users should use diligence when using multiple datasets together to do some intercomparisons and confirm temporal variations of related parameters match one another without obvious systematic shifts.

## 8 Conclusions

A collection of airborne datasets is introduced here that serves as a resource for investigations of aerosol-cloud-meteorology interactions, along with studies more interested in measurements of exclusively just trace gases, aerosol particles, clouds, precipitation, and/or atmospheric state parameters. The datasets cover the northwest Atlantic extending from the coastal area of the mid-Atlantic states and New England to much farther offshore around the vicinity of Bermuda where more remote marine conditions are present that are less perturbed by continental emissions. The data span all seasons with collection periods

between November-June and August-September for 2020 through 2022. This paper serves as a resource for any potential data users to guide them in what data products are available with associated descriptions and how to access them. Of particular interest to most data users of the Falcon data is likely the merged dataset of variables generated at different time resolutions of interest (e.g., 1, 5, 10, 15, 30, 60 s, or matching an individual data product start and stop times). Data products and codes have



also been developed to help users for joint analysis of data between the two aircraft based on specific criteria of interest related

to time and space separation.

**Appendix A: Summary of abbreviations**

| Abbreviation | Definition |
| --- | --- |
| 2D-S | Two-Dimensional Stereo |
| ABI | Advanced Baseline Imager |
| ABL | Above boundary layer top |
| AC3 | Axial cyclone cloud water collector |
| ACB | Above cloud base |
| ACE-ENA | Aerosol and Cloud Experiments in the Eastern North Atlantic |
| ACT | Above cloud top |
| ACTIVATE | Aerosol Cloud meTeorology Interactions oVer the western ATlantic Experiment |
| AERONET | Aerosol Robotic Network |
| AMS | Aerosol mass spectrometer |
| AOD | Aerosol optical depth |
| ASDC | Atmospheric Science Data Center |
| ASTER | Advanced Spaceborne Thermal Emission and Reflection Radiometer |
| AVAPS | Airborne Vertical Atmospheric Profiling System |
| AVHRR | Advanced Very High Resolution Radiometer |
| BBL | Below boundary layer top |
| BC | Black carbon |
| BCB | Below cloud base |
| BCT | Below cloud top |
| BLEACH | Bermuda boundary Layer Experiment on the Atmospheric Chemistry of Halogens |
| BMI | Brechtel Manufacturing Inc. |
| CALIPSO | Cloud-Aerosol Lidar and Infrared Pathfinder Satellite Observations |
| CAMP$^2$Ex | Cloud, Aerosol and Monsoon Processes Philippines Experiment |
| CAO | Cold air outbreak |
| CAS | Cloud and aerosol spectrometer |
| CCN | Cloud condensation nuclei |
| CDNN | Cloud detection neural network |
| CDP | Cloud droplet probe |
| CERES | Cloud and the Earth's Radiant Energy System |
| $CH_4$ | Methane |
| CO | Carbon monoxide |
| $CO_2$ | Carbon dioxide |
| CN | Condensation nuclei |



| CPC | Condensation particle counter |
|---|---|
| CVI | Counterflow virtual impactor |
| DAAC | Distributed Active Archive Center |
| DJF | December-January-February |
| DLH | Diode laser hygrometer |
| DMT | Droplet Measurement Technologies |
| DOI | Digital object identifier |
| EVS-3 | Earth venture suborbital - 3 |
| f(RH) | Ratio of total light scattering between high and low RHs |
| FCDP | Fast cloud droplet probe |
| FLEXPART | FLEXible PARTicle dispersion model |
| GEOS-5 | Goddard Earth Observing System, version 5 |
| GOCART | Goddard Chemistry, Aerosol, Radiation, and Transport model |
| GOES | Geostationary Operational Environmental Satellite |
| GPS | Global positioning system |
| $H_2O(v)$ | Water vapor |
| HDF | Hierarchical data format |
| HSRL-2 | High Spectral Resolution Lidar - generation 2 |
| IC | Ion chromatography |
| ICARTT | International Consortium for Atmospheric Research on Transport and Transformation |
| ICP-MS | Inductively coupled plasma mass spectrometry |
| IMPACTS | Investigation of Microphysics and Precipitation for Atlantic Coast-Threatening Snowstorms |
| IR | Infrared |
| JJA | June-July-August |
| LaRC | Langley Research Center (NASA) |
| LARGE | Langley Aerosol Research Group Experiment |
| LAS | Laser Aerosol Spectrometer |
| LES | Large eddy simulation |
| LWC | Liquid water content |
| MBL | Marine boundary layer |
| MERRA-2 | Modern-Era Retrospective analysis for Research and Applications, version 2 |
| MinAlt | Minimum altitude the Falcon can fly at |
| MISR | Multi-angle Imaging Spectroradiometer |
| MLH | Mixed-layer height |
| MODIS | Moderate Resolution Imaging Spectroradiometer |
| $N_a$ | Aerosol particle number concentration |
| NAAMES | North Atlantic Aerosols and Marine Ecosystems Study |
| NASA | National Aeronautics and Space Administration |
| NCAR | National Center for Atmospheric Research |



| $N_d$ | Cloud droplet number concentration |
|---|---|
| netCDF | Network Common Data Form |
| $NO_x$ | Nitrogen oxides |
| $O_3$ | Ozone |
| OC | Organic carbon |
| OTREC | Organization of Tropical East Pacific Convection |
| PBL | Planetary boundary layer |
| PILS | Particle-into-liquid sampler |
| PI | Principal investigator |
| PPT | Precision pressure transducers |
| PSAP | Particle soot absorption photometer |
| RF | Research flight |
| RH | Relative humidity |
| RS | Remote sensing |
| RSP | Research scanning polarimeter |
| RT | Residence time |
| SatCORPS | Satellite ClOud and Radiation Property System |
| SEAC$^4$RS | Studies of Emissions and Atmospheric Composition, Clouds and Climate Coupling by Regional Surveys |
| SMPS | Scanning mobility particle sizer |
| $SO_2$ | Sulfur dioxide |
| SOOT | Sub-Orbital Order Tool |
| SSA | Single scattering albedo |
| SSF | Single Scanner Footprint |
| STP | Standard temperature and pressure |
| TAMMS | Turbulent Air Motion Measurement System |
| TAS | True airspeed |
| UTC | Coordinated Universal Time |
| VIIRS | Visible Infrared Imaging Radiometer Suite |
| VOC | Volatile organic compound |

**Author contributions**

Conceptualization, resources, funding acquisition, supervision: AS, JWH, RCF, XZ

Writing – original draft preparation: AS

Project administration: MK

Data curation: GC, JMK, KEP, MEB, MAS, NJ, SL

Formal analysis, investigation, methodology, software, validation, visualization: All authors

Writing - reviewing and editing: All authors



**Competing interests**

The authors declare that they have no conflict of interest.

**Disclaimer**

Publisher's note: Copernicus Publications remains neutral with regard to jurisdictional claims in published maps and institutional affiliations.

**Acknowledgements**

The authors thank pilots and aircraft maintenance personnel of NASA Langley Research Services Directorate for successful execution of ACTIVATE flights. The work was funded by ACTIVATE, a NASA Earth Venture Suborbital-3 (EVS-3) investigation funded by NASA's Earth Science Division and managed through the Earth System Science Pathfinder Program Office. We thank pilots and aircraft maintenance personnel of NASA Langley Research Services Directorate for successful execution of ACTIVATE flights.


**Financial support**

University of Arizona investigators were supported by the National Aeronautics and Space Administration grant no. 80NSSC19K0442. C.V. and S.K. thank funding by the Deutsche Forschungsgemeinschaft (DFG, German Research Foundation) – TRR 301 – Project-ID 428312742 and SPP 1294 HALO under contract VO 1504/7-1. National Institute of

Aerospace investigators were supported by NASA grant 80NSSC19K0389. J.S.S. was supported by an appointment to the NASA Postdoctoral Program at NASA Langley Research Center, administered by Oak Ridge Associated Universities under contract with NASA. University of Miami gratefully acknowledges financial support through NASA grant 80NSSC19K0390.

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



Table 1. Overall summary of ACTIVATE flight metrics categorized by each of the six deployments between 2020 and 2022. Joint ensembles represent when both planes were in coordination and conducting the series of legs (in some combination) shown in Fig. 2. The number of dropsondes shown represent those with full profiles of all variables with good parachute performance.

| | Research Flights | | | Flight Hours | | Joint Ensembles | | Underflights | | Process Study Flights | Dropsondes |
| --- | --- | --- | --- | --- | --- | --- | --- | --- | --- | --- | --- |
| | Falcon | King Air | Joint | Falcon | King Air | Cloudy | Clear | ASTER | CALIPSO | | |
| Winter 2020 (14 Feb – 12 Mar) | 22 | 17 | 17 | 73 | 59 | 43 | 28 | 1 | - | 2 | 59 |
| Summer 2020 (13 Aug – 30 Sep) | 18 | 18 | 18 | 60 | 67 | 58 | 36 | 1 | 3 | 2 | 107 |
| Winter 2021 (27 Jan – 2 Apr) | 17 | 19 | 15 | 56 | 66 | 47 | 25 | 1 | 3 | - | 100 |
| Summer 2021 (13 May – 30 Jun) | 32 | 32 | 32 | 106 | 108 | 103 | 74 | 1 | 1 | 2 | 150 |
| Winter 2021-2022 (30 Nov – 29 Mar) | 55 | 54 | 53 | 182 | 193 | 198 | 72 | - | 1 | 2 | 214 |
| Summer 2022 (3 May – 18 Jun) | 30 | 28 | 27 | 97 | 98 | 86 | 46 | 2 | 3 | 4 | 155 |
| *Sum* | *174* | *168* | *162* | *574* | *592* | *535* | *281* | *6* | *11* | *12* | *785* |





Table 2. Summary of ACTIVATE research flights with pertinent details associated with date, times, and special notes. Research flights 48-61 included a reduced operational Falcon payload due to an aircraft maintenance limitation. Deployments are separated by blank rows: Deployment 1 (RF1-RF22), Deployment 2 (RF23-RF40), Deployment 3 (RF41-RF61), Deployment 4 (RF62-RF93), Deployment 5 (RF94-RF148), Deployment 6 (RF149-RF179).

| RF | Date | Joint/ Single | Flight Type | King Air Take Off (UTC) | King Air Land (UTC) | # Sondes | HU-25 Falcon Take Off (UTC) | HU-25 Falcon Land (UTC) | Special Notes |
|---|---|---|---|---|---|---|---|---|---|
| 1 | 2/14/2020 | Joint | Statistical Survey | 17:04:42 | 20:35:34 | 4 | 17:01:23 | 20:04:20 | Landed at Newport News and stationed there until end of Winter 2020 deployment |
| 2 | 2/15/2020 | Joint | Statistical Survey | 16:42:19 | 19:55:40 | 4 | 16:48:20 | 19:58:02 | Some precipitation and air traffic challenges affecting Falcon ensemble leg order |
| 3 | 2/17/2020 | Joint | Statistical Survey | 16:04:11 | 19:18:04 | 4 | 16:02:55 | 19:18:35 | Relatively cloud-free with relatively high number of clear ensembles |
| 4 | 2/21/2020 | Single-Falcon | Statistical Survey | N/A | N/A | 0 | 18:37:28 | 21:55:03 | King Air maintenance issue; spiral sounding and 'wall' pattern |
| 5 | 2/22/2020 | Single-Falcon | Statistical Survey | N/A | N/A | 0 | 13:54:11 | 17:02:40 | King Air maintenance; characterize area downwind of where the next flight focused on |
| 6 | 2/22/2020 | Single-Falcon | Statistical Survey | N/A | N/A | 0 | 18:59:14 | 22:26:40 | King Air maintenance; wall pattern focusing on air mass sampled in RF5 in morning; spiral soundings |
| 7 | 2/23/2020 | Single-Falcon | Statistical Survey | N/A | N/A | 0 | 13:30:55 | 16:54:06 | King Air maintenance; Notes of MBL being more shallow closer to land with colder water |
| 8 | 2/23/2020 | Single-Falcon | Statistical Survey | N/A | N/A | 0 | 18:25:54 | 21:55:32 | King Air maintenance; transited high to far east point to buy range and save fuel; descended for cloud wall and then stat surveys back to base; precip below cloud |
| 9 | 2/27/2020 | Joint | Statistical Survey | 18:05:40 | 21:30:10 | 2 | 17:56:35 | 21:27:05 | Falcon conducted multiple "racetrack" delay loops to improve spatial coordination with King Air |
| 10 | 2/28/2020 | Joint | Process Study | 14:05:07 | 18:18:53 | 11 | 14:20:42 | 17:41:44 | Complex cloud scene with multiple cloud types in a single column where "wall" and associated spiral sounding occurred; 11 dropsondes |
| 11 | 2/28/2020 | Joint | Statistical Survey | 19:20:00 | 23:25:46 | 2 | 19:36:01 | 22:49:25 | Captured the evolution of the complex cloud field in the previous flight within the circle |



| 12 | 2/29/2020 | Joint | Statistical Survey | 14:28:32 | 17:46:31 | 2 | 13:51:55 | 17:37:27 | Forecasted to be clear but was actually a good cloudy day; Falcon "racetrack" delay loop to improve coordination |
| 13 | 3/1/2020 | Joint | Process Study | 13:37:05 | 17:22:45 | 11 | 13:31:37 | 17:04:24 | Cold air outbreak with same flight plan as RF10; 11 dropsondes |
| 14 | 3/1/2020 | Joint | Statistical Survey | 18:36:49 | 22:05:44 | 2 | 18:32:24 | 21:47:50 | Captured the evolution of the complex cloud field in the previous flight within the circle |
| 15 | 3/2/2020 | Joint | Statistical Survey | 16:55:22 | 20:10:15 | 2 | 16:54:05 | 20:02:28 | Biomass burning sampled towards end of flight; changing cloud base heights and precipitation observed with Falcon trying to optimize levels to maximize time in cloud |
| 16 | 3/6/2020 | Joint | Statistical Survey | 18:19:06 | 21:45:24 | 3 | 18:09:58 | 21:28:19 | High cloud fraction |
| 17 | 3/8/2020 | Joint | Statistical Survey | 14:17:09 | 17:09:00 | 2 | 13:48:48 | 17:00:21 | Good cloud flight |
| 18 | 3/8/2020 | Joint | Statistical Survey | 18:25:20 | 21:56:15 | 2 | 18:32:39 | 21:57:45 | Nearly identical track to RF17 from morning; forecasted clear but there were clouds |
| 19 | 3/9/2020 | Joint | Statistical Survey | 16:15:08 | 19:58:44 | 2 | 16:33:40 | 19:51:15 | Observations of smoke on return to base (visual and from HSRL-2) |
| 20 | 3/11/2020 | Joint | Statistical Survey | 12:39:30 | 15:47:06 | 2 | 12:44:39 | 15:40:26 | Real-time maneuvering with new waypoints and altitude changes required in flight due to convective weather |
| 21 | 3/12/2020 | Joint | Statistical Survey | 13:45:47 | 17:20:17 | 2 | 14:07:19 | 17:15:37 | ASTER underflight; northern end of the ASTER track had reduced cirrus compared to southern end |
| 22 | 3/12/2020 | Joint | Statistical Survey | 19:00:18 | 22:30:17 | 2 | 18:57:32 | 22:16:50 | Convective weather and icing concerns caused some King Air deviations in flight track; precipitation observed |
| 23 | 8/13/2020 | Joint | Statistical Survey | 13:55:26 | 17:24:09 | 5 | 14:04:50 | 17:26:11 | Convective weather with lightning; potential cold pool area; gradient in $CO_2$ and $CH_4$ on the southern end of track due to presumed different air mass |
| 24 | 8/17/2020 | Joint | Statistical Survey | 14:31:44 | 18:17:05 | 6 | 14:28:24 | 17:55:34 | Smoke observed at high altitude |



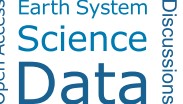

| 25 | 8/20/2020 | Joint | Statistical Survey | 14:01:57 | 17:35:37 | 5 | 13:59:39 | 17:23:26 | Forecasted to have minimal low cloud but had good low cloud (similar to RF12); high $N_d$ values; did special maneuvers to improve aircraft coordination during flight; low cloud LWC prevented cloud water collection |
| 26 | 8/21/2020 | Joint | Statistical Survey | 13:59:46 | 17:33:17 | 5 | 14:01:30 | 17:11:51 | Low cloud LWC prevented cloud water collection; King Air maneuvered to avoid flying in cirrus |
| 27 | 8/25/2020 | Joint | Statistical Survey | 13:57:23 | 17:57:51 | 6 | 14:03:00 | 17:25:15 | Less cloud vertical development compared to previous Summer 2020 flights; note of distinct sulfate layer above cloud tops; HSRL-2 observed high altitude aerosol layers; lack of cloud water due to low LWC |
| 28 | 8/26/2020 | Joint | Statistical Survey | 13:54:06 | 17:41:47 | 6 | 13:52:27 | 17:08:11 | CALIPSO underflight; smoke layers; unicorn aerosol module (described in sect. 2.4) with polluted conditions during Falcon vertical spiral sounding |
| 29 | 8/28/2020 | Joint | Statistical Survey | 16:33:23 | 20:25:59 | 8 | 16:44:03 | 20:02:19 | Falcon transited at high altitude at start and end to accommodate CALIPSO overpass location as it was a CALIPSO underflight; mostly cloud-free; smoke; unicorn aerosol module |
| 30 | 9/2/2020 | Joint | Statistical Survey | 15:14:31 | 19:07:24 | 6 | 15:23:58 | 18:45:19 | High variability in MBL height and cloud fraction, along with vertically developing clouds making it challenging to do all cloud ensemble legs in order |
| 31 | 9/3/2020 | Joint | Statistical Survey | 14:33:04 | 18:13:51 | 6 | 14:43:47 | 17:50:43 | Precipitation noted during flight; a higher aerosol scattering day than normal potentially due to smoke |
| 32 | 9/10/2020 | Joint | Statistical Survey | 16:56:25 | 20:01:34 | 4 | 17:05:12 | 20:02:56 | Generally cleaner conditions than normal with low $N_a$ and $N_d$ |
| 33 | 9/11/2020 | Joint | Statistical Survey | 14:10:24 | 17:43:19 | 6 | 14:28:40 | 17:40:09 | ASTER underflight; ATC challenges led to Falcon being higher than desired at times |
| 34 | 9/15/2020 | Joint | Statistical Survey | 15:53:39 | 19:42:08 | 6 | 16:04:50 | 19:17:38 | Smoke observed; higher cloud fraction and vertically constrained clouds as compared to previous Summer 2020 flights |



| 35 | 9/16/2020 | Joint | Process Study | 15:49:49 | 19:33:10 | 0 | 15:58:52 | 19:26:54 | Easterly winds at times allowed for sampling of cloud processed air closer to shore west of clouds and the wall pattern; notes of possible smoke in air |
| 36 | 9/21/2020 | Joint | Statistical Survey | 16:03:45 | 20:01:10 | 5 | 16:15:11 | 19:36:09 | High sea salt due to high winds; high number of cloud water samples (10) |
| 37 | 9/22/2020 | Joint | Statistical Survey | 17:35:20 | 21:47:53 | 7 | 17:51:57 | 21:27:29 | Relatively high $N_d$ (in contrast with lower values previous day); significant aerosol gradients |
| 38 | 9/23/2020 | Joint | Statistical Survey | 16:39:21 | 20:16:08 | 8 | 16:33:18 | 20:11:57 | CALIPSO underflight; smoke influence from western N. America; relatively cloud-free day with low cirrus |
| 39 | 9/29/2020 | Joint | Process Study | 14:04:03 | 18:02:49 | 13 | 14:01:18 | 17:22:08 | King Air did a "Wheel and Spoke" pattern; Falcon wall had many vertical levels flown; 13 dropsondes |
| 40 | 9/30/2020 | Joint | Statistical Survey | 15:59:23 | 19:38:21 | 5 | 16:07:38 | 19:31:33 | Good $N_d$ gradients; turbulent Falcon flight; dry conditions noted aloft typical of post-frontal conditions |
| 41 | 1/27/2021 | Single-Falcon | Statistical Survey | N/A | N/A | 0 | 17:59:24 | 20:38:19 | Extra high altitude work for instrument quality control checks; Pilot staffing limitations allow for single aircraft flights this week (RF41-43) |
| 42 | 1/29/2021 | Single-King Air | Statistical Survey | 12:57:24 | 15:52:52 | 2 | N/A | N/A | Cold air outbreak |
| 43 | 1/29/2021 | Single-Falcon | Statistical Survey | N/A | N/A | 0 | 17:40:12 | 20:39:41 | Cold air outbreak; flew in same area as morning flight; steam fog that visible atop ocean surface in a band near SST rise; turbulence observed; icing motivated descents to MinAlt for shedding; supercooled droplets to mixed phase as plane moved downwind; cloud base changes significant as crossed Gulf Stream edge; uptrend in $SO_4$ offshore and a significant change in the aerosol size distribution between MBL and the coastal PBL |



| 44 | 2/3/2021 | Joint | Statistical Survey | 14:10:34 | 17:23:42 | 5 | 14:14:14 | 17:18:16 | Captured transition from SCu clouds to open cell cloud field; possible Asian dust; icing was issue in BCT legs; cloud water collected near and below bases during precipitation |
|----|----------|-------|--------------------|----------|----------|---|----------|----------|---|
| 45 | 2/10/2021 | Single-King Air | Statistical Survey | 15:05:09 | 18:43:58 | 2 | N/A | N/A | Falcon ground this and next two flights for maintenance issue |
| 46 | 2/20/2021 | Single-King Air | Statistical Survey | 14:50:18 | 18:04:45 | 8 | N/A | N/A | Cold air outbreak; characterized transition from clear to closed cell to open cell |
| 47 | 2/21/2021 | Single-King Air | Statistical Survey | 14:28:01 | 18:23:45 | 10 | N/A | N/A | Cold air outbreak; characterized transition from clear to closed cell to open cell |
| 48 | 3/4/2021 | Joint | Statistical Survey | 17:44:46 | 20:50:07 | 6 | 17:47:39 | 20:46:46 | CALIPSO underflight; first flight with reduced Falcon payload for Winter 2021 campaign |
| 49 | 3/5/2021 | Joint | Statistical Survey | 13:43:52 | 17:11:24 | 5 | 13:40:51 | 17:07:59 | Evolution of cold air outbreak cloud field potential high altitude aerosol layer due to dust; high cloud bases and cold clouds |
| 50 | 3/5/2021 | Joint | Statistical Survey | 18:40:27 | 21:56:57 | 5 | 18:43:16 | 21:51:03 | Characterized upwind aerosol data feeding the cloud field sampled in first flight; many notes from morning flight apply here too |
| 51 | 3/8/2021 | Joint | Statistical Survey | 16:59:05 | 20:06:56 | 4 | 16:57:24 | 20:19:25 | Cold air outbreak conditions; clouds were shallow overall, and appeared to be strongly affected by the overlying dry air; bases were high and the sub-cloud layer seemed to be well-mixed; aerosol gradient was notable with distance downwind; a couple adjacent tracks southwest of OXANA may allow for clear/cloudy contrast |
| 52 | 3/9/2021 | Joint | Statistical Survey | 13:57:41 | 17:16:14 | 4 | 13:55:17 | 17:09:10 | Flew around same area as previous day but this day was more cloud-free to allow for contrast; smoke observed close to land due to local burning; Falcon did some wind calibration work |
| 53 | 3/12/2021 | Joint | Statistical Survey | 12:39:36 | 15:58:13 | 5 | 12:37:25 | 16:01:40 | Smoke sampled over land and by coast |
| 54 | 3/12/2021 | Joint | Statistical Survey | 17:23:19 | 20:52:59 | 5 | 17:19:52 | 20:47:35 | CALIPSO underflight; similar flight plan as morning flight |



| 55 | 3/20/2021 | Joint | Statistical Survey | 12:33:31 | 15:55:44 | 4 | 12:30:58 | 15:53:30 | Interesting layer of depolarizing aerosol right above clouds near the end of flight - possible residual layer of sea salt in dry conditions and/or dust |
| 56 | 3/23/2021 | Joint | Statistical Survey | 15:56:14 | 19:56:54 | 5 | 16:33:50 | 19:51:19 | Falcon delayed takeoff due to ATC issues; Falcon did wind calibration work; relatively clean day with low aerosol and cloud drop number concentrations |
| 57 | 3/29/2021 | Joint | Statistical Survey | 14:53:19 | 18:45:19 | 4 | 14:50:55 | 18:38:00 | ASTER underflight; well defined inversion marking top of clouds; white caps visible most of the flight |
| 58 | 3/30/2021 | Joint | Statistical Survey | 12:01:47 | 15:22:53 | 3 | 11:59:42 | 15:17:14 | Good and consistent cloud conditions; thin aerosol layers above cloud deck |
| 59 | 3/30/2021 | Joint | Statistical Survey | 17:02:08 | 20:38:53 | 5 | 17:04:52 | 20:42:23 | CALIPSO underflight; relatively high absorption aerosol layer on return track; notable cloud boundary which appeared to be collocated with the Gulf Stream with clear sky over the colder water to the north |
| 60 | 4/2/2021 | Joint | Statistical Survey | 12:29:48 | 16:07:44 | 9 | 12:32:40 | 16:01:06 | Cold air outbreak: Deeper cloud structure along track, more precip than usual; sharp offshore $N_d$ gradient |
| 61 | 4/2/2021 | Joint | Statistical Survey | 17:25:18 | 21:07:29 | 9 | 17:29:15 | 21:02:28 | Repeated morning track with similar features; last flight with reduced Falcon payload |
| 62 | 5/13/2021 | Joint | Statistical Survey | 17:06:41 | 20:48:23 | 3 | 17:03:34 | 20:22:58 | Mostly cloud-free; shorter flight than normal; major transition happened across the SST gradient; well-developed cloud line near the edge of the cloudy region. |
| 63 | 5/14/2021 | Joint | Statistical Survey | 12:46:41 | 16:29:30 | 4 | 12:39:53 | 16:16:56 | Complex cloud scene split into two layer maxima with a few clouds developing from the lower layer and connecting to the upper layer which had a more stratiform appearance and appeared to be detraining from the developed cumulus below |
| 64 | 5/14/2021 | Joint | Statistical Survey | 17:49:41 | 21:17:03 | 4 | 17:41:38 | 21:14:15 | Similar conditions to first flight this day. Falcon focused more on lower clouds as the higher clouds were less defined this flight |

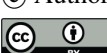



| 65 | 5/15/2021 | Joint | Statistical Survey | 17:43:00 | 21:10:34 | 4 | 17:40:20 | 21:04:18 | Dynamic cloud scene with considerable convection |
| 66 | 5/18/2021 | Joint | Statistical Survey | 15:30:18 | 19:03:09 | 4 | 15:28:14 | 18:54:28 | Conditions similar to RF65; enhanced aerosol farther offshore compared to the coastal (over water) region |
| 67 | 5/19/2021 | Joint | Statistical Survey | 12:31:12 | 15:55:48 | 5 | 12:27:04 | 15:49:56 | Mostly clear air flight |
| 68 | 5/19/2021 | Joint | Statistical Survey | 17:39:33 | 21:04:53 | 4 | 17:30:32 | 20:58:36 | CALIPSO underflight; mostly clear air flight |
| 69 | 5/20/2021 | Joint | Statistical Survey | 14:59:01 | 18:42:18 | 4 | 15:11:23 | 18:27:47 | Smoke aerosol layers observed |
| 70 | 5/21/2021 | Joint | Statistical Survey | 12:27:19 | 16:00:47 | 5 | 12:25:15 | 16:03:35 | Possible cold pool near the turn point; possible smoke/dust aloft; excellent day for cloud water collection with many samples |
| 71 | 5/21/2021 | Joint | Statistical Survey | 17:15:43 | 20:33:33 | 4 | 17:20:08 | 20:42:10 | Large number cloud water samples; in some cases it appeared as the cloud was interacting with the surface as fog |
| 72 | 5/25/2021 | Joint | Statistical Survey | 15:56:59 | 19:19:44 | 4 | 16:00:04 | 19:15:03 | Nothing too notable; Falcon conducted a higher than normal ACT leg during the 3rd cloud ensemble because King Air noted an elevated aerosol by HSRL |
| 73 | 5/26/2021 | Joint | Statistical Survey | 12:37:06 | 15:54:59 | 4 | 12:35:13 | 15:51:26 | Clouds very complicated - it was impossible to follow the standard statistical survey plan; there was at times up to 4 separate layers of cloud and in places there were possible wave clouds which were not constrained to a consistent altitude range |
| 74 | 5/26/2021 | Joint | Statistical Survey | 17:21:20 | 20:31:36 | 4 | 17:17:16 | 20:30:03 | High aerosol variability with especially hazy conditions near land |
| 75 | 6/1/2021 | Joint | Statistical Survey | 14:31:21 | 18:05:48 | 4 | 14:34:00 | 17:57:38 | Shallow cumulus clouds over land on both the outbound and return legs |
| 76 | 6/2/2021 | Joint | Statistical Survey | 12:31:07 | 15:55:10 | 4 | 12:36:32 | 15:47:25 | Considerable convection and precipitation |
| 77 | 6/2/2021 | Joint | Process Study | 17:25:19 | 20:29:11 | 12 | 17:22:55 | 20:41:00 | Excellent summertime cumulus characterization flight; Falcon did ~7 legs in cloud during its wall pattern. |

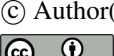



| 78 | 6/5/2021 | Joint | Statistical Survey | 14:09:33 | 17:30:32 | 4 | 14:06:28 | 17:16:50 | Low clouds/fog stayed too low and Falcon couldn't get underneath; good day for data above low cloud tops; interesting AMS organic features noted at low altitude; good candidate for in situ closure analysis for aerosol properties and comparisons with remote sensors |
| 79 | 6/7/2021 | Joint | Statistical Survey | 12:31:53 | 15:59:51 | 4 | 12:28:55 | 15:52:01 | Very shallow MBL noted |
| 80 | 6/7/2021 | Joint | Process Study | 17:37:15 | 20:29:56 | 14 | 17:35:00 | 20:24:32 | Multiple cloud levels probed by Falcon in a wall pattern with high number of cloud water samples |
| 81 | 6/8/2021 | Joint | Statistical Survey | 12:31:27 | 15:46:28 | 4 | 12:28:28 | 15:51:21 | Quick transition from drizzle near coastline to precipitation over the ocean; data suggested higher levels of coarse aerosol than normal |
| 82 | 6/8/2021 | Joint | Statistical Survey | 17:28:09 | 21:02:26 | 4 | 17:31:19 | 20:58:49 | Some aircraft issues made flying typical ensemble legs more challenging |
| 83 | 6/15/2021 | Joint | Statistical Survey | 15:57:36 | 19:10:08 | 4 | 16:03:25 | 19:07:04 | Low clouds were quite variable and did not form in a consistent altitude range with multiple cloud layers at times; clouds at one point were too low to allow Falcon to reach its usual low altitudes |
| 84 | 6/16/2021 | Joint | Statistical Survey | 14:26:35 | 18:09:50 | 5 | 14:29:55 | 17:58:20 | Uniform conditions during the flight; mostly cloud free |
| 85 | 6/17/2021 | Joint | Statistical Survey | 14:30:34 | 17:29:12 | 4 | 14:28:35 | 17:37:00 | ASTER underflight |
| 86 | 6/22/2021 | Joint | Statistical Survey | 12:14:35 | 15:29:04 | 4 | 12:17:12 | 15:31:20 | Shallow MBL with tenuous/small clouds; very hazy due to suspected high humidity and sea salt |
| 87 | 6/24/2021 | Joint | Statistical Survey | 12:23:15 | 15:51:35 | 4 | 12:20:52 | 15:37:15 | Clouds included significant stratiform cloud connected to embedded cumulus; widespread precipitation both in the sub-cloud environment and observed aloft originating from detraining layers; extensive precipitation challenged the ability to achieve sub-cloud aerosol sampling in many locations |
| 88 | 6/26/2021 | Joint | Statistical Survey | 12:28:49 | 15:53:57 | 4 | 12:33:25 | 15:48:45 | Subtropical high conditions; low aerosol concentrations noted |



| 89 | 6/26/2021 | Joint | Statistical Survey | 17:25:01 | 20:49:35 | 5 | 17:20:51 | 20:42:23 | Flight originally planned to be process study but changed to stat survey since targets did not build as desired; decent shallow cumulus sampling |
| 90 | 6/28/2021 | Joint | Statistical Survey | 12:28:31 | 15:43:55 | 4 | 12:31:10 | 15:45:57 | Mostly shallow cumulus with some developed regions that appeared to be organized as convergence lines/streets |
| 91 | 6/29/2021 | Joint | Statistical Survey | 12:16:58 | 15:34:41 | 4 | 12:19:55 | 15:36:59 | Very similar conditions as RF90 |
| 92 | 6/30/2021 | Joint | Statistical Survey | 12:21:16 | 15:40:27 | 4 | 12:23:54 | 15:41:41 | Relatively low aerosol concentrations; patchy cumulus clouds |
| 93 | 6/30/2021 | Joint | Statistical Survey | 17:09:17 | 20:30:05 | 5 | 17:13:33 | 20:33:48 | Similar conditions as morning flight (RF92); crossed over a large discrete cloud clearing east of ZIBUT |
| 94 | 11/30/2021 | Joint | Statistical Survey | 16:23:37 | 19:53:32 | 4 | 16:17:54 | 19:34:39 | ATC issues kept Falcon higher than desired at times; well-defined boundary layer with energetic/mixed sub-cloud layer |
| 95 | 12/1/2021 | Joint | Statistical Survey | 15:23:20 | 18:54:36 | 4 | 15:20:40 | 18:45:40 | Similar conditions to RF94; cloud bases were high again with a deep well mixed sub-cloud layer; smoke in boundary layer near coast |
| 96 | 12/7/2021 | Joint | Statistical Survey | 16:58:05 | 20:28:35 | 4 | 16:55:46 | 20:17:52 | Complex cloud scene split into two layer maxima with a few clouds developing from the lower layer and connecting to the upper layer which had a more stratiform appearance and appeared to be detraining from the developed cumulus below |
| 97 | 12/9/2021 | Joint | Statistical Survey | 12:47:48 | 16:12:26 | 5 | 12:52:54 | 15:54:40 | Landed at Quonset State Airport; nice cloud conditions with transitions between open/closed cells; aerosol gradient during flight |
| 98 | 12/9/2021 | Joint | Statistical Survey | 17:25:23 | 20:55:22 | 6 | 17:28:54 | 20:36:05 | Return to LaRC from Quonset State Airport; similar conditions as RF97 in morning |
| 99 | 12/10/2021 | Joint | Statistical Survey | 17:49:41 | 21:04:36 | 4 | 17:47:11 | 21:00:38 | Military traffic during this flight prevented Falcon from doing most of its typical above cloud top (ACT) legs |
| 100 | 1/11/2022 | Joint | Statistical Survey | 13:35:19 | 17:08:18 | 7 | 13:42:50 | 16:57:58 | Cold air outbreak; did upwind work in clear air along with cloud work; P3 from IMPACTS |





| | | | | | | | | | |
|---|---|---|---|---|---|---|---|---|---|
| | | | | | | | | | mission flew in general vicinity this flight day |
| 101 | 1/11/2022 | Joint | Statistical Survey | 18:34:09 | 22:05:19 | 6 | 18:38:34 | 21:47:02 | Cold air outbreak; icing was more of an issue for Falcon this second flight of the day leading to more MinAlt flying to de-ice |
| 102 | 1/12/2022 | Joint | Statistical Survey | 13:22:05 | 16:38:28 | 4 | 13:20:05 | 16:31:22 | Marked gradient in drop number concentration along flight track that appeared to correlate with an increase in the prevalence of precipitating cells |
| 103 | 1/12/2022 | Joint | Statistical Survey | 18:00:03 | 21:18:49 | 5 | 17:58:25 | 21:13:33 | CALIPSO underflight; similar conditions to morning flight (RF102) |
| 104 | 1/15/2022 | Joint | Statistical Survey | 12:56:34 | 16:36:53 | 6 | 12:50:36 | 16:29:28 | Clouds thickened substantially from near overcast at ZIBUT with ice and liquid precip observed to the east and subsequent breakup of the overcast to broken but deeper cells |
| 105 | 1/18/2022 | Joint | Statistical Survey | 13:17:57 | 16:55:03 | 8 | 13:24:32 | 16:36:33 | Cold air outbreak; did upwind work in clear air along with cloud work (similar to RF100) |
| 106 | 1/18/2022 | Joint | Statistical Survey | 18:32:53 | 22:21:00 | 5 | 18:31:15 | 21:54:40 | Cold air outbreak; similar to RF101 where the second flight of the day continues sampling the cloud field probed in the morning flight; light precip widespread but with stronger showers associated with cores; strong $N_d$ gradient |
| 107 | 1/19/2022 | Joint | Statistical Survey | 13:14:08 | 16:40:51 | 4 | 13:19:53 | 16:34:10 | Complex cloud scene with multiple cloud layers at times |
| 108 | 1/19/2022 | Joint | Statistical Survey | 18:35:06 | 21:59:37 | 4 | 18:41:04 | 21:52:52 | Similar conditions as morning flight (RF107) |
| 109 | 1/24/2022 | Joint | Statistical Survey | 13:38:57 | 17:01:11 | 4 | 13:34:18 | 16:45:18 | Sharp gradient in MBL height offshore especially once over warmer water where it rapidly deepened and was topped with small cumulus-like clouds |
| 110 | 1/24/2022 | Joint | Statistical Survey | 18:15:53 | 21:39:35 | 4 | 18:21:33 | 21:29:36 | Similar conditions as morning flight (RF109) |
| 111 | 1/26/2022 | Joint | Statistical Survey | 13:10:52 | 16:51:45 | 4 | 12:56:10 | 16:28:48 | Multiple cloud layers; aerosol layer above cloud at times; interesting AMS organic structure noted |



| 112 | 1/26/2022 | Joint | Statistical Survey | 18:07:54 | 21:45:56 | 3 | 18:05:39 | 21:24:00 | Markedly different conditions observed above cloud top during this flight compared to morning flight; dryer conditions in the lower free troposphere than the morning |
| 113 | 1/27/2022 | Joint | Statistical Survey | 12:54:53 | 15:58:18 | 4 | 12:57:30 | 15:50:45 | Landed at Providence Airport; very dry above cloud; considerable icing for Falcon during flight; decoupled layers noted |
| 114 | 1/27/2022 | Joint | Statistical Survey | 17:32:31 | 20:58:31 | 4 | 17:34:28 | 20:43:00 | Return to LaRC from Providence; cloud scene became even more complex than morning with more evidence of decoupling of the upper part of the cloud layer with sometimes 3 distinct strata; ice imagery data from 2D-S showed differences with morning flight |
| 115 | 2/1/2022 | Joint | Statistical Survey | 13:22:28 | 16:40:01 | 4 | 13:24:43 | 16:31:43 | Aerosol gradient observed; thicker regions of the clouds were precipitating and in some regions it was quite significant with visible showers below cloud base |
| 116 | 2/2/2022 | Joint | Statistical Survey | 18:19:17 | 21:59:02 | 4 | 18:26:40 | 21:50:00 | Mix of shallow cumulus with some deeper cells with showers and a possible cold pool crossing; MBL had decoupled structure |
| 117 | 2/3/2022 | Joint | Statistical Survey | 13:25:51 | 16:43:35 | 4 | 13:23:47 | 16:34:23 | Sub-cloud environment was warmer and more humid than normal |
| 118 | 2/3/2022 | Joint | Statistical Survey | 18:10:48 | 21:24:52 | 4 | 18:08:29 | 21:28:10 | Similar conditions as morning flight (RF117) |
| 119 | 2/5/2022 | Joint | Statistical Survey | 13:44:32 | 17:05:26 | 3 | 13:42:26 | 16:58:58 | Characterized the initial stages of the post-frontal environment as it advects offshore; a 2nd flight this day was planned but scrubbed due to maintenance issue |
| 120 | 2/15/2022 | Joint | Statistical Survey | 13:34:04 | 17:06:08 | 4 | 13:31:40 | 16:48:02 | Cumulus feeding an upper stratiform layer near the inversion; in thicker cloud regions, some mixed phase and precipitation observed with sub-cloud drizzle below the melting level; elevated aerosol by coast |



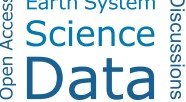

| 121 | 2/15/2022 | Joint | Statistical Survey | 18:26:24 | 22:22:17 | 3 | 18:07:41 | 22:03:21 | Similar conditions as morning flight (RF120) |
| 122 | 2/16/2022 | Joint | Statistical Survey | 13:25:05 | 16:50:49 | 3 | 13:22:18 | 16:31:40 | Clouds had the appearance of an overcast near the inversion with cumulus feeding from below; sulfate-rich aerosol |
| 123 | 2/16/2022 | Joint | Statistical Survey | 18:24:32 | 22:03:02 | 3 | 18:28:10 | 21:59:34 | Complex cloud and boundary layer structure; moisture profile near coast suggested marine air was previously lofted and then had become disconnected from the surface; $N_d$ gradient offshore |
| 124 | 2/19/2022 | Joint | Statistical Survey | 13:32:00 | 17:25:52 | 2 | 13:51:21 | 17:07:23 | Multiple cloud layers; airspace restrictions (rocket launch from Wallops) affected areas we could fly |
| 125 | 2/19/2022 | Joint | Statistical Survey | 18:36:30 | 22:06:48 | 3 | 18:34:55 | 22:01:19 | Continued airspace restrictions; irregularly shaped particles detected by 2D-S |
| 126 | 2/22/2022 | Joint | Statistical Survey | 13:58:48 | 17:15:43 | 3 | 13:34:25 | 16:55:03 | Falcon ascended higher than normal at times to sample an aerosol layer aloft flagged by HSRL-2 |
| 127 | 2/22/2022 | Joint | Statistical Survey | 18:43:33 | 22:16:25 | 3 | 18:41:10 | 21:59:38 | Areas sampled with relatively low aerosol/cloud number concentrations |
| 128 | 2/26/2022 | Joint | Statistical Survey | 13:23:33 | 16:24:30 | 4 | 13:18:30 | 16:03:13 | Landed at Providence Airport; extensive low cloud under a dense high cloud deck for most of the flight |
| 129 | 2/26/2022 | Single-Falcon | Statistical Survey | 20:56:17 | 22:59:23 | 0 | 18:13:41 | 20:52:34 | Return to LaRC from Providence; similar conditions as morning flight; due to a maintenance issue with King Air it flew back but could not collect data |
| 130 | 3/2/2022 | Joint | Statistical Survey | 19:10:25 | 22:53:14 | 4 | 19:08:19 | 22:29:10 | Unicorn aerosol module; aerosol enhancements above boundary layer |
| 131 | 3/3/2022 | Joint | Statistical Survey | 13:32:56 | 16:58:32 | 3 | 13:30:32 | 16:52:08 | Unicorn aerosol module; similar to RF130 there was relatively high AOD for the winter season with interesting aerosol structure throughout flight |
| 132 | 3/3/2022 | Joint | Statistical Survey | 18:32:07 | 21:52:14 | 3 | 18:27:27 | 21:42:40 | Sampled different airmasses during flight |
| 133 | 3/4/2022 | Joint | Statistical Survey | 13:45:14 | 17:28:27 | 4 | 13:43:00 | 17:03:22 | At the far turnpoint we crossed the convergence line that was flown the previous day |





| 134 | 3/4/2022 | Joint | Statistical Survey | 18:42:03 | 22:22:29 | 3 | 18:32:00 | 21:54:27 | Markedly different conditions from the morning flight and a good contrast case for two flights on same day |
| 135 | 3/7/2022 | Joint | Statistical Survey | 13:28:48 | 16:51:59 | 3 | 13:25:44 | 16:44:18 | On the way out, high aerosol loading above boundary layer with areas of elevated aerosol depolarization near the top of the residual layer |
| 136 | 3/7/2022 | Single-Falcon | Statistical Survey | N/A | N/A | 0 | 18:39:20 | 21:57:41 | King Air experienced maintenance issue prior to take off and was grounded; similar conditions to morning flight for Falcon |
| 137 | 3/13/2022 | Joint | Process Study | 12:28:41 | 16:24:46 | 11 | 12:35:23 | 16:14:50 | Excellent cold air outbreak day with marine boundary layer winds westerly/northwesterly and a 'transition' (from solid to open cloud field) within reach; Falcon conducted mini "walls" upwind, at, and downwind of the transition zone; steam fog observed |
| 138 | 3/13/2022 | Joint | Statistical Survey | 17:32:47 | 21:22:10 | 3 | 17:36:37 | 20:48:16 | Extending the line from morning flight farther upwind to characterize clear air |
| 139 | 3/14/2022 | Joint | Statistical Survey | 12:32:35 | 15:52:52 | 3 | 12:35:48 | 15:45:45 | Clouds had a decoupled appearance with small cumulus topping a deep mixed layer with some cumulus developing up to a more extensive stratiform near the inversion; drizzle observed; generally clean aerosol conditions this flight |
| 140 | 3/14/2022 | Joint | Statistical Survey | 17:22:26 | 20:49:25 | 3 | 17:26:15 | 20:44:46 | Similar conditions to RF139; smoke plume emanating from a woodland fire sampled on the inbound leg over North Carolina |
| 141 | 3/18/2022 | Joint | Statistical Survey | 14:55:12 | 18:15:47 | 3 | 14:48:07 | 17:59:00 | Lots of fog in the morning that prevented an earlier flight; clouds were sometimes too low to get under |
| 142 | 3/22/2022 | Joint | Statistical Survey | 12:50:47 | 15:23:47 | 3 | 12:45:45 | 15:25:58 | First flight to Bermuda; mostly cloud-free and indications of aerosol gradient offshore towards Bermuda |
| 143 | 3/22/2022 | Joint | Statistical Survey | 17:12:14 | 21:00:01 | 4 | 17:36:21 | 21:12:02 | Return from Bermuda to LaRC; owing to lack of a functional power cart at Bermuda, some Falcon instruments needed extra time to stabilize to collect good data this flight |



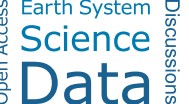

| 144 | 3/26/2022 | Joint | Statistical Survey | 12:14:27 | 16:01:09 | 3 | 12:30:09 | 16:12:35 | Dust, smoke, possibly pollen; unicorn aerosol module |
|---|---|---|---|---|---|---|---|---|---|
| 145 | 3/26/2022 | Joint | Statistical Survey | 17:22:48 | 21:20:22 | 3 | 17:31:10 | 21:23:49 | Similar aerosol conditions as RF145 but with higher cloud coverage |
| 146 | 3/28/2022 | Joint | Statistical Survey | 16:52:05 | 20:49:49 | 4 | 16:49:41 | 20:19:50 | Nothing too noteworthy documented other than it being a good data for added statistics |
| 147 | 3/29/2022 | Joint | Statistical Survey | 12:41:46 | 16:34:31 | 4 | 12:34:53 | 16:21:04 | Excellent cold air outbreak day; flew counterclockwise partly to help with aircraft coordination on the most important leg aligned with the boundary layer winds; did upwind aerosol characterization and cloud work |
| 148 | 3/29/2022 | Joint | Process Study | 17:48:08 | 21:26:17 | 4 | 17:44:42 | 21:33:17 | Similar conditions to morning flight; Falcon conducted mini "walls" like RF137 |
| 149 | 5/3/2022 | Joint | Statistical Survey | 13:45:00 | 16:56:25 | 4 | 13:48:45 | 16:51:01 | Convective data with relatively high AOD and smoke aerosol (possibly from New Mexico area) |
| 150 | 5/5/2022 | Joint | Statistical Survey | 12:27:06 | 15:46:26 | 4 | 12:23:27 | 15:41:20 | Landed at Providence Airport; high number of cloud water samples collected as unbroken long sampling times in cloud were achieved |
| 151 | 5/5/2022 | Joint | Statistical Survey | 17:10:28 | 20:40:49 | 4 | 17:14:06 | 20:30:32 | Return to LaRC from Providence; similar to morning flight but with less extensive cloud coverage |
| 152 | 5/10/2022 | Joint | Statistical Survey | 12:31:00 | 15:55:21 | 4 | 12:34:05 | 15:52:00 | Pronounced 'pure' sea salt aerosol case; hard to get below clouds at times as they were low; drizzle was frequent |
| 153 | 5/16/2022 | Joint | Statistical Survey | 12:21:28 | 15:40:39 | 4 | 12:24:44 | 15:37:17 | Nothing too noteworthy documented other than it being a good data for added statistics |
| 154 | 5/16/2022 | Joint | Statistical Survey | 17:11:51 | 20:38:43 | 4 | 17:15:35 | 20:29:09 | Convective weather led to some flight deviations this flight |
| 155 | 5/17/2022 | Joint | Statistical Survey | 14:04:10 | 17:32:00 | 3 | 13:50:37 | 17:00:08 | Unicorn aerosol module |
| 156 | 5/18/2022 | Joint | Statistical Survey | 12:27:10 | 15:25:35 | 4 | 12:25:31 | 15:28:34 | Flight to Bermuda; offshore gradient in aerosol parameters |
| 157 | 5/18/2022 | Joint | Statistical Survey | 17:02:45 | 21:12:33 | 4 | 17:25:45 | 20:55:33 | Return from Bermuda to Langley; CALIPSO underflight; |





|  |  |  |  |  |  |  |  |  |  |
|---|---|---|---|---|---|---|---|---|---|
|  |  |  |  |  |  |  |  |  | possible indications of bioaerosol |
| 158 | 5/20/2022 | Joint | Statistical Survey | 13:33:43 | 16:55:37 | 4 | 13:38:25 | 16:58:14 | Hazy day with indications of bioaerosol and multiple layers of aerosol |
| 159 | 5/21/2022 | Joint | Statistical Survey | 12:09:49 | 15:14:00 | 5 | 12:13:30 | 15:06:39 | To Bermuda |
| 160 | 5/21/2022 | Joint | Statistical Survey | 16:51:03 | 20:30:27 | 5 | 17:07:18 | 20:19:46 | Return from Bermuda to Langley; CALIPSO underflight |
| 161 | 5/31/2022 | Joint | Statistical Survey | 12:33:39 | 16:09:35 | 3 | 12:36:07 | 15:56:16 | Transit to Bermuda for 3-week deployment based in Bermuda |
| 162 | 6/2/2022 | Single-Falcon | Statistical Survey | N/A | N/A | 0 | 11:19:14 | 14:19:17 | King Air experienced maintenance issue prior to take off; Tudor Hill spiral |
| 163 | 6/2/2022 | Single-Falcon | Process Study | N/A | N/A | 0 | 16:03:00 | 19:01:26 | Falcon conducted wall patterns in both cloud and cloud-free air; Tudor Hill spiral |
| 164 | 6/3/2022 | Single-Falcon | Statistical Survey | N/A | N/A | 0 | 12:48:53 | 15:10:51 | Flight cut short as Falcon was needed to assist with King Air maintenance issue |
| 165 | 6/5/2022 | Joint | Statistical Survey | 11:02:20 | 14:26:12 | 4 | 11:08:21 | 14:20:20 | Flight executed early to avoid an approaching tropical storm |
| 166 | 6/7/2022 | Joint | Statistical Survey | 11:17:40 | 15:00:14 | 5 | 11:38:43 | 15:02:09 | Overpass of BIOS underwater glider; Tudor Hill spiral |
| 167 | 6/7/2022 | Joint | Statistical Survey | 15:57:31 | 19:28:19 | 5 | 16:14:20 | 19:33:24 | Uniform HSRL-2 data curtains for aerosol during flight; free troposphere mostly clean; Tudor Hill spiral |
| 168 | 6/8/2022 | Joint | Statistical Survey | 12:56:12 | 16:14:14 | 5 | 13:12:41 | 16:08:58 | ASTER underflight; fairly clean again in free troposphere like previous flight |
| 169 | 6/8/2022 | Joint | Statistical Survey | 17:13:56 | 20:53:50 | 5 | 17:32:12 | 20:56:22 | Tudor Hill spiral |
| 170 | 6/10/2022 | Joint | Statistical Survey | 11:57:01 | 15:35:19 | 7 | 12:20:04 | 15:37:27 | ASTER underflight; possible African dust; Tudor Hill spiral |
| 171 | 6/10/2022 | Joint | Process Study | 17:08:55 | 21:13:31 | 16 | 17:30:18 | 20:51:35 | Exceptional flight (one of the best) in that two adjacent Falcon walls were conducted with contrasts in cloud development along with varying degrees of dust influence |
| 172 | 6/11/2022 | Joint | Statistical Survey | 12:00:01 | 13:55:07 | 4 | 12:24:00 | 16:00:54 | Continued influence of what seems to be African dust; Tudor Hill spiral |
| 173 | 6/11/2022 | Joint | Process Study | 17:09:36 | 20:55:48 | 23 | 17:24:10 | 20:45:27 | More African dust; record number of dropsondes for an |

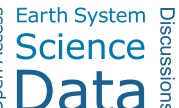

| | | | | | | | | | |
|---|---|---|---|---|---|---|---|---|---|
| | | | | | | | | | ACTIVATE flight (23); excellent wall profiles of 2 cloud systems |
| 174 | 6/13/2022 | Joint | Statistical Survey | 11:15:17 | 14:55:27 | 3 | 11:43:05 | 14:59:05 | Got into cleaner air farther removed from dust to allow for contrasting; Tudor Hill spiral |
| 175 | 6/13/2022 | Joint | Statistical Survey | 16:26:06 | 19:59:59 | 5 | 16:49:10 | 20:16:30 | CALIPSO underflight; Tudor Hill spiral |
| 176 | 6/14/2022 | Joint | Process Study | 12:59:24 | 16:47:39 | 5 | 13:28:57 | 16:44:12 | Dust influence again; Falcon conducted another wall pattern with high number of legs at different altitude in the cloud system |
| 177 | 6/16/2022 | Single-King Air | Statistical Survey | 10:59:45 | 12:51:24 | 3 | N/A | N/A | Falcon experienced maintenance issue prior to take off and stayed on ground |
| 178 | 6/17/2022 | Joint | Statistical Survey | 12:57:16 | 16:47:22 | 8 | 13:25:31 | 16:57:04 | Tudor Hill spiral |
| 179 | 6/18/2022 | Joint | Statistical Survey | 11:56:10 | 15:37:35 | 5 | 12:05:15 | 15:23:37 | Return from Bermuda; some flight deviations needed to account for thunderstorm activity |




Table 3. Summary of King Air instrumentation and measurements. [§]Uncertainties, which represent a combination of measurement precision and accuracy, are presented for typical measurement conditions. *"$x$ m / $y$ m" indicates $x$-m vertical resolution and $y$-m horizontal resolution along track. [†]Cross-track by along-track. [‡]Non-imaging: along-track product with single cross-track elements for RSP.

| Instrument and Relation to Objectives | Measured/Retrieved Parameter | Resolution | Uncertainty[§] | Reference/Notes |
|---|---|---|---|---|
| HSRL-2 (aerosol and cloud properties; prototype of possible satellite aerosol-cloud lidar retrievals) | Particulate Backscatter Profiles (355, 532, and 1064 nm) | 30 m x 1 km* | 0.2 Mm⁻¹sr⁻¹ | Hair et al., 2008; Burton et al., 2015; Burton et al., 2018 |
| | Particulate Depolarization (355, 532, and 1064 nm) | 30 m x 1 km* | ~ 2-5 % | See Burton et al. (2015) for details regarding aerosol depolarization uncertainties; uncertainty values are approximate and dependent on scattering levels |
| | Particulate Extinction Profiles (355 and 532 nm) | 225 m x 6 km* | 0.01 km⁻¹ | |
| | Particulate Lidar Ratio (355 and 532 nm) | 225 m x 6 km* | ~10 % | Uncertainty values are approximate and dependent on scattering levels |
| | Ångstrom Exponent - Extinction (532/355 nm) | 225 m x 6 km* | ~10 % | Uncertainty values are approximate and dependent on scattering levels |
| | Ångstrom Exponent - Backscatter (532/355 nm, 1064/532 nm) | 30 m x 1 km* | ~10 % | Uncertainty values are approximate and dependent on scattering levels |
| | Aerosol Optical Depth (355, 532 nm) | | | |
| |    1-D Full Column (Aircraft-to-Surface) | Integrated product x 6 km* | 0.02 | |
| |    2-D Vertically Resolved (Altitude-Bin-to-Surface) | 30 m x 6 km* | ≤ 0.02 | |
| | Mixed Layer Height | 15 m x 1 km* | ~100 m | Scarino et al., 2014 |
| | Aerosol Type (Qualitative) | 135 m x 6 km* | N/A | Burton et al., 2012 |
| | Surface Wind Speed (10 m) | 1.25 m x 1 km* | 0.16 m s⁻¹ (± 1.94 m s⁻¹) | Dmitrovic et al., forthcoming |
| | Cloud Top Height (1-D) | 1.25 m x 50 m* | ~ 5 m | Hair et al., forthcoming; Cloud top height uncertainties are approximate and based upon a threshold of the backscatter |
| | Cloud Top Extinction | 1.25 m x 50 m* | < 20 % | Still being evaluated; assumes liquid-phase only clouds |
| | Cloud Top Lidar Ratio (extinction-to-backscatter) | Integrated product x 50 m* | < 20 % | Still being evaluated; assumes liquid-phase only clouds |
| | 10 m Ocean Subsurface Particulate Backscatter (532 nm) | N/A x 1 km* | < 10% | Schulien et al., 2017; Only available for select flights |
| RSP (aerosol and cloud properties; development of combined lidar-polarimeter aerosol-cloud retrievals) | Aerosol Optical Depth for each mode of a bimodal distribution (column) | 100 m x 600 m[†] | 0.02/7% | Stamnes et al., 2018 |
| | Aerosol Size: effective radius (column) | 100 m x 600 m[†,‡] | 0.05 μm/10% | Stamnes et al., 2018 |
| | Aerosol Size: effective variance (column) | 100 m x 600 m[†,‡] | 0.3/50% | Stamnes et al., 2018 |
| | Aerosol Single Scatter Albedo (column) | 100 m x 4 km[†,‡] | 0.03 | Stamnes et al., 2018 |
| | Aerosol Refractive Index (column) | 100 m x 4 km[†,‡] | 0.02 | Stamnes et al., 2018 |
| | Aerosol Particle Number Concentration | 100 m x 4 km[†,‡] | 10-70% | Schlosser et al., 2022 |
| | Aerosol Top Height | 100 m x 4 km[†,‡] | < 1 km | Wu et al., 2016 |



| | | | | |
|---|---|---|---|---|
| | Surface Wind Speed | 100 m x 4 km[†,‡] | 0.5 m s-1 | Stamnes et al., 2018 |
| | Chlorophyll A Concentration | 100 m x 4 km[†,‡] | 0.7 mg m-3 | Stamnes et al., 2018 |
| | Ocean diffuse attenuation coefficient | 100 m x 4 km[†,‡] | 40% | Stamnes et al., 2018 |
| | Ocean hemispherical backscatter coefficient | 100 m x 4 km[†,‡] | 10% | Stamnes et al., 2018 |
| | Cloud Flag/Test | 100 m x 100 m[†,‡] | 10% | Comparisons with HSRL-2 cloud detection |
| | Cloud Top Phase Index | 100 m x 600 m[†,‡] | 10% | Van Diedenhoven et al., 2012 |
| | Cloud Top Effective Radius | 100 m x 600 m[†,‡] | 1 µm/10% | Alexandrov et al., 2012a/b |
| | Cloud Top Effective Variance | 100 m x 600 m[†,‡] | 0.05/50% | Alexandrov et al., 2012a/b |
| | Cloud Mean Effective Radius | 100 m x 600 m[†,‡] | 20% | Alexandrov et al., 2012a/b |
| | Cloud Optical Depth | 100 m x 600 m[†,‡] | 10% | Nakajima and King, 1990 |
| | Liquid Water Path | 100 m x 600 m[†,‡] | 25% | Uncertainties for optical depth and effective radius added in quadrature |
| | Columnar Water Vapor (Above Surface or Cloud) | 100 m x 600 m[†,‡] | 10% | Nielsen et al., forthcoming |
| | Cloud Top Height | 100 m x 600 m[†,‡] | 15% | Sinclair et al., 2017 |
| | Cloud Droplet Number Concentration | 100 m x 600 m[†,‡] | 25% | Sinclair et al., 2019 |
| | Cloud Albedo | 100 m x 600 m[†,‡] | 10% | Radiometric accuracy of 5% |
| Vaisala NRD41 Dropsonde (meteorological state) | Latitude/Longitude | | NA | |
| | Altitude | | NA | |
| | GPS Altitude | | NA | |
| | Pressure | | 0.5 hPa | |
| | Temperature | /~11 m | 0.2°C | Vömel et al., 2021; Vömel et al., forthcoming |
| | Dew Point Temperature | | | |
| | Relative Humidity | | 3% | |
| | Horizontal Wind (u and v components) | | 0.5 m s$^{-1}$ | |
| | Vertical Wind | | 1 m s$^{-1}$ | |
| Applanix 610 (Navigational) | Day and Time | 1 s | NA | |
| | Latitude/Longitude | 1 s | 1.5 m/1.5 m | |
| | GPS Altitude | 1 s | 3 m | |
| | Ground Speed | 1 s | 0.03 m s$^{-1}$ | |
| | Vertical Speed | 1 s | 3 m s$^{-1}$ | |
| | True Heading | 1 s | 0.03° | |
| | Track Angle | 1 s | 0.03° | |
| | Drift Angle | 1 s | NA | |
| | Pitch Angle | 1 s | 0.005° | |
| | Roll Angle | 1 s | 0.005° | |


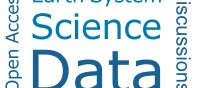

Table 4. Summary of camera details on the King Air and HU-25 Falcon. The first column represents the research flight number for which a certain set of cameras were installed to replace pre-existing ones with the same swap-out dates for the nadir and forward cameras. HFOV = horizontal field of view. The time resolution of the cameras was 1-2 seconds.


| | King Air - Nadir Camera | | | | | | King Air & HU-25 Falcon - Forward Camera | | | | | |
|---|---|---|---|---|---|---|---|---|---|---|---|---|
| RF | Make | Model | Lens | HFOV | Focal Length | Aperture | Make | Model | Lens | HFOV | Focal Length | Aperture |
| 1 | Garmin | VIRB Ultra 30 | None | 62 | N/A | N/A | GoPro | Hero 6 Black | None | N/A | N/A | N/A |
| 41 | Garmin | VIRB Ultra 30 | None | 62 | N/A | N/A | Axis | F-1005-E | None | 113 | 2.8 mm | 2 |
| 62 | Axis | F-1005-E | None | 113 | 2.8 mm | 2 | Axis | F-1005-E | None | 113 | 2.8 mm | 2 |
| 100 | Axis | F-1005-E | M12 16mm F1.8 | 22 | 16 mm | 1.8 | Axis | F-1005-E | None | 113 | 2.8 mm | 2 |
| 149 | Axis | F-1005-E | M12 6mm F1.9 | 56 | 6 mm | 1.9 | Axis | F-1005-E | None | 113 | 2.8 mm | 2 |



Table 5. Summary of HU-25 Falcon instrumentation and measurements.

| Instrument | Measured Parameter | Uncertainty | Size Range (µm) | Time Resolution (s) | Reference/Notes |
|---|---|---|---|---|---|
| | | Aerosol Particles | | | |
| BMI Counterflow Virtual Impactor vs. Isokinetic Inlet | Inlet Flag | NA | NA | 1 | |
| TSI-3776 Condensation Particle Counter (CPC) | Particle Concentration | 10% | 0.003 - 5 | 1 | Moore et al., 2017 |
| TSI-3772 CPC | Particle Concentration | 10% | 0.01 - 5 | 1 | Moore et al., 2017 |
| TSI-3772 with Thermal Denuder (350° C) | Nonvolatile (350°C) Particle Concentration | 10% | 0.01 - 5 | 1 | Moore et al., 2017 |
| TSI Scanning Mobility Particle Sizer (SMPS); Model 3085 DMA, Model 3776 CPC, and Model 3088 Neutralizer | Total and Nonvolatile Dry Aerosol Size Distributions | 20% | 0.003–0.1 | 45 | Moore et al., 2017 |
| TSI-3340 Laser Aerosol Spectrometer (LAS) | | 20% | 0.1–5 | 1 | Froyd et al., 2019 |
| TSI-3563 Nephelometer | Dry Scattering Coefficient (450, 550, and 700 nm) | 20% | <1 (2021-2022), < 5 (2020) | 1 | Ziemba et al., 2013 |
| TSI-3563 Nephelometer with 80% humidification | f(RH) for Scattering (450, 550, and 700 nm) | 20% | <1 (2021-2022), < 5 (2020) | 1 | Ziemba et al., 2013 |
| Radiance Research Particle Soot Absorption Photometer (PSAP) | Aerosol Absorption (470, 532, and 660 nm) | 15% | <5 | 1 | Mason et al., 2018 |
| Aerodyne HR-ToF-AMS | Non-refractory Chemically Resolved Mass Concentration | <50% | 0.06-0.6 | 25 | DeCarlo et al., 2008 |
| DMT Cloud Condensation Nuclei (CCN) spectrometer | CCN Concentration and Spectra | 10% 0.04 % SS | <5 | 1 | Moore et al., 2009 |
| BMI PILS Coupled to Offline Ion Chromatography | Water-Soluble Aerosol Chemical Composition | <20% (species dependent) | <5 | 300-420 | Sorooshian et al., 2006 |
| | | Clouds | | | |
| DMT Cloud Droplet Probe (CDP) | Aerosol and Cloud Droplet Number Concentration, Liquid Water Content, Effective Radius/Variance | 20% | 2-50 | 1 | Lance et al., 2012 |
| DMT Cloud and Aerosol Spectrometer (CAS) | Aerosol and Cloud Droplet Number Concentration, Liquid Water Content, Effective Radius/Variance | 20% | 0.5-50 | 1 | Baumgardner et al., 2001; Lance et al., 2012 |
| SPEC Inc. Fast Cloud Droplet Probe (FCDP) | Aerosol and Cloud Droplet Number Size Distribution, Liquid Water Content, Effective Diameter, Median Volume Diameter | 15-50% | 3-50 | 1 | Kirschler et al., 2022 |



| | | | | | |
|---|---|---|---|---|---|
| SPEC Inc. Two-Dimensional Stereo (2D-S) Vertical-Arm | Cloud Number Size Distribution for Liquid/Ice/Total, Liquid and Ice Water Content, Ice Flag, Effective Diameter for Liquid/Ice/Total, Median Volume Diameter for Liquid and Total | 15-60% | 29–1465 | 1 | Kirschler et al., in prep |
| SPEC Inc. Two-Dimensional Stereo (2D-S) Horizontal-Arm | same as 2D-S Vertical Arm | 15-60% | 29–1465 | 1 | Kirschler et al., in prep |
| AC3 and offline chemistry | Cloud Water Chemical Composition | <20% (species dependent) | >8 (droplet diameter) | Function of cloud LWC | Crosbie et al., 2018 |
| Meteorological State Parameters and Trace Gases | | | | | |
| Applanix 610 (Navigational) | Day and Time | NA | N/A | 1/0.05 | |
| | Latitude/Longitude | 1.5 m/1.5 m | N/A | 1/0.05 | |
| | GPS Altitude | 3 m | N/A | 1/0.05 | |
| | Pressure Altitude | 3 m | N/A | 1/0.05 | |
| | Ground Speed | 0.03 m s$^{-1}$ | N/A | 1/0.05 | |
| | Vertical Speed | 3 m s$^{-1}$ | N/A | 1/0.05 | |
| | True Heading | 0.03° | N/A | 1/0.05 | |
| | True Air Speed | 5% | N/A | 1/0.05 | |
| | Track Angle | 0.03° | N/A | 1/0.05 | |
| | Drift Angle | NA | N/A | 1/0.05 | |
| | Pitch Angle | 0.005° | N/A | 1/0.05 | |
| | Roll Angle | 0.005° | N/A | 1/0.05 | |
| 5-port pressure system (TAMMS) | 3-D Winds | w: 10 cm/s | N/A | 0.05 | Thornhill et al., 2003 |
| | | u,v: 50 cm/s | N/A | | |
| Rosemount 102 Sensor | Temperature | 0.5°C | N/A | 0.05 | |
| Heitronics KT-15 Infrared Thermometer | Infrared Surface Temperature | 5% | N/A | 1 s | |
| Diode Laser Hygrometer (DLH) | Water Vapor | 5% or 0.1 ppmv | N/A | <0.05 | Diskin et al., 2002 |
| Picarro model G2401-m | CO, $CO_2$, $CH_4$ | 5 ppb (CO) | N/A | 2.5 | DiGangi et al., 2021 |
| | | 0.1 ppm ($CO_2$) | | 2.5 | |
| | | 1 ppb ($CH_4$) | | 2.5 | |
| 2B Tech. Inc. model 205 | $O_3$ | 6 ppb | N/A | 2 | Wei et al., 2021 |




Table 6. MERRA-2 data fields sampled along the Falcon flight tracks during ACTIVATE (see sect. 5.5). STP = standard temperature (0°C) and pressure (1013.25 hPa).

| Variable Name | Unit | Field |
|---|---|---|
| Time_Stop | seconds | Number of seconds from 00:00 UTC |
| Lat_flight | deg | Latitude |
| lon_flight | deg | Longitude |
| press_flight | hPa | Pressure calculated from aircraft pressure altitude |
| M2_CO | ppbv | Carbon monoxide volume mixing ratio |
| M2_O3 | ppbv | Ozone volume mixing ratio |
| M2_DMS | ppbv | Dimethylsulphide volume mixing ratio |
| M2_SO2 | ppbv | Sulphur dioxide volume mixing ratio |
| M2_MSA | $\mu g.m^{-3}$ | Methanesulphonic acid concentration at STP |
| M2_SO4 | $\mu g.m^{-3}$ | Sulphate aerosol concentration at STP |
| M2_SS001 | $\mu g.m^{-3}$ | Sea salt concentration (bin 001, 0.03-0.1 μm) at STP |
| M2_SS002 | $\mu g.m^{-3}$ | Sea salt concentration (bin 002, 0.1-0.5 μm) at STP |
| M2_SS003 | $\mu g.m^{-3}$ | Sea salt concentration (bin 003, 0.5-1.5 μm) at STP |
| M2_SS004 | $\mu g.m^{-3}$ | Sea salt concentration (bin 004, 1.5-5 μm) at STP |
| M2_SS005 | $\mu g.m^{-3}$ | Sea salt concentration (bin 005, 5-10 μm) at STP |
| M2_DU001 | $\mu g.m^{-3}$ | Dust concentration (bin 001, 0.1-1.0 μm) at STP |
| M2_DU002 | $\mu g.m^{-3}$ | Dust concentration (bin 002, 1.0-1.5 μm) at STP |
| M2_DU003 | $\mu g.m^{-3}$ | Dust concentration (bin 003, 1.5-3.0 μm) at STP |
| M2_DU004 | $\mu g.m^{-3}$ | Dust concentration (bin 004, 3.0-7.0 μm) at STP |
| M2_DU005 | $\mu g.m^{-3}$ | Dust concentration (bin 005, 7.0-10 μm) at STP |
| M2_BCPHILIC | $\mu g.m^{-3}$ | Hydrophilic black carbon concentration at STP |
| M2_BCPHOBIC | $\mu g.m^{-3}$ | Hydrophobic black carbon concentration at STP |
| M2_OCPHILIC | $\mu g.m^{-3}$ | Hydrophilic organic carbon (Particulate Matter) concentration at STP |
| M2_OCPHOBIC | $\mu g.m^{-3}$ | Hydrophobic organic carbon (Particulate Matter) concentration at STP |
| M2_stdPTfac | 1 | Factor used to convert $\mu g.m^{-3}$ at ambient conditions to $\mu g.m^{-3}$ at STP |
| M2_RH | % | Relative humidity |
| M2_T | K | Air temperature |
| M2_QI | $kg.kg^{-1}$ | Mass fraction of cloud ice water |
| M2_QL | $kg.kg^{-1}$ | Mass fraction of cloud liquid water |
| M2_QV | $kg.kg^{-1}$ | Specific humidity |




Table 7. Summary of where to access different datasets and resources described in this paper.

| Dataset/Resource | Paper Section | Website | DOI |
|---|---|---|---|
| All aircraft instrument data | 3-4 | https://asdc.larc.nasa.gov/project/ACTIVATE | 10.5067/SUBORBITAL/ACTIVATE/DATA001 |
| Falcon merge files | 4.8 | https://asdc.larc.nasa.gov/project/ACTIVATE/ACTIVATE_Merge_Data_1 | 10.5067/ASDC/SUBORBITAL/ACTIVATE_Merge_Data_1 |
| Flight reports | 5.1 | https://asdc.larc.nasa.gov/project/ACTIVATE/pdocuments | N/A |
| Falcon leg index | 5.2 | https://asdc.larc.nasa.gov/project/ACTIVATE/ACTIVATE_MetNav_AircraftInSitu_Falcon_Data_1 | 10.5067/ASDC/ACTIVATE_MetNav_AircraftInSitu_Falcon_Data_1 |
| Aircraft collocation product | 5.3 | Data: https://asdc.larc.nasa.gov/project/ACTIVATE/ACTIVATE_Miscellaneous_Data_1 | 10.5067/ASDC/SUBORBITAL/ACTIVATE_Miscellaneous_Data_1 |
| Aircraft collocation product | 5.3 | Code: https://doi.org/10.6084/m9.figshare.20489442.v2 | 10.6084/m9.figshare.20489442.v2 |
| Cloud detection neural network algorithm | 5.4 | https://asdc.larc.nasa.gov/project/ACTIVATE/ACTIVATE_Miscellaneous_Data_1 | 10.5067/ASDC/SUBORBITAL/ACTIVATE_Miscellaneous_Data_1 |
| MERRA-2 along flight tracks | 5.5 | https://asdc.larc.nasa.gov/project/ACTIVATE/ACTIVATE_Model_Data_1 | 10.5067/ASDC/SUBORBITAL/ACTIVATE_Model_Data_1 |
| FLEXPART trajectory data | 5.6 | https://asdc.larc.nasa.gov/ACTIVATE/ACTIVATE-FLEXPART_1 | 10.5067/ASDC/SUBORBITAL/ACTIVATE-FLEXPART_1 |
| MODIS | 5.7 | https://asdc.larc.nasa.gov/project/ACTIVATE/ACTIVATE-MODIS-MERRA2_1 | 10.5067/ASDC/SUBORBITAL/ACTIVATE-MODIS-MERRA2_1 |
| GOES-16 | 5.7 | https://asdc.larc.nasa.gov/ACTIVATE/ACTIVATE-Satellite_1 | 10.5067/ASDC/SUBORBITAL/ACTIVATE-Satellite_1 |
| MERRA-2 | 5.7 | https://asdc.larc.nasa.gov/project/ACTIVATE/ACTIVATE-MODIS-MERRA2_1 | 10.5067/ASDC/SUBORBITAL/ACTIVATE-MODIS-MERRA2_1 |
| Open data workshop recordings and slides | 7 | https://asdc.larc.nasa.gov/news/activate-data-webinar-materials | N/A |





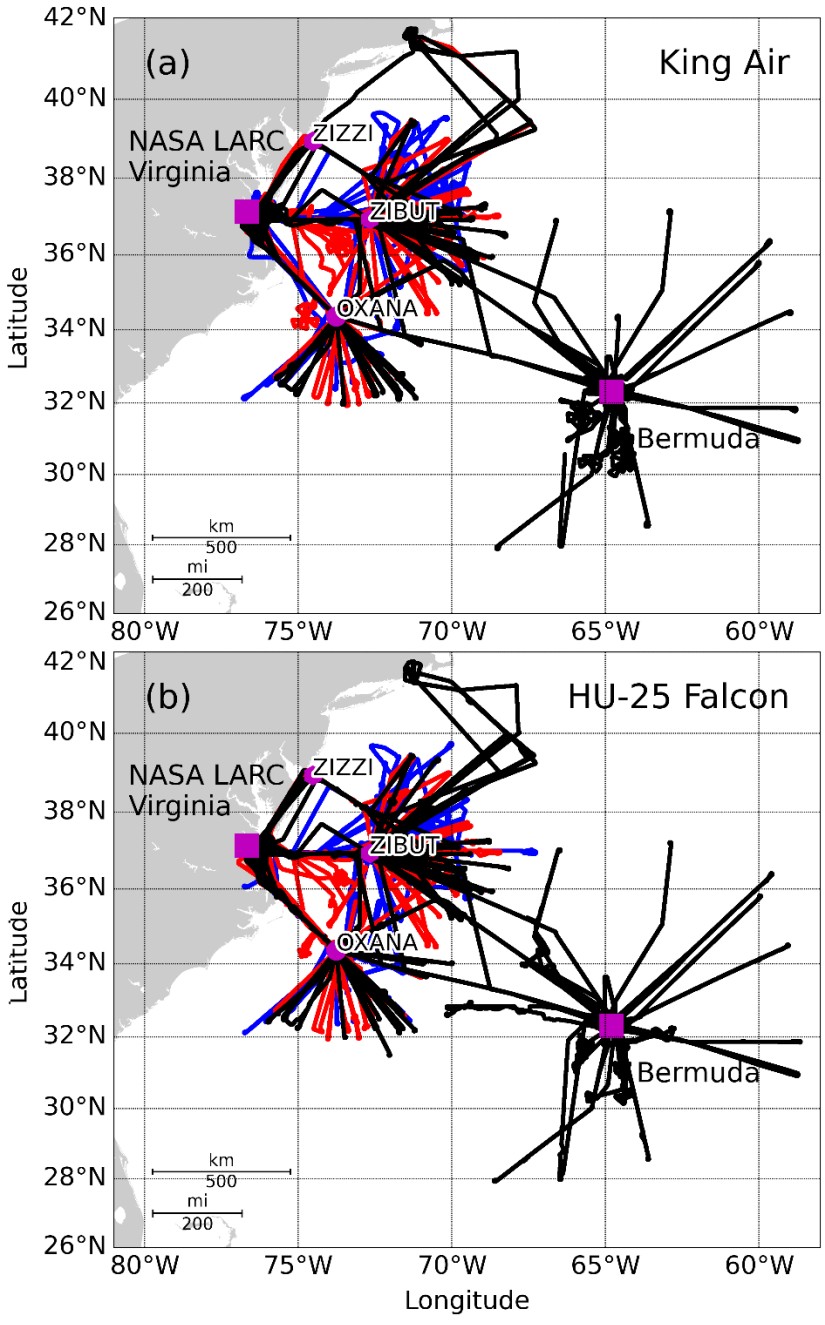

Figure 1: Flight tracks for the (a) King Air and (b) HU-25 Falcon across all three years of flights (blue = 2020, red = 2021, black = 2022). ZIBUT and OXANA are two waypoints used in most flights to adhere to air traffic control restrictions, while ZIZZI was less commonly used.



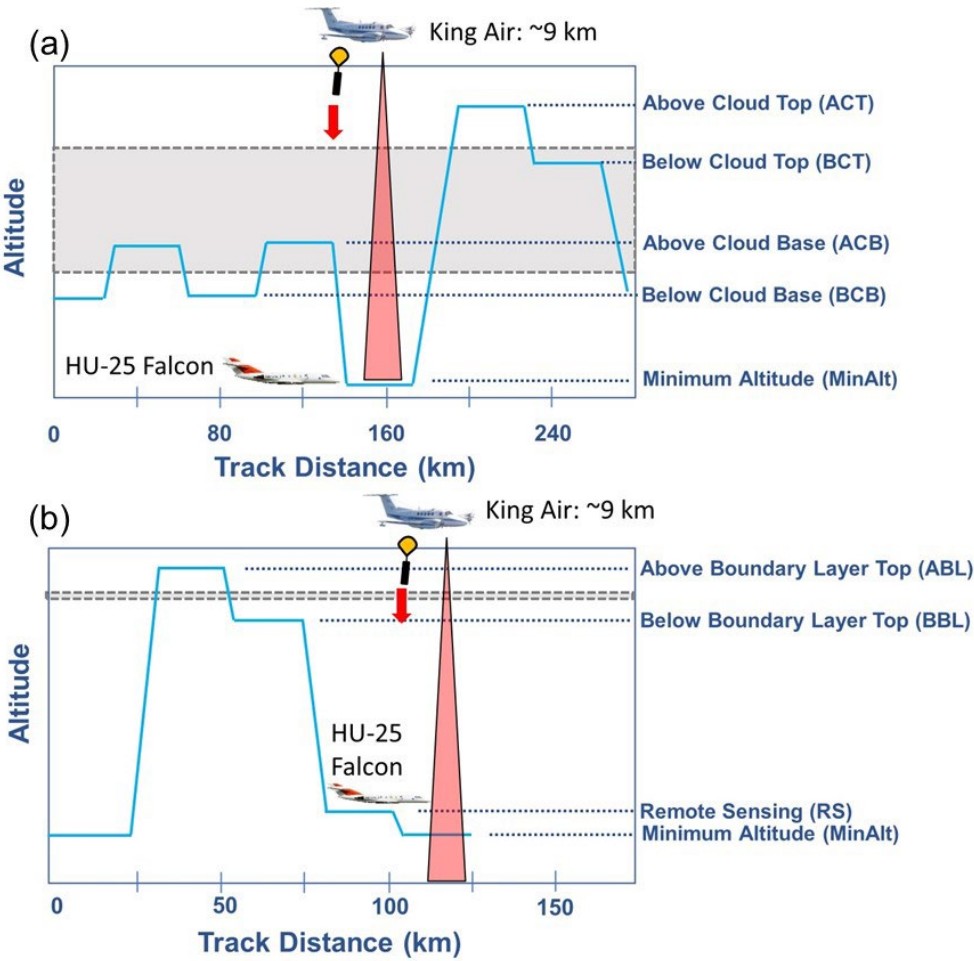

Figure 2: (a) Nominal flight pattern constituting a "cloud ensemble" as part of ACTIVATE flights whereby the Falcon conducts stairstepping (shown in light blue lines) at various levels (~3 min each usually) below, in, and immediately above boundary layer clouds. Note that MinAlt represents the lowest altitude the Falcon could operationally fly at (~150 m above sea level). The King Air flies overhead around ~9 km. The gray shaded area represents a cloud. Typical statistical survey flights included ~3 cloud ensembles. (b) Nominal flight pattern for "clear ensembles" whereby the Falcon stairsteps at levels immediately above and below the boundary layer top (represented by the horizontal gray bar) and legs near the lowest operational altitude the aircraft could fly at. The Remote Sensing leg was an additional leg just above the MinAlt leg that was more reasonable in terms of applications involving data comparisons with the remote sensors on the King Air. The vertical axes are compressed to show both aircraft.


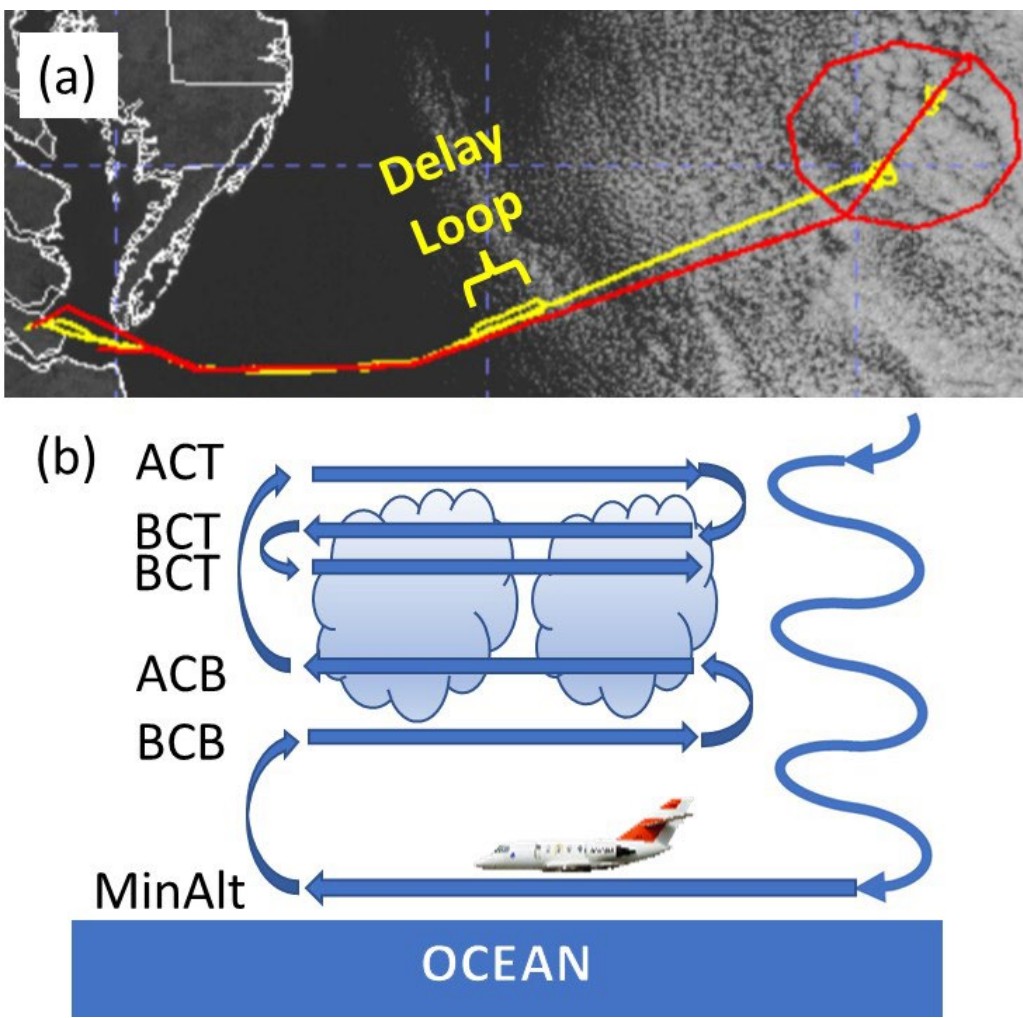


Figure 3: (a) Visual summary of Research Flight 13 (1 March 2020, L1) tracks for both the (yellow) Falcon and (red) King Air overlaid on GOES-16 imagery (UTC 15:21). Highlighted in the flight is a "delay loop" (described in sect. 2.4) executed by the Falcon to improve coordination with the King Air. (b) The generic Falcon pattern used in process study flights including stacked level legs ("wall") with spiral soundings before and after the wall; meanwhile the King Air (not shown in panel b) flies

aloft characterizing the same area. In this flight, in place of a spiral sounding at the end of a wall, the Falcon conducted a slant descent from the last BCT leg to a subsequent MinAlt leg.



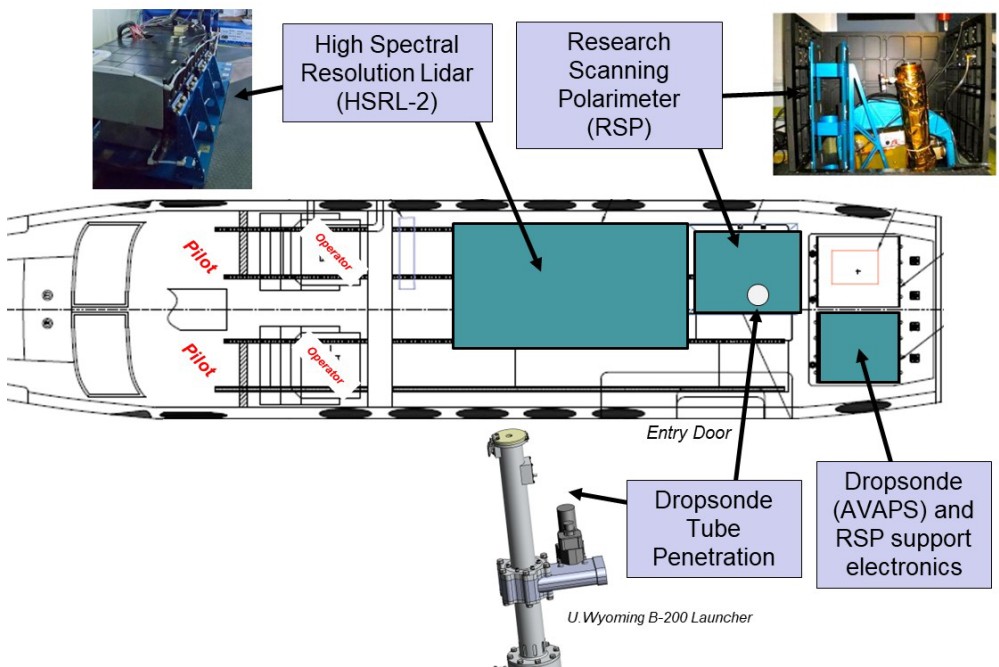

Figure 4: King Air interior layout.




Figure 5: A qualitative visualization of selected HSRL-2 data products archived for a representative ACTIVATE flight (RF157 on 18 May 2022, L2). This flight was the second one on this day, returning from Bermuda to NASA LaRC. (a) A curtain vertical profiles of aerosol backscatter (532 nm) as a function of UTC time for the entire flight provides context of the aerosol particles measured. The labeled boxes indicate regions where subsets of HSRL-2 data products are highlighted in the corresponding small boxes below panel (a). (b) Cloud data: blue dots show (left) cloud top height and (right) cloud top extinction, averaged over the first optical depth. Both are overlaid on the backscatter curtain at the same times, with extinction being plotted on a secondary y-axis (not shown) (c) Boundary layer and lower troposphere aerosol particles: mixed layer height (blue dots), surface wind speed (black line), aerosol type, aerosol depolarization (UV (355 nm), VIS (532 nm), IR (1064 nm)), and backscatter Ångström exponents corresponding to spherical and nonspherical particles. (d) Elevated aerosol layer: aerosol backscatter (UV (355 nm), VIS (532 nm), IR (1064 nm)), backscatter Ångström exponents (VIS/UV and IR/VIS), lidar ratios (UV and VIS), aerosol extinction (UV and VIS), extinction Ångström exponent (UV/VIS), and total column AOT (UV and





VIS). The opaque cloud average extinction, surface wind speed, and total column AOT products are all overlaid on the
backscatter curtains for context but plotted on a secondary y-axis and scaled for visibility inside the inset.





Figure 6: Visual summary of HU-25 Falcon (a) exterior probes and (b) interior layout. The Cloud Aerosol and Precipitation Spectrometer in (a) includes the Cloud and Aerosol Spectrometer (CAS) probe described in sect. 4.5.



Figure 7: Closure analysis for particle number concentration measurements derived from an ultrafine CPC, SMPS, and LAS. (a-b) Time series data are shown for Research Flight 12 on 29 February 2020, (c) an average size distribution (SMPS in blue and LAS in magenta) during a BCB leg at approximately UTC 16:15 (bottom left), and (d) a scatterplot of the integrated number concentration derived from LAS+SMPS instruments against number concentration directly measured by a CPC. Units of scm$^{-3}$ represent standard cm$^{-3}$.



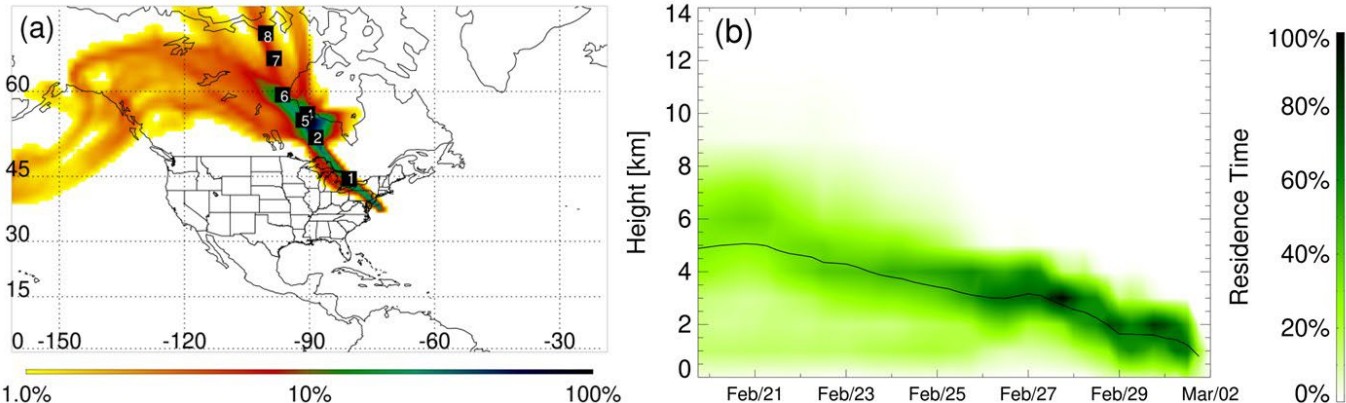

Figure 8: (a) Horizontal and (b) vertical views of simulated air mass residence time for flight measurements at 19:22 UTC on
1 March 2020 (RF14). The labels with white numbers on the map in (a) indicate the approximate locations of the center of the
plume and the corresponding upwind days. Residence time is color coded by logarithmic (a) and linear (b) grades representing
the ratios to the maximal integrated residence times of each view, respectively.


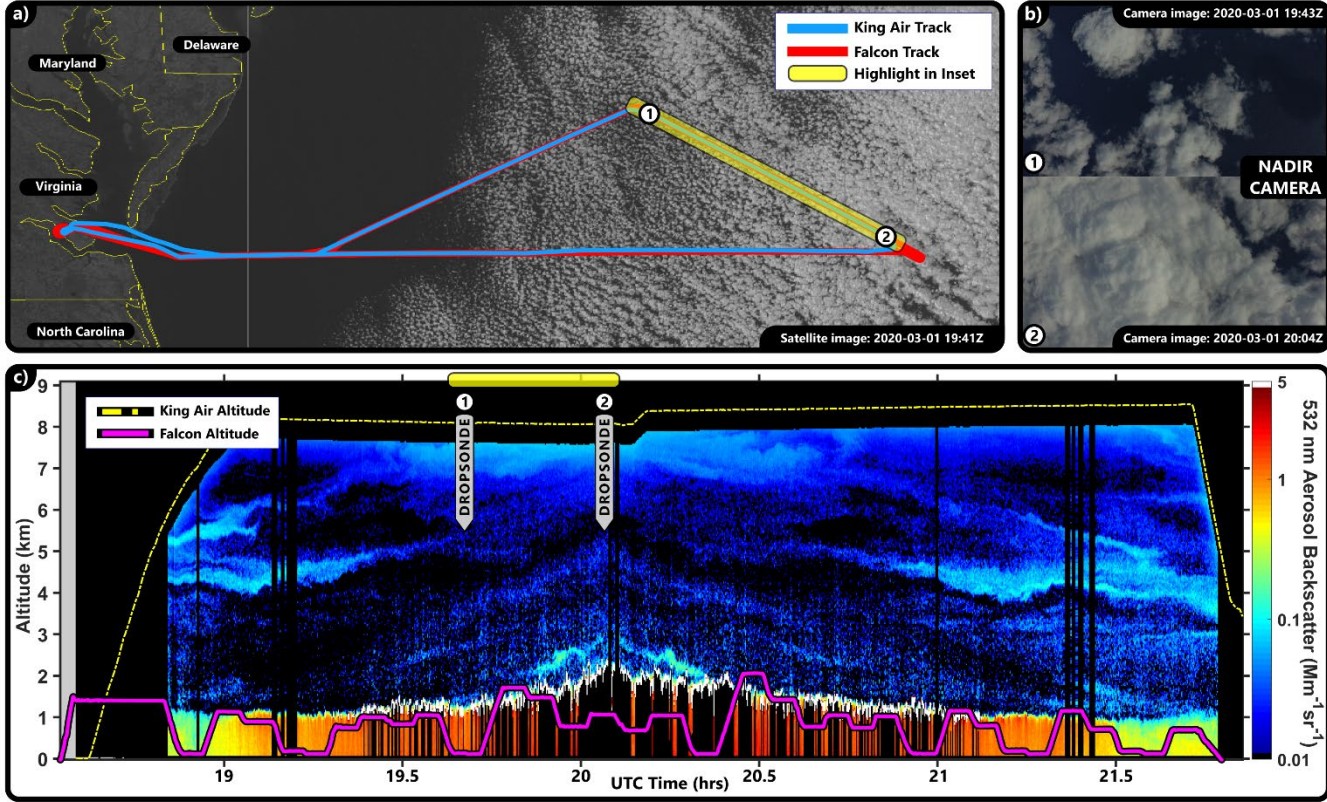

Figure 9: (a) Flight tracks of the King Air and HU-25 Falcon on RF14 on 1 March 2020 overlaid on GOES-16 visible imagery captured at UTC 19:41. The number 1 and 2 labels correspond to where the two dropsondes were launched along the downwind leg (highlighted in yellow) during this flight. These indicators are consistent in all three panels. (b) Nadir camera imagery from the King Air at the time the two dropsondes were launched. (c) Time series of the King Air aerosol backscatter shown as curtain profiles, along with the altitude trace of the King Air and Falcon aircraft. Shown also are the locations of where the 1515    two dropsondes were launched and the downwind leg is highlighted in yellow.

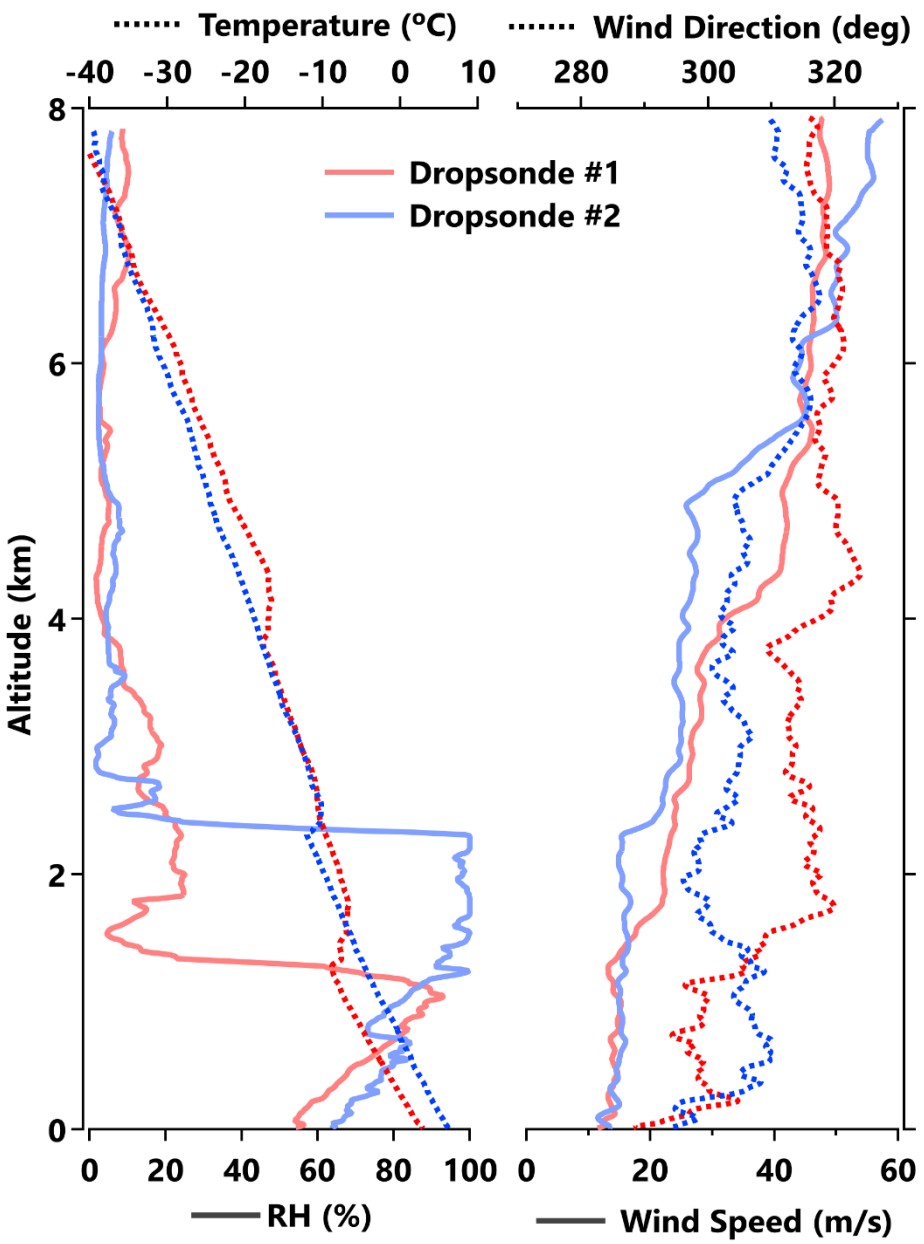


Figure 10: Vertical profiles of variables measured with the two dropsondes launched in RF14 (1 March 2020) with the markings of the drop locations shown in Fig. 9.


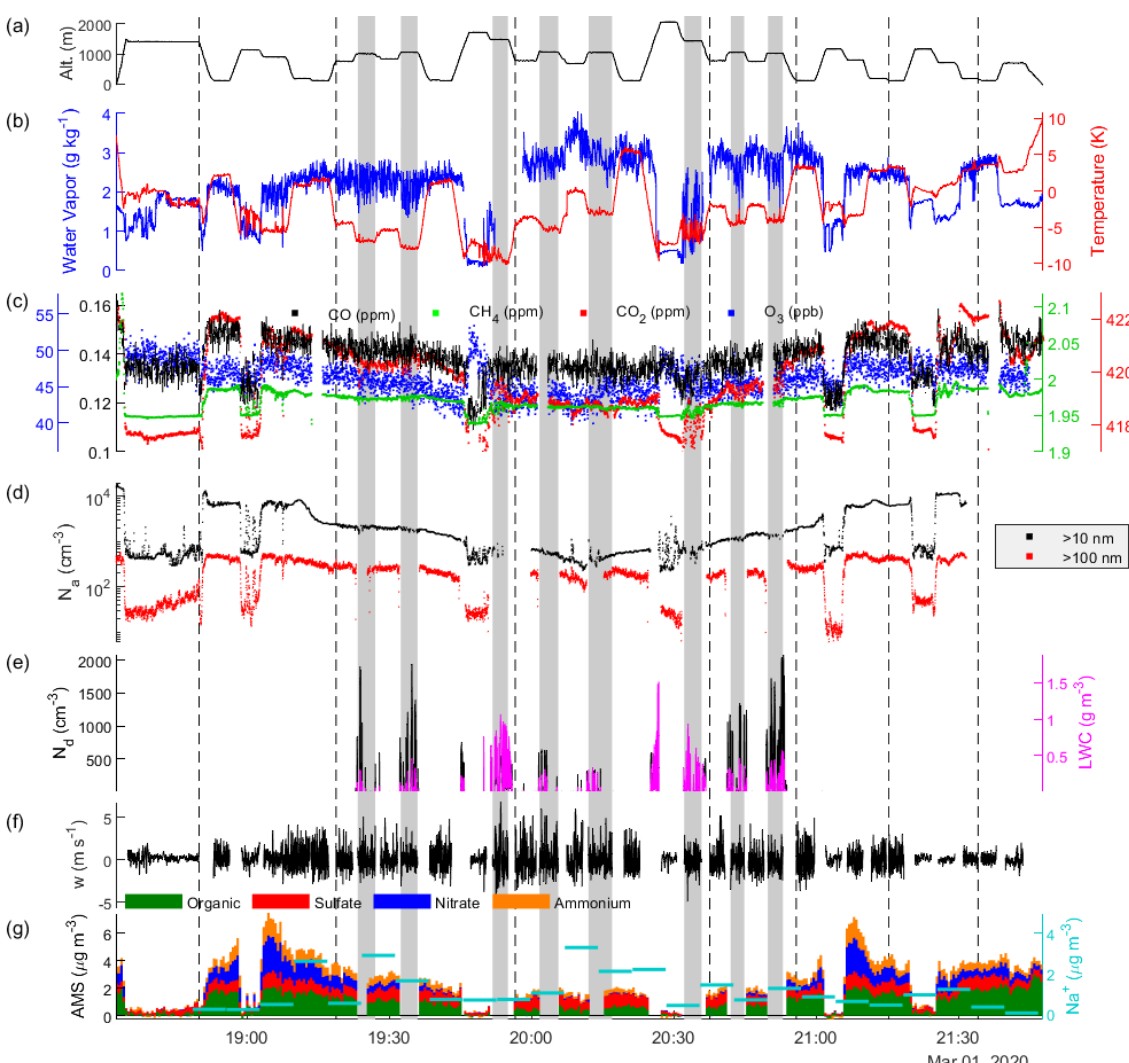

Figure 11: Time series (UTC time) of Falcon data for RF14 on 1 March 2020. Shown are the archived Falcon in situ data for (a) altitude (Applanix 610), (b) water vapor (DLH) and temperature (Rosemount 102 sensor), (c) trace gases (Picarro model G2401-m for CO, $CO_2$, $CH_4$ and 2B Tech. Inc. model 205 for $O_3$), (d) aerosol particle number concentration for diameter > 10 nm (TSI-3772 CPC) and > 100 nm (LAS), (e ) cloud droplet number concentration and LWC (FCDP), (f) vertical wind speed (TAMMS), and (g) speciated aerosol mass concentrations from the AMS (organic, sulfate, nitrate, ammonium) and PILS (sodium). Shaded gray vertical sections denote the two level leg types in cloud (above cloud base [ACB] and below cloud top [BCT]). The dashed vertical black bars mark the beginning of either clear or cloud ensembles (ensembles in order: clear, cloud, cloud, cloud, clear, clear).



Figure 12: Representative data products derived from FCDP and 2D-S on the Falcon for RF14 on 1 March 2020. (a) Time series of cloud droplet size distribution for RF14 on 1 March 2020 based on combining FCDP and 2D-S data, (b) average size distribution of liquid (FCDP and 2D-S Horizontal) and ice (2D-S Horizontal) for cloud measurements with LWC > 0.02 g m$^{-3}$ and $N_d$ > 10 cm$^{-3}$, and (c) example images captured by the 2D-S Horizontal probe for UTC 20:05:35 – 20:05:50.