# Peer review of "Spatially-coordinated airborne data and complementary products for aerosol, gas, cloud, and meteorological studies"

_Earth System Science Data, 2023_

## Author Comment (AC1)

We thank both reviewers for helpful comments to help us improve the paper. Below are our responses in blue font.

Reviewer 1:

The paper by Sorooshian et al. describes the datasets associated with the ACTIVATE field campaign. The description is very thorough including the flights, measurement techniques and additional derived products that greatly support the usage of the data for answering science questions. One minor comments if that it would be great if authors could add a section on summary of science results available so far or to come leading to answering the science questions of the investigation. Additional by line comments below

Response: This is something in fact we thought about and decided to avoid as we plan to have a more comprehensive science overview paper. While we can add an overview of science results as of this date, it would reduce the "thunder" from that planned paper. We added the following sentence to the end of the Introduction section to clarify this point:

"A forthcoming paper will provide a comprehensive overview of the science results from ACTIVATE and how those fit into the larger picture of past campaigns focused on aerosol-cloud interactions."

**Comments by Line**

Introduction: It would be good to summarize other field campaigns performed in the region that had some similarities (for instance, DoE's Two-Column Aerosol Project, maybe IMPACTS too) and also mention other aerosol-cloud focused field campaigns performed in other regions to provide context on where this campaign fits

Response: As noted in the previous comment's response, we plan for a more scientific overview which will naturally reference past aerosol-cloud campaigns for compare/contrast purposes. But this is a good idea to at least concisely mention a few others in other regions and also for the ACTIVATE region. We added the following text:

"The northwest Atlantic differs from subtropical regions often chosen for aerosol-cloud interaction campaigns due to multiple cloud types within reach, rather than the stratocumulus clouds that are simpler to characterize owing to their high cloud fraction and well-defined vertical structure as demonstrated by campaigns over the northeast Pacific (e.g., Durkee et al., 2000; Sorooshian et al., 2018), southeast Pacific (e.g., Mechoso et al., 2014), and southeast Atlantic (e.g., Zuidema et al., 2016; Redemann et al., 2021). ACTIVATE adds to the much needed inventory of data over the northwest Atlantic to build on efforts from projects such as the North Atlantic Regional Experiment (NARE; Leaitch et al., 1996), the Surface Ocean-Lower Atmosphere Study (SOLAS; Leaitch et al., 2010), the International Consortium for Atmospheric Research on Transport and Transformation (ICARTT; Avey et al., 2007), the Two-Column

Aerosol Project (TCAP), and the Investigation of Microphysics and Precipitation for Atlantic Coast-Threatening Snowstorms (IMPACTS)."

Table 2. Associated to this table, it would be nice to have a visual reference of the flight track and cloud conditions for each flight to allow looking for flights by visual inspection. For this purpose a multi-panel Figure could be added to the Appendix with the flight track over a representative visible image for that day

Response: This is an excellent idea. It would be too much in our view to do this for ACTIVATE as we had 179 flights. Of high relevance to this comment is that we included these types of figures in our flight reports, which are publicly available and discussed in the paper. We added the following sentence to Sect. 5.1:

"Of particular importance in the flight reports is inclusion of flight tracks overlaid on satellite imagery to show cloud conditions."

1. "complex refractive index" is listed here but "Aerosol Refractive Index (column)" is listed on Table 3. I'm guessing they refer to the same? Table 3 lists the uncertainty, is this the same uncertainty for both the real and the imaginary parts?

Response: Thanks for your clarification. We have updated Table 3 to correctly list the "Aerosol Real Refractive Index (column)", which is now also reflected in the body of the paper where we change "complex refractive index" with "real part of the refractive index". The uncertainty in the imaginary refractive index is captured in the aerosol single scattering albedo parameter.

Section 3.4. Is N_a the only variable planned to be derived jointly? If there are others please list them, including the ones that are improvements over the ones measured by each instrument individually.

Response: We have added just one other jointly derived variable (i.e., cloud liquid water content at cloud top) that the science team knows about right now that will be archived in the future:

"Forthcoming work will summarize additional joint retrieval products that will be archived for public use once they are developed, including retrievals of $N_d$, liquid water content (LWC), and autoconversion rate at cloud top."

809-810. Can you describe a bit more how the residence time is computed and plotted? I think I understand Fig 8b (for a given time, see what fraction of particles are at what height) but Fig 8a is a bit more confusing as it seems to be mixing multiple times

Response: Thanks for pointing this out. We have rewritten the paragraph in question with a description of how the residence time is computed and plotted:

"In the FLEXPART backward mode, a plume of passive particles is released from the aircraft location and advected and dispersed backwards in time. For each 60-second merged aircraft measurement every 10 minutes, FLEXPART initiates 10,000 passive particles at the sampling location and calculates backwards for 10 days. The released particles represent the air masses (plume) intercepted by the aircraft. For a completed backward simulation, the total residence time (RT) of the plume in a given 1°×1° grid cell can be calculated by summing the time duration of all particles that have been resident in the cell during the 10-day transport. If a large fraction of particles passes through a surface grid cell multiple times, the grid cell would accumulate a long residence time, and emissions therein would have a large contribution to the plume intercepted by the aircraft. The horizontal distribution of vertically integrated residence times (Fig. 8a) can be readily used to determine a trajectory-like transport pathway, while the vertical distribution of RT (Fig. 8b) can clearly indicate plume transport height and acquisition of surface emissions."

We also revised the Figure 8 caption:

"Figure 8: (a) Horizontal and (b) vertical views of simulated air mass residence time (RT) for flight measurements at 19:22 UTC on 1 March 2020 (RF14). The labels with white numbers on the map in (a) indicate the locations of maximal RT for the corresponding upwind day. Transport pathways differ significantly, and absolute RT values may vary a lot between cases. For a better comparison of transport pathways between cases, RT is expressed as a percentage of the maximal integrated value during the 10-day trajectory period. RT is color-coded with (a) logarithmic and (b) linear scales, respectively."

**Technical corrections**

 Table 2. Some sentences seem to be cutoff on the "Special notes" columns (e.g., for RF 5).

Response: Thanks for pointing this out. That flight in particular was not actually cut off and maybe the issue is that it is hard to tell apart flights. We have edited the table to fix this issue by separating rows with horizontal lines.

1. The aerosol hygroscopic growth product is not mentioned on Table 3

Response: Since the product is under development with data specifications ready (which is the spirit of Table 3), we add the following line to the caption:

 "Products under development are omitted from this table and readers are referred to Sect. 3 for more description."

1. Virkula reference in wrong format and missing a period between sentences

Response: Fixed as shown here:

"Scattering coefficient measurements have been corrected for angular truncation (Anderson and Ogren, 1998) and absorption coefficients were corrected using guidance from Virkkula (2010)."

Reviewer 2:

Review Sorooshian et al., essd-2023-109

General comments:

The paper by Sorooshian et al. provides a guide for the airborne data collected in the ACTIVATE project and its best use. The main issue with data collected in airborne field campaigns is indeed that the use of data is often much easier for the campaign participants. This is because contextual knowledge on data format and known issues are often restricted to the participating science team. The ACTIVATE project in particular, due to its high number of flights hours (almost 600 hrs on each of two aircraft) and hence data volumes, is prone to such challenges. Hence, the concept of the submitted work is very much appreciated and will serve a large community beyond the ACTIVATE team. Admittedly, I struggled with the idea to exclude preliminary scientific findings in ACTIVATE from the discussions of data specifics (i.e., the subject of this paper). However, after reading the paper in its entirety, I understand the purpose of this paper, which I would generally describe as facilitating access to, and scientific use of, ACTIVATE data by a broader community than the ACTIVATE science team.

The ACTIVATE dataset is completely unique and will undoubtedly be the basis for many studies of aerosol-cloud-meteorology interactions in the future. It includes a number of excellent additions relative to previous datasets of similar scope and intent – most notably, I appreciate the creation of a collocation mask (section 5.3), the MERRA-2 reanalysis collocation exercise (section 5.5), the FLEXPART analyses (section 5.6), and the correlative satellite data inclusion (5.7).

The paper is well structured and well-written, and therefore it is easy to read. Figures and Tables are of good quality throughout. The descriptions on data details will save future users tremendous amounts of time, provided that the caveats stated in the paper will be heeded. I have no specific comments, but I include below two overarching requests for additions and a list of technical comments. I consider the authors' addressing neither the overarching requests nor the minor suggestions as crucial for my acceptance of this manuscript. I would suggest the paper is acceptable for publication with minor revisions, and I will leave it to the authors to choose which of my suggestions to address.

Response: Thank you for the encouraging reflection on the draft article.

Overarching requests:

- I am missing in the paper a guide on where to look in the data set to find the most relevant data for addressing the specific ACTIVATE science objectives. Given the vast amounts of data available, this could limit the use of the data if future users would have to hunt for appropriate data to address a specific science objective. Would it be possible to include a general mapping of the ACTIVATE flights (or flight periods) onto the science objectives, maybe something simply indicating the relevance of a flight (or flight period) for addressing the ACTIVATE (or additional) science objectives?

Response: This is an excellent idea. Because of the reliance on a routine flight strategy with 179 flights, it is hard to divide flights into which particular science objectives they map best onto. We have already designated which flights were "process study" flights, which provide opportunities for model simulations and intercomparisons with field data. Also, the satellite underflights and 'unicorn aerosol modules' have already been marked up in Table 2, which is helpful for addressing remote sensing objectives. Objectives 1-2 can arguably be addressed using statistics from all flights. As a result, we add the following text:

"It is difficult to assign specific flights to ACTIVATE's individual scientific objectives (sect. 2.1) because statistics from all flights can be helpful to each objective; however, that being said, Table 2's notes of special features and designation of some flights as "process study" flights (described in sect. 2) can be helpful for data users most interested in remote sensing objectives (e.g., satellite underflights, relatively more cloud-free conditions with high aerosol levels) and modeling activities such as large eddy simulation of cold air outbreak conditions (e.g., Li et al., 2022)."

- I think the value of the "statistical survey flight" nature of 90% of the ACTIVATE flights is understated. There have only been a few campaigns that use this concept for a portion of airborne flight campaigns, instead of targeting the largest signals or most interesting atmospheric (hot spot) features. Please include a small discussion in section 2.2 highlighting how novel this approach is, how valuable it is for measuring the probability density distribution of aerosol and cloud properties in a region, and how this traces back to the processes the campaigns are after.

Response: Thank you for this great point. We added a paragraph about this as shown here:

"The disciplined approach of statistical surveys is uncommon for airborne flight projects as often the temptation is to target the most interesting features on a given day such as the strongest aerosol signal (e.g., smoke or dust plume) or opportunistic experimental conditions suited for aerosol-cloud interactions (e.g., ship tracks) (e.g., Christensen et al., 2022). Building routine statistics below, within, and above boundary layer clouds with a consistent flight strategy across a large number of flights is advantageous for developing probability density distributions of aerosol, cloud, and meteorological properties in a given region, which can be used to trace back onto processes. Furthermore, this approach provided a consistent dataset to better optimize data use among a diverse set of users."

Minor suggestions:

1. Line 39: "Falcon flew conducted profiling at different level legs…" – this phrasing is awkward. Profiling at different level legs seems a contradiction in terms. The word "flew" can be struck it seems?!

Response: Change made. Here is the revised sentence:

"The HU-25 Falcon conducted profiling at different level legs below, in, and just above boundary layer clouds (<3 km) and obtained in situ measurements of trace gases, aerosol particles, clouds, and atmospheric state parameters."

    2.  Line 43: Mention total number of dropsondes.

Response: Change made:

"The King Air (the high-flyer) flew at approximately ~9 km conducting remote sensing with a lidar and polarimeter while also launching dropsondes **(785 in total)**."

    3.  Line 52: The implication that the study of all interactions related to anthropogenic forcing absolutely require airborne measurements seems a stretch.

Response: Sentence revised:

"This uncertainty stems partly from the difficulty in experimentally characterizing such interactions in the atmosphere due to the need for **methods such as with** airborne platforms."

    4.  Line 59: please specify the targeted seasons.

Response: Change made and note that we preface the two seasons with "e.g.," as we ended up capturing more than just those two seasons.

"ACTIVATE flights were strategically executed in different seasons **(e.g., winter and summer)**…"

    5.  Line2 81-86: The wording makes it unclear if these are the original ACTIVATE science objectives. Please clarify.

Response: Text added to clarify these were the original objectives:

"ACTIVATE generated a novel dataset that can be used to address three overarching objectives **that were developed during the conception of the mission plan**: …"

    6.  Line 86: Please specify the number of flight hours targeted for the threshold and baseline science mission, if possible, and/or state how the number traces to science objectives.

Response: We updated Table 1 to show the threshold/baseline mission flight goals, which comes down to just number of clear and cloud ensembles. We also added the following text:

"The threshold and baseline science mission success metrics from a flight perspective hinged on acquiring many of these ensembles for more robust calculations of aerosol-cloud-meteorology interactions. ACTIVATE far surpassed the number of ensembles needed for threshold and baseline mission requirements. Ensemble numbers and definitions of these mission categories are provided in Table 1."

Table 1. Overall summary of ACTIVATE flight metrics categorized by each of the six deployments between 2020 and 2022.

Joint ensembles represent when both planes were in coordination and conducting the series of legs (in some combination) shown in Fig. 2. The number of dropsondes shown represent those with full profiles of all variables with good parachute performance. **The threshold science mission goal for cloud ensembles required only 100 of the 200 to be with joint aircraft and the remainder to be at least with just the Falcon. The threshold science mission represents a descoped version of the baseline mission to satisfy the minimum science acceptable for the investment, while the baseline mission satisfies performance requirements necessary to achieve the full science objectives of the mission.**

| | Research Flights | | | Flight Hours | | Joint Ensembles | | Underflights | | Process Study Flights | Dropsondes |
|---|---|---|---|---|---|---|---|---|---|---|---|
| | Falcon | King Air | Joint | Falcon | King Air | Cloudy | Clear | ASTER | CALIPSO | | |
| Winter 2020 (14 Feb – 12 Mar) | 22 | 17 | 17 | 73 | 59 | 43 | 28 | 1 | - | 2 | 59 |
| Summer 2020 (13 Aug – 30 Sep) | 18 | 18 | 18 | 60 | 67 | 58 | 36 | 1 | 3 | 2 | 107 |
| Winter 2021 (27 Jan – 2 Apr) | 17 | 19 | 15 | 56 | 66 | 47 | 25 | 1 | 3 | - | 100 |
| Summer 2021 (13 May – 30 Jun) | 32 | 32 | 32 | 106 | 108 | 103 | 74 | 1 | 1 | 2 | 150 |
| Winter 2021-2022 (30 Nov – 29 Mar) | 55 | 54 | 53 | 182 | 193 | 198 | 72 | - | 1 | 2 | 214 |
| Summer 2022 (3 May – 18 Jun) | 30 | 28 | 27 | 97 | 98 | 86 | 46 | 2 | 3 | 4 | 155 |
| *Sum* | *174* | *168* | *162* | *574* | *592* | *535* | *281* | *6* | *11* | *12* | *785* |
| *Threshold Mission Goal* | | | | | | *200* | *12* | | | | |
| *Baseline Mission Goal* | | | | | | *250* | *15* | | | | |

7. Line 106: Please comment briefly on impact of deviations on science.

Response: Text added to note there was no impact on science:

"As a result of operational delays, aircraft maintenance challenges, and COVID-19 emerging during the first deployment, deviations were necessary relative to the original **flight schedule plan; however, the overall science plan was unaffected**."

8. Line 117: Unclear wording. How does one ascent provide multiple profiles?

Response: Fixed by making ascent plural:

"The slant ascent**s** from MinAlt to ACT provided multiple in…"

9. Line 122: Please define conditions of high aircraft coincidence.

Response: Text added:

"The RS leg was implemented under conditions of high aircraft coincidence **(<5 min and <6 km of separation between Falcon and King Air)** and when no clouds affected the field of view."

10. Line 128: Is there a reason to mix units (min and km) for the clear ensembles only?

Response: Change made to allow for better consistency:

"The time span (distance) of each leg and cloud ensemble was ~3.3 min (~24 km) and ~35 min (~250 km), respectively, while clear ensembles were typically ~15 min **(~100 km)** (Dadashazar et al., 2022b)."

11. Line 139: Please use high-flying aircraft throughout.

Response: Change made to line in question and checks were made throughout the paper too.

"During that time, the high-flying…"

12. Line 149: I find the nomenclature of L1 and L2 unfortunate, given the remote sensing community's use of the letter L to designate processing levels of remote sensing data.

Response: We also think this is an unfortunate coincidence. It is difficult for us to change this though as this nomenclature is mostly hard-wired at this point with airborne field data (as in past campaigns too). We added the following text to try to address this comment:

"Note here that launch number refers to the aircraft launch number per day following the International Consortium for Atmospheric Research on Transport and Transformation (ICARTT; described more in

sect. 7) naming convention (Northup et al., 2017) and not processing level as employed by the satellite and remote sensing community."

13. Line 179-190: Grammatically, there is a mix of conditions (i + ii) and actions (iii + iv) which reads awkwardly. Also, the bulletized list conflicts with the list one level higher (also i to v).

Response: We changed the 5 most outer numbered bullets to just be circular bullets.

14. Line 191: Awkward phrasing. I suggest replacing "underneath" with "in coordination with".

Response: Change made:

"Numerous flights were conducted directly in coordination with satellites to…"

15. Line 195: Suggest tot replace "data" with "instruments".

Response: Change made:

"In a few instances, the two aircraft coordinated to observe aerosol particles in clear sky conditions with the complete set of remote sensing polarimeter and lidar **instruments** with a matching full vertical profile of in-situ observations. This type of aircraft observation module that must include an ascent/descent or spiraling aircraft pattern by the in-situ aircraft, became known as "unicorn aerosol modules"."

16. Line 198" I think that we did such unicorn aerosol modules in ORACLES-2016, unless the emphasis is on severe clear sky conditions (in which case you can ignore this comment). The paper by Xu et al. (2021) is a joint lidar-polarimeter retrieval with in situ aircraft validation for such a case. Happy for you to ignore this comment - I do not mean to be petty.

Response: This is a good point. We revised the text:

"In a few instances, the two aircraft coordinated to observe aerosol particles in clear sky conditions with the complete set of remote sensing polarimeter and lidar instruments with a matching full vertical profile of in-situ observations; this is related in part to past attempts to do such coordinated maneuvers in other regions (Xu et al., 2021).

17. Line 212: A reference to the Tudor Hill site observations would be helpful.

Response: Those data are not publicly available as of yet and so we cannot report a DOI number; this is based on a communication with the NSF project PI (Becky Alexander) on 30 May 2023. For now, we can suggest to leave the text as is and trust that future data users will find those data on their own when they get archived/published.

18. Line 254-257: Some information on the availability of the derived lidar products would be helpful (i.e., how often were conditions conducive for deriving these products), maybe as percentage of the availability of primary data.

Response: We added new text to sect. 3.2:

"Cloud screening is performed using a convolution of the measured 532 nm signal with a Haar wavelet to enhance edges (Davis et al., 2000) separating the sharper cloud edges from less pronounced aerosol features in each lidar profile. Cloud top altitudes are provided. Both cloud screened and non-cloud screened aerosol scattering ratio (i.e., ratio of aerosol scattering to molecular scattering), aerosol backscatter, and aerosol depolarization profiles are computed and provided at the three wavelengths. Aerosol extinction, aerosol optical thickness, and lidar ratio at 355 and 532 nm are provided only for cloud-free regions. If a cloud is detected in a profile, these data products are restricted to the region above the cloud top. The 532 nm molecular scattering signal for each profile is used to check that signal levels are sufficiently high to derive these aerosol products. Aerosol depolarization at 532 nm and 1064 nm (355 nm) is computed when aerosol scattering ratio values exceed 0.2 (0.068). The HSRL-2 backscatter and depolarization products are reported as 10 second averages while the extinction and lidar ratio products are averaged to 60 seconds. Higher resolution products are available from the HSRL-2 team upon request."

Davis, K. J., Gamage, N., Hagelberg, C. R., Kiemle, C., Lenschow, D. H., and Sullivan, P. P.: An Objective Method for Deriving Atmospheric Structure from Airborne Lidar Observations, Journal of Atmospheric and Oceanic Technology, 17, 1455-1468, https://doi.org/10.1175/1520-0426(2000)017<1455:AOMFDA>2.0.CO;2, 2000.

19. Line 411: Practically, where do readers find instrument team contact information? This info is not provided in Tables 3 or 5.

Response: Good point about clarifying this better. We updated text in a couple areas including sect. 7, which focused on the ICARTT files:

Just after the previous Line 411: "Details about instrument team contact information are discussed in sect. 7."

Section 7: "The contents of each ICARTT file include data notes in a README tab including **contact information for the instrument data (i.e., instrument principal investigator [PI] name and data manager [DM])**, PI institution, campaign name,…"

20. Line 509: Please include a recommendation for how to approximate total ambient extinction from the 2021-2022 flights when the scattering coefficients and f(RH) were observed for submicron aerosol only.

Response: We added the following text:

"For the 2021-2022 datasets, we recommend using FCDP microphysical data (which are measured at ambient RH and described in sect. 4.5) and Mie Theory assumptions to calculate ambient extinction for the supermicrometer particle population."

21. Line 532: If the results in Figure 7 are representative of the closure in CN, please provide the closure results for Figure 7d in statistical formats (i.e., fit equation, bias error, mean absolute error), as those will be useful to estimate the expected accuracy of the CN measurements. A reference back to (comparison with) the uncertainty stated in Table 5 would be great.

Response: This text was added to the Figure 7 caption:

"For panel d, orthogonal distance regression (ODR) linear fitting resulted in a slope of 0.961, intercept of -1.07 cm$^{-3}$, and coefficient of determination (r$^2$) of 0.868. Mean absolute error (MAE) and mean absolute percentage error (MAPE) values of 148 cm$^{-3}$ and 8.45%, respectively, are well within stated uncertainties in Table 5 and demonstrate excellent measurement closure."

22. Line 540: Please add a description of the SS scanning pattern and/or the general approach for cycling through the SS range.

Response: We added the following text:

"The reported CCN concentration depends on the instrument supersaturation, which is also reported in the data files. For the 2020 dataset, the instrument supersaturation was linearly scanned between approximately 0.2-0.7% supersaturation with a single upscan or downscan consisting of 60 seconds. For the 2021 and 2022 datasets, the instrument supersaturation was held constant at approximately 0.4% supersaturation for each flight. Data users are encouraged to consult the data files for the precise, calibrated instrument supersaturation corresponding to each data point."

23. Line 574: see also comment 20) above: Please discuss the impact of calculating total extinction in this way (i.e., by combining submicron scattering and total absorption).

Response: We added the following text:

"Since scattering is typically the dominant component of extinction and absorption is assumed to be dominated by brown carbon and black carbon in continental outflow, archived optical properties calculated using a combination of nephelometer and PSAP measurements (i.e., extinction coefficient and SSA) should be treated as representing submicrometer aerosol. Care should be taken for cases suspected to be influenced by absorbing dust, which do not satisfy the assumptions above."

24. Line 577: I think you should clarify that these are only ambient values if absorption humidification is equal to unity.

Response: We added the following text:

"Note that ambient scaling assumes that there is no absorption enhancement due to humidification, since we do not have the necessary information regarding the particle mixing state to calculate those enhancements accurately."

25. Line 594-595: Awkward phrasing to me.

Response: Text was revised to make it (hopefully) less awkward:

"It is recommended that data users use strict criteria to only use aerosol data in cloud-free conditions. As one example, a recent ACTIVATE study used aerosol data only when cloud liquid water content (LWC) was less than 0.001 g m$^{-3}$ (Schlosser et al., 2022)."

26. Line 639: Please replace "generated" with "generate".

Response: Change made:

"The recorded ensemble of 'slices' obtained rapidly by triggered photodiodes help generate 2D images…"

27. Line 652: Particle shapes would affect calibrations as well.

Response: Good point. We added the following text:

"For instance, the sizing for these probes is calibrated assuming water droplets with a corresponding refractive index; thus, if coarse mode dust, biological particles, and/or sea salt particles are present, there will be sizing biases due to the varying refractive indices and possible aspherical shapes of these aerosol types relative to water droplets."

28. Line 654: Please add "particle" before "size".

Response: Change made:

"If the particle size distributions of the FCDP and 2D-S…"

29. Line 708: Please discuss the latest version of merge files and whether they should be considered "final". Until what time should users expect updates to merge files based on updates of individual instrument files?

Response: We added the following text:

"We caution that it is difficult to consider any version of the merge files as "final" due to the potential for instrument PIs to submit new data sometimes months or even years after flights are completed. However, once new data are submitted, the merge files are typically generated within a month."

30. Line 722: I think that the parenthetical comment should be moved to end of sentence.

Response: Change made:

"To address this, an individual file was generated per flight day that the Falcon flew identifying 14 different leg types with start and stop times per leg in flight (i.e., a single file contains two flights for double flight days)."

31. Line 726: Maybe refer back to Fig. 2 to remind reader of acronyms?

Response: The following text was added:

"The 14 leg types identified include **(see also Fig. 2)**: …"

32. Line 757: Do you mean above the aircraft or do you mean in the direct beam towards the sun? From the rest of the paragraph it seems you are talking about the latter? If so, please change this wording.

Response: Indeed we meant above the aircraft. We felt no edits were needed due to this line explicitly stating that we meant above the aircraft:

"For ACTIVATE, the forward-facing camera on the King Air (sect. 3.6) was used to create a manual cloud mask product indicating whether or not a cloud is present **above the aircraft**."

33. Entire section 5.7: After reading this subsection, I am mildly confused about the resolution of the various satellite products as they are included in these contextual files. The first part of the section speaks to L3 (1×1 deg data), while later on in the section full resolution data is discussed. Could you please state clearly at the beginning of the section which resolutions are included and what the intended use of the gridded vs. native resolution data is? The info is all there I think, it is just presented in a slightly confusing order (to me).

Response: The reviewer's comment is greatly appreciated. We have included the following paragraph to clarify the resolution of the different products:

"The dataset is comprised of products generated at 2 spatial resolutions: 1°×1° and 2 km (satellite pixel resolution). 1°×1° data correspond to aerosol and cloud properties derived from MODIS Aqua (Level 3 product), paired with MERRA-2 meteorological parameters re-gridded to the same resolution. Satellite pixel-level cloud properties are from the Advanced Baseline Imager (ABI) on the 16[th] Geostationary Operational Environmental Satellite (GOES-16), with a continuous spatiotemporal sampling of the ACTIVATE domain. While the Level 3 products are intended for understanding the large-scale and climatological features of the study region, the pixel-level GOES-16 retrievals are valuable for monitoring the spatiotemporal evolution of the cloud fields during research flights."

34. Section 6: I am unclear about the exact motivation for highlighting this particular flight. The first sentence indicates that this flight was representative of the majority of flights, while latter discussion paints the picture of a "golden day". Is it really both? To be clear, I appreciated the discussion, I would like to just understand better whether or not to think of this as a canonical ACTIVATE flight or a special flight when science objectives were particularly well met.

Response: This is a good point. This was a "classic/canonical" type of flight based on how we flew our statistical survey pattern, but the conditions were better than normal which gave it a "golden" feeling. We added the following text:

"While this flight was a canonical type of ACTIVATE flight due to it being a statistical survey, the actual conditions presented qualified this as an excellent flight day as anticipated from the weather forecasting meeting on the previous day. This is because of forecasted cold air outbreak (CAO) indicators of boundary layer instability (Papritz et al., 2015; Painemal et al., 2021; Fletcher et al., 2016) coinciding with strong, cold, northwesterly winds and "cloud streets" (Dadashazar et al., 2021b). The day was forecasted also to have high cloud fraction and no high level cirrus and mid-tropospheric cloud layers that would negatively impact remote sensing objectives. Forecasting analysis conducted the previous day suggested there would be a broken to overcast low cloud deck (deepening to the east) with a western edge moving farther offshore throughout the day."

35. Line 884: I suggest to replace "without" with "no".

Response: Change made:

"The day was forecasted also to have high cloud fraction and **no** high level cirrus and mid-tropospheric cloud layers…"

36. Figure 1: Maybe use slightly thinner lines for flight tracks? They appear as a bit of a blob of lines.

Response: Good idea. We changed the figure as shown here: